# A medullary hub for controlling REM sleep and pontine waves

Amanda L. Schott[1], Justin Baik[1], Shinjae Chung [1] & Franz Weber [1]✉

Rapid-eye-movement (REM) sleep is a distinct behavioral state associated with vivid dreaming and memory processing. Phasic bursts of electrical activity, measurable as spike-like pontine (P)-waves, are a hallmark of REM sleep implicated in memory consolidation. However, the brainstem circuits regulating P-waves, and their interactions with circuits generating REM sleep, remain largely unknown. Here, we show that an excitatory population of dorsomedial medulla (dmM) neurons expressing corticotropin-releasing-hormone (CRH) regulates both REM sleep and P-waves in mice. Calcium imaging showed that dmM CRH neurons are selectively activated during REM sleep and recruited during P-waves, and opto- and chemogenetic experiments revealed that this population promotes REM sleep. Chemogenetic manipulation also induced prolonged changes in P-wave frequency, while brief optogenetic activation reliably triggered P-waves along with transiently accelerated theta oscillations in the electroencephalogram (EEG). Together, these findings anatomically and functionally delineate a common medullary hub for the regulation of both REM sleep and P-waves.

Mammalian sleep occurs in two major forms, rapid-eye-movement (REM) and non-REM (NREM) sleep. REM sleep is characterized by tonic features such as muscle atonia and hippocampal theta oscillations, as well as phasic events such as rapid eye movements, skeletal muscle twitches, and spike-like field potentials called ponto-geniculo-occipital (PGO) waves in cats and monkeys, and pontine (P)-waves in rodents[1,2]. P-waves appear most prominently in the subcoeruleus nucleus of the pons, and they primarily occur during REM sleep and the intermediate sleep stage (IS) that precedes transitions from NREM to REM sleep[3–6]. PGO/P-waves have been shown to coincide with theta bursts, i.e. phasic increases in the power and frequency of theta oscillations in the hippocampus and amygdala[7–9]. Both P-waves and theta bursts are thought to allow for synchronization of neural activity across brain areas, and have been consequently implicated in memory consolidation[10–12]. For example, training in a learning task increases the frequency of P-waves during REM sleep, and performance in subsequent memory tests is positively correlated with P-wave density and phasic theta accelerations[13–15]. As demonstrated in these studies, the frequency of P-waves can strongly differ depending on prior wake experience, suggesting that REM sleep and P-waves are, to a large degree, independently regulated. However, despite their functional importance, circuits that regulate the density of P-waves during REM sleep are still largely unclear, and thus the extent to which these circuits overlap with those controlling REM sleep remains unknown.

The defining electrophysiological and behavioral features of REM sleep are generated by core circuits located in the pons and medulla oblongata. Within the latter, in vivo electrophysiology and c-Fos experiments have demonstrated the existence of REM sleep-active neurons in the dorsomedial medulla (dmM)[16–18]. Electrical stimulation of the dmM increases both the frequency and duration of REM sleep episodes[19], and a recent study showed that optogenetic activation of dmM GABAergic neurons strongly promotes IS and REM sleep[20]. Additionally, calbindin neurons in the dmM have been demonstrated to be necessary for the generation of rapid eye movements during REM sleep, which are temporally correlated with PGO/P-waves in cats and rats[21,22]. Although the medulla is involved in multiple aspects of REM sleep, its causal role in regulating P-waves has not yet been examined.

[1]Department of Neuroscience, Perelman School of Medicine, Chronobiology and Sleep Institute, University of Pennsylvania, Philadelphia, PA 19104, USA.
✉e-mail: fweber@pennmedicine.upenn.edu

While previous work on REM sleep regulation by the dmM has primarily focused on GABAergic neurons, this area contains multiple other cell types expressing various neurotransmitters and neuropeptides, whose roles in brain state control are largely unexplored[23]. Here, we identified a population of excitatory dmM neurons expressing corticotropin-releasing-hormone (CRH) that plays a dual role in regulating both REM sleep and P-waves. Fiber photometry recordings demonstrated that these neurons are selectively active during REM sleep, and their calcium activity during REM sleep specifically correlates with P-waves. Chemogenetic and optogenetic activation of dmM CRH neurons robustly promoted IS and REM sleep, while suppression of this population decreased both states. Bidirectional chemogenetic manipulation showed that the dmM CRH neurons also regulate the frequency of P-waves during IS and REM sleep, and brief optogenetic activation reliably triggered P-waves. In sum, our findings demonstrate that dmM CRH neurons constitute a hub in the REM sleep circuitry that promotes REM sleep with enhanced P-wave density.

## Results

### Excitatory dmM CRH neurons are selectively activated during REM sleep

To identify candidate genes expressed specifically in the dmM, we systematically searched through the in situ hybridization data available at the Allen Brain Atlas (mouse.brain-map.org), and found that the neuropeptide CRH labels a spatially restricted population of neurons within the dmM. Performing fluorescence in situ hybridization (FISH) using a probe against *Crh* mRNA, we confirmed the existence of a distinct population of CRH cells in the dmM, primarily in the nucleus prepositus hypoglossi, and to a lesser extent in the neighboring dorsal paragigantocellular nucleus and medial vestibular nucleus (Fig. 1a; Supplementary Fig. 1a). Surprisingly, the vast majority of dmM CRH neurons co-expressed *Vglut1* (Fig. 1b), a marker for excitatory neurons, which is otherwise sparsely expressed in the brainstem. We also found significant co-labeling with *Vglut2* mRNA, but there was almost no overlap with *Gad2* cells in the region (Fig. 1b; Supplementary Fig. 1b–d), suggesting that the CRH neurons are glutamatergic. To identify downstream brain areas receiving projections from dmM CRH neurons, we performed anterograde tracing by injecting Cre-inducible adeno-associated viruses (AAVs) expressing channelrhodopsin-2 (ChR2-eYFP) into the dmM of CRH-Cre mice (Supplementary Fig. 1e, i). Fluorescence labeled axons were found in the subcoeruleus nucleus of the pons (Supplementary Fig. 1f, i), a structure known to be important for P-wave generation in rats and mice[3,4,24,25]. We also observed dmM CRH projections to the nucleus incertus (NI; Supplementary Fig. 1g–i), a region implicated in modulating the hippocampal theta rhythm that characterizes REM sleep[26,27]. These findings indicate that dmM CRH neurons may be involved in regulating theta oscillations and P-waves, both of which are characteristic of REM sleep, via excitatory interactions with postsynaptic targets in the pons.

To observe the endogenous activity of dmM CRH neurons in vivo during natural sleep, we used fiber photometry to record the calcium activity of this population. We injected Cre-inducible AAVs encoding the fluorescent calcium indicator GCaMP6s into the dmM of CRH-Cre mice (Fig. 1d). We implanted an optic fiber above the dmM to measure the calcium-dependent fluorescence of GCaMP6s (Supplementary Fig. 2a), and monitored brain states – wake, REM sleep, NREM sleep, and IS – using electroencephalogram (EEG) and electromyogram (EMG) recordings. IS is a transitional state that precedes REM sleep or failed attempts to enter REM sleep[28], and is characterized by oscillations in both the delta (δ, 0.5–4 Hz) and theta (θ, 6–10 Hz) range of the EEG (Fig. 1c; Supplementary Fig. 2b) as well as the presence of sleep spindles. We found that dmM CRH neurons are selectively activated during REM sleep (Fig. 1e, f, see Supplementary Table 1 for detailed statistical results). Their activity was steadily rising throughout IS,

significantly increasing from baseline 5 s before entering REM sleep (Supplementary Fig. 2c, d). The calcium activity of this population continued rising during early REM sleep, peaking approximately 25 s after the transition and remaining high until significantly decreasing within 5 s of the termination of REM sleep (Supplementary Fig. 2b, d, e). This observed time course indicates that dmM CRH activity may contribute to both the initiation and maintenance of REM sleep.

Examining the dynamics of the dmM CRH neuron activity during REM sleep, we observed that the calcium signal was associated with brief increases in the power and frequency of the theta band in the EEG (Fig. 1g). To analyze the relationship between CRH neurons and the spectral composition of the EEG in more detail, we used a linear regression model to correlate the EEG spectrogram with the dmM CRH activity during REM sleep (Methods). Using this spectrotemporal correlation analysis, we found that the CRH neural activity is indeed correlated with phasic increases in the power of frequencies in the high theta range (8–15 Hz) (Fig. 1h). In summary, dmM CRH neurons are selectively REM sleep-active, and their calcium activity is associated with transient EEG theta bursts within REM sleep.

### P-waves are correlated with dmM CRH neural activity during REM sleep

Given the known temporal correlation between hippocampal theta bursts and PGO/P-waves in cats, monkeys, and rodents during REM sleep[7,9,25,29], we hypothesized that dmM CRH neurons may be involved in regulating P-waves. To record P-waves in vivo, we implanted a bipolar microelectrode into the subcoeruleus nucleus of the pons (Fig. 2a; Supplementary Fig. 3a). We observed spike-like field potentials occurring primarily during REM sleep as isolated events or as clusters of several waves, while IS and NREM sleep contained only single waveforms (Fig. 2b, c; Supplementary Fig. 3e, f)[5,6]. The frequency of P-waves gradually increased during IS and early REM sleep, and decreased within 5 s of transitioning to wake (Fig. 2d; Supplementary Fig. 3b–d). Consistent with earlier work in cats and rodents[7,24,28], analysis of the mean EEG spectrogram surrounding P-waves during REM sleep showed that these events coincide with transient increases in theta power and frequency (Fig. 2e, f), with clustered P-waves being associated with a significantly larger theta amplification than single P-waves (Supplementary Fig. 3g–i). Interestingly, we found no relationship between P-waves and theta activity during wakefulness (Supplementary Fig. 3j), suggesting that this association is specific to REM sleep.

To determine the temporal relationship between P-waves and dmM CRH activity, we performed simultaneous recordings of CRH calcium activity in the dmM and local field potentials (LFPs) in the pons (Fig. 2g, h). Averaging the calcium-dependent fluorescence signal surrounding each P-wave, we found that P-waves coincide with a sharp rise in dmM CRH neuron activity, with the calcium signal significantly increasing from baseline 0.4 s before the P-wave and peaking approximately 0.5 s afterwards, followed by a slower return to baseline levels (Fig. 2i, j). Both single and clustered P-waves were associated with increased dmM CRH activity, and the amplitude of the CRH calcium signal was significantly higher during clustered P-waves (Fig. 2k; Supplementary Fig. 2f). In control analyses, we found no correlation between the dmM CRH calcium signal and randomly selected time points during REM sleep, nor between P-waves and the amplitude of the calcium-independent isosbestic signal (Supplementary Fig. 2i, j). To quantify the precise time lag between dmM CRH activity and P-waves, we performed for each REM sleep period a cross-correlation of the CRH calcium signal with a binary vector encoding P-wave time points (Supplementary Fig. 2h). We again found that the initial rise in the CRH calcium activity precedes P-waves by 0.4 s, while there was no correlation between the two signals after randomly shifting the time points of P-waves. Finally, comparing the dmM CRH neuron response between brain states, we observed a significant increase in dmM CRH calcium activity following P-waves for every state (Supplementary Fig. 2g). Taken together, our fiber

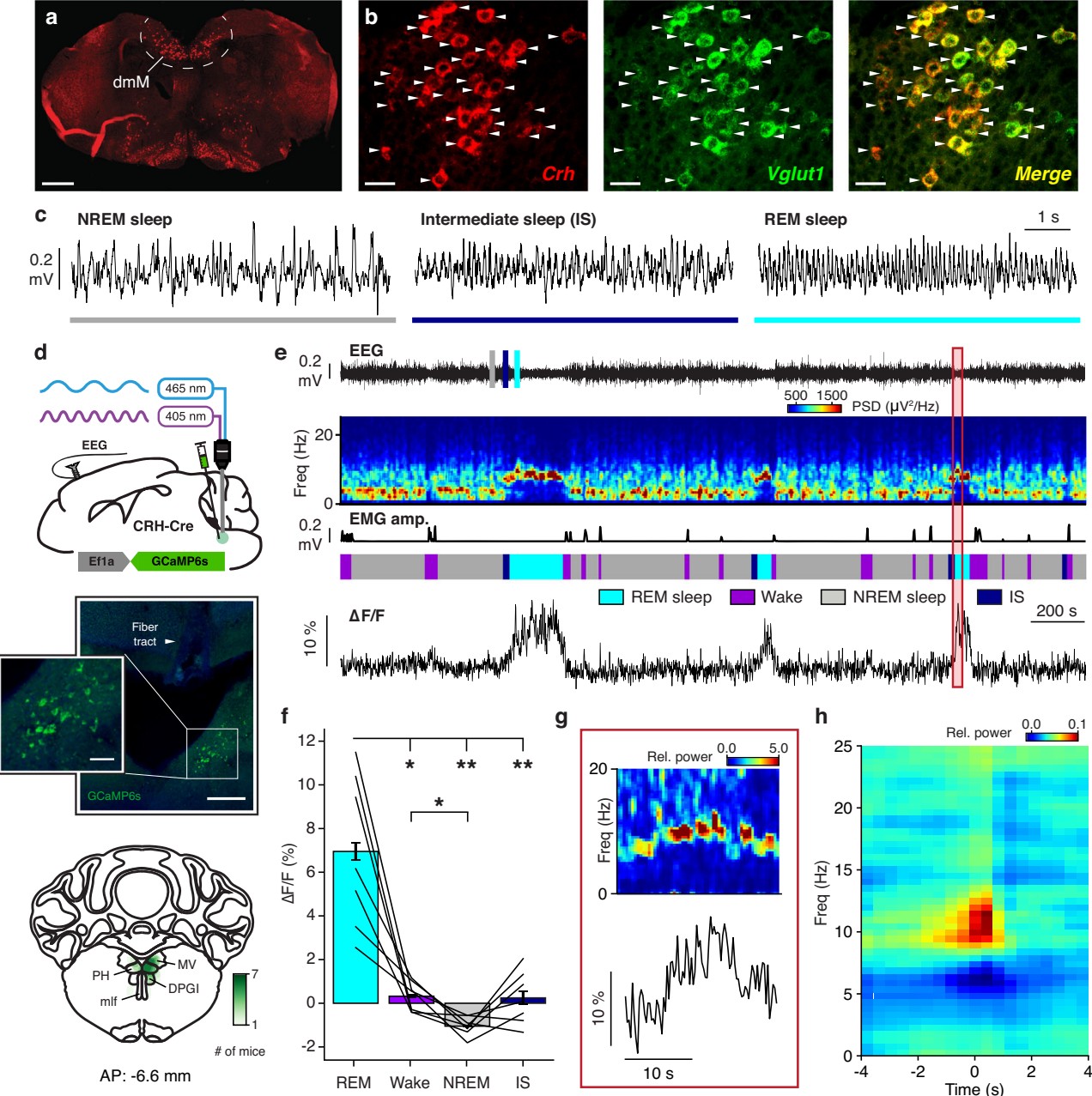

**Fig. 1 | Excitatory dmM CRH neurons are selectively activated during REM sleep. a** Localization of fluorescent probes against *Crh* mRNA in the dmM. Scale bar, 0.5 mm. **b** Localization of fluorescent probes against *Crh* mRNA (red), *Vglut1* mRNA (green), and overlay of both channels (yellow). Arrowheads, cells co-expressing *Crh* and *Vglut1*. Scale bar, 30 µm. **c** Example EEG traces during NREM sleep, IS, and REM sleep, expanded from panel **e**. **d** Top, experimental approach for fiber photometry. Middle, coronal fluorescence images of dmM in a CRH-Cre mouse expressing GCaMP6s (green). Arrowhead, lesion from the optic fiber. Blue, Hoechst stain. Scale bars, 150 µm (main), 40 µm (inset). Bottom, heat map outlining areas with cell bodies expressing GCaMP6s. The green color code indicates the number of mice expressing the virus at the corresponding location (*n* = 7 mice). Coronal brain scheme adapted from the Allen Reference Atlas – Mouse Brain (atlas.brain-map.org). PH, nucleus prepositus; DPGI, dorsal paragigantocellular nucleus; MV, medial vestibular nucleus; mlf, medial longitudinal fasciculus.

**e** Example fiber photometry recording, including EEG trace, EEG spectrogram, EMG amplitude, sleep annotation, and ΔF/F signal. EEG raw traces for selected time points (colored rectangles) are shown on an expanded timescale in panel **c**. Spectrogram and ΔF/F within the red box are shown on an expanded timescale in panel **g**. PSD, power spectral density. **f** Quantification of dmM CRH calcium activity (*n* = 7 mice). One-way repeated measures ANOVA (*P* = 1.38e−6), with Bonferroni post-hoc (REM vs Wake, *P* = 0.022; REM vs NREM, *P* = 0.009; REM vs IS, *P* = 0.008; Wake vs NREM, *P* = 0.035). Statistical details shown in Supplementary Table 1. **g** Example theta burst in the EEG spectrogram with corresponding ΔF/F signal, expanded from panel **e**. Each frequency was normalized by its mean power across the recording. **h** Linear filter mapping the normalized EEG spectrogram onto the dmM CRH neural response (*n* = 7 mice; Methods). Time point 0 s corresponds to the predicted neural response. Error bars indicate ± standard error of the mean (SEM). Lines, individual mice. *\*P* < 0.05; *\*\*P* < 0.01. Source data are provided as a Source Data file.

photometry experiments show that the endogenous dmM CRH neuron activity is closely associated with REM sleep and specifically coincides with P-waves, providing correlative evidence for a role of dmM CRH activity in regulating both REM sleep and P-waves.

**Bidirectional optogenetic manipulation of dmM CRH neurons**
To determine whether dmM CRH neuron activation is sufficient to trigger REM sleep, we injected ChR2-eYFP and implanted an optic fiber into the dmM of CRH-Cre mice (Fig. 3a). We first validated the

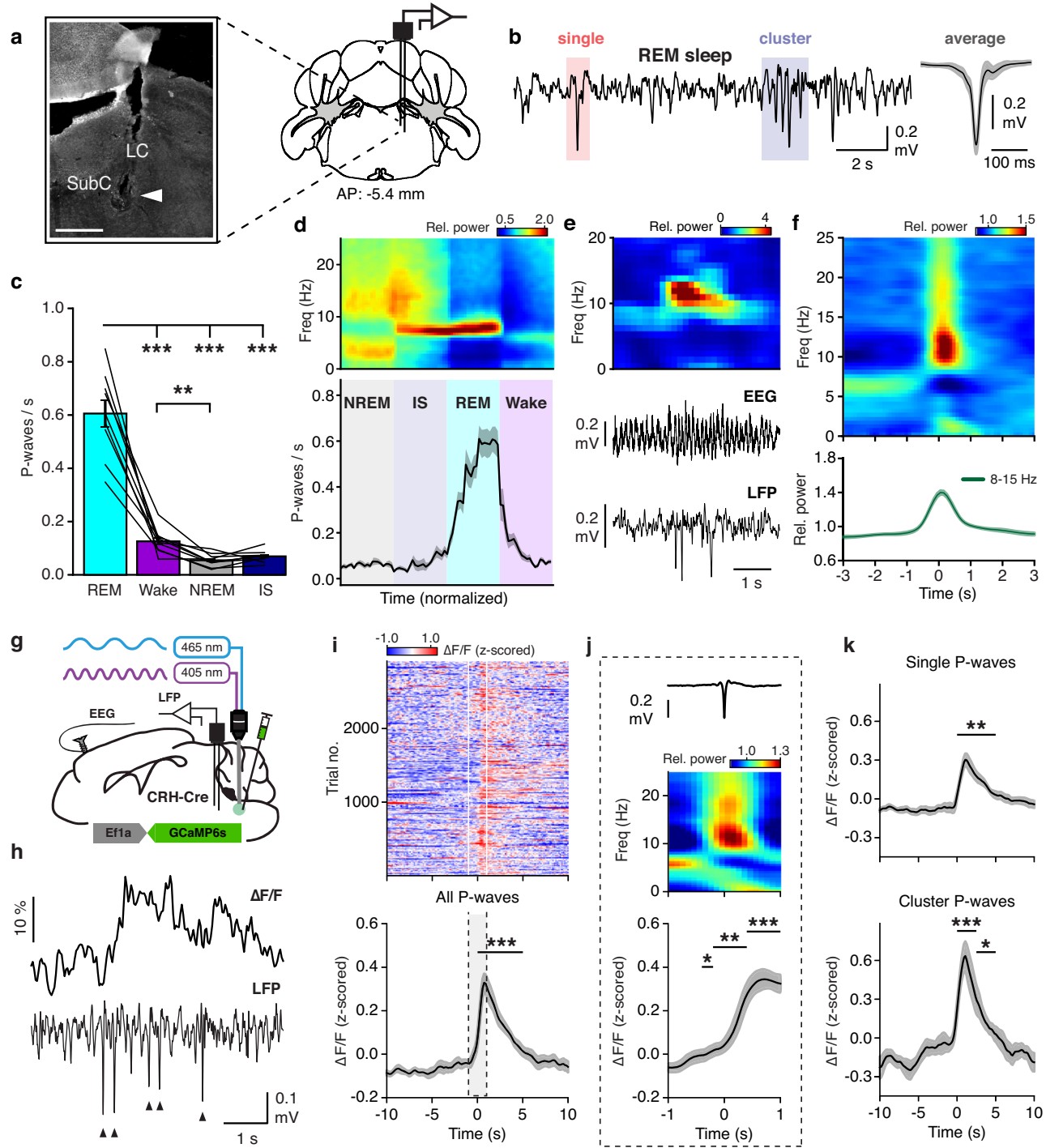

efficacy of optogenetic activation in vivo, using double FISH to quantify *cFos* expression in dmM CRH neurons, and found that repeated laser stimulation produced widespread bilateral excitation of the CRH population, with approximately 90% of the counted CRH neurons co-expressing *cFos* (Supplementary Fig. 4a–c; Methods). To test the role of the dmM CRH neurons in REM sleep regulation, we delivered laser pulse trains with 120 s duration (10 Hz, 473 nm, 2 mW) randomly every 15–25 min over the course of 5 hr recording sessions (Fig. 3b). Optogenetic activation of dmM CRH neurons significantly increased the percentage of REM sleep and IS during the laser stimulation interval, and decreased the percentage of NREM sleep (Fig. 3b–d; Supplementary Fig. 4d). Laser stimulation of the dmM CRH neurons also significantly increased the probability of

successful transitions from IS to REM sleep (Supplementary Fig. 4f). Consistent with this REM sleep-promoting effect, the laser trial-averaged EEG spectrogram showed increases in the theta (θ, 6–10 Hz) and gamma (γ, 55–99 Hz) power during the laser interval, as well as decreases in the delta (δ, 0.5–4 Hz) and sigma (σ, 11–15 Hz) power (Fig. 3e; Supplementary Fig. 4e). In contrast, laser stimulation in eYFP-expressing control mice had no significant effect on the time spent in each brain state, nor on the EEG spectrogram (Supplementary Fig. 4h–k). In ChR2 mice, we found no significant differences in the mean EEG spectrogram between spontaneous and laser-triggered transitions from NREM to REM sleep (Supplementary Fig. 4g), and in both ChR2 and eYFP mice, the power spectral density of the EEG during brain states overlapping with laser stimulation was

**Fig. 2 | P-waves are correlated with dmM CRH neural activity during REM sleep.**
**a** Right, experimental approach for recording P-waves in the subcoeruleus nucleus (SubC). Coronal brain scheme adapted from the Allen Reference Atlas – Mouse Brain (atlas.brain-map.org). Left, image of a lesion caused by LFP electrodes in the dorsal pons. Similar electrode locations were observed in $n = 9$ mice (Supplementary Fig. 3a). Arrowhead, tip of electrode. Scale bar, 300 μm. LC, locus coeruleus. **b** Left, example trace of pontine LFP during REM sleep. Right, mean waveform averaged across all P-waves ($n = 9$ mice). Shading, ±standard deviation (SD). **c** Average frequency of P-waves in each brain state ($n = 9$ mice). One-way repeated measures ANOVA ($P = 1.15e{-}13$) with Bonferroni post-hoc (REM vs Wake, $P = 3.97e{-}4$; REM vs NREM, $P = 5.32e{-}5$; REM vs IS, $P = 1.15e{-}13$; Wake vs NREM, $P = 0.007$). **d** Averaged EEG spectrogram (top) and P-wave frequency (bottom) during NREM → IS → REM→wake transitions ($n = 9$ mice). The duration of each state episode was normalized in time. **e** Example theta burst during REM sleep, including EEG spectrogram, EEG raw trace, and pontine LFP trace. **f** Top, averaged EEG spectrogram surrounding P-waves during REM sleep ($n = 9$ mice). Bottom, mean normalized power in the high theta frequency band (8–15 Hz). Time point 0 s corresponds to the negative peak of P-waves. **g** Experimental approach for

simultaneously recording P-waves and dmM CRH neuron calcium activity. **h** Example traces of the ΔF/F signal and pontine LFP during REM sleep. Arrowheads indicate P-waves. **i** Z-scored ΔF/F signal surrounding P-waves during all brain states ($n = 8$ mice). Top, color-coded ΔF/F activity surrounding each P-wave (rows). Bottom, averaged ΔF/F signal. Baseline interval, −10 to −7.5 s. Paired t-tests ($P_{0-2.5s} = 1.29e{-}4$; $P_{2.5-5s} = 9.27e{-}4$, Bonferroni-corrected). **j** Averaged LFP signal, EEG spectrogram, and ΔF/F signal during the 2 s interval surrounding P-waves ($n = 8$ mice; expanded from panel **i**. Baseline interval, −1 to −0.8 s. Paired t-tests ($P_{-0.4--0.2s} = 0.021$; $P_{-0.2-0s} = 0.004$; $P_{0-0.2s} = 0.005$; $P_{0.2-0.4s} = 0.002$; $P_{0.4-0.6s} = 8.70e{-}5$; $P_{0.6-0.8s} = 1.16e{-}4$; $P_{0.8-1s} = 1.73e{-}4$, Bonferroni-corrected). **k** ΔF/F signal surrounding single (top) and clustered (bottom) P-waves ($n = 8$ mice). Baseline interval, −10 to −7.5 s. Paired t-tests (single, $P_{0-2.5s} = 0.003$; $P_{2.5-5s} = 0.008$; clustered, $P_{0-2.5s} = 6.54e{-}4$; $P_{2.5-5s} = 0.028$, Bonferroni-corrected). All statistical comparisons were two-tailed. EEG spectrograms were normalized at each frequency by its mean power across the recording (**d**, **e**) or across the shown interval (**f**, **j**). Error bars and shadings indicate ±SEM unless otherwise stated. Lines, individual mice. *$P < 0.05$; **$P < 0.01$; ***$P < 0.001$. Statistical details (**c**, **i**–**k**) shown in Supplementary Table 1. Source data are provided as a Source Data file.

---

statistically indistinguishable from that during episodes without laser (Supplementary Fig. 4m).

To determine whether dmM CRH neurons are involved in maintaining REM sleep, we applied a closed-loop protocol to activate or inhibit this population specifically during REM sleep episodes (Fig. 3f). For these experiments, the brain state was automatically classified in real time (Methods). Each detected REM episode was randomly assigned as a "laser-on" or a "laser-off" bout with 50% probability, and laser stimulation was applied for the entire duration of laser-on REM bouts. We found that in ChR2 mice, closed-loop activation significantly prolonged REM sleep episodes relative to laser-off REM bouts, while administration of laser pulse trains did not affect the duration of REM sleep periods in eYFP control mice (Fig. 3g, h left).

To determine the likelihood of observing the same difference in duration between laser-on and laser-off REM sleep periods by chance, given the same distribution of REM sleep durations as in our ChR2 dataset, we randomly shuffled the designations of laser-on and laser-off REM bouts. Using bootstrapping (Methods), we then estimated the sampling distribution of the mean difference in duration between the shuffled laser-on and laser-off REM episodes. For comparison, we also determined the distribution of the mean difference in duration for the true (non-shuffled) REM sleep periods in both ChR2 and eYFP mice. The mean difference for the shuffled ChR2 dataset was not significantly different from 0 s, and the true mean difference in ChR2 mice was significantly increased compared with that obtained for the shuffled dataset and the eYFP mice (Fig. 3h right), demonstrating that the increased duration of laser-on episodes in ChR2 mice is the result of dmM CRH neuron activation.

To test whether dmM CRH neurons play a necessary role in REM sleep maintenance, we selectively expressed a light-activated chloride channel (iC++-eYFP) in this population and delivered continuous inhibition during laser-on REM episodes. Inhibition of dmM CRH neurons significantly reduced the duration of REM sleep bouts in iC++ mice but not in eYFP controls (Fig. 3i left), and estimation analysis verified that the reduced duration of laser-on REM sleep episodes in iC ++ mice is a true laser effect (Fig. 3i right).

The mean duration of laser-off REM episodes in ChR2 mice was nearly 20 s shorter than laser-off REM episodes in eYFP mice, while laser-off bouts in iC++ mice were approximately 20 s longer than in their eYFP controls. However, the mean duration of all REM sleep episodes (averaged across both laser-on and laser-off bouts) did not differ between ChR2 and eYFP mice, nor between iC++ and eYFP mice (Supplementary Fig. 4l bottom). There was also no significant difference in the total percent time spent in REM sleep between control and experimental groups (Supplementary Fig. 4l top). These results indicate that extending or shortening the duration of laser-on REM sleep

periods by optogenetic excitation or inhibition of dmM CRH neurons impacts the duration of subsequent laser-off REM episodes, an effect which is likely the result of homeostatic mechanisms controlling the cumulative amount of REM sleep over time. This observation is in line with a previous study showing that the durations of two successive REM sleep periods are anti-correlated[30]. Thus, laser-on REM sleep episodes prolonged (or shortened) by dmM CRH neuron manipulation are expected to be followed by shorter (or longer) laser-off REM episodes, a pattern which consequently preserves the overall duration and amount of REM sleep across experimental groups.

**Brief optogenetic activation of dmM CRH neurons elicits P-waves**

The above experiments showed that activation of dmM CRH neurons robustly initiates and maintains REM sleep. To determine whether this population is also important for regulating P-waves, we delivered brief laser pulses (10–20 ms) in mice expressing ChR2-eYFP in dmM CRH neurons, while simultaneously recording LFPs in the pons (Fig. 4a). We found that transient activation of dmM CRH neurons reliably elicited spike-like waveforms that were equivalent to spontaneous P-waves in shape, measured by amplitude and half-width duration (Fig. 4b–d; Supplementary Fig. 5a, b). Importantly, as found for spontaneous P-waves, laser-triggered P-waves coincided with phasic amplifications and accelerations in the EEG theta oscillations during REM sleep (Fig. 4c, d), and the increase in theta power surrounding laser-triggered clustered P-waves was larger than that surrounding single waves (Supplementary Fig. 5c). To directly compare the EEG during spontaneous and laser-triggered P-waves, we examined the normalized power spectral density within 1 s intervals surrounding each event, and found that the spectral profile accompanying laser-triggered P-waves was indistinguishable from the profile for spontaneous P-waves (Fig. 4f). Both profiles were characterized by an increased power of frequencies in the high theta range (8–15 Hz) and above. In contrast, laser pulses that failed to evoke a P-wave were not associated with substantial LFP deflections or increases in theta power (Fig. 4e). The spectral profile for failed laser pulses closely resembled that for randomly selected time points during REM, and these profiles were significantly different from those of spontaneous and laser-triggered P-waves (Fig. 4f). These analyses provide validation that P-waves evoked by dmM CRH neuron excitation are physiologically equivalent to spontaneously occurring P-waves.

Approximately 14% of all P-waves recorded in ChR2 mice were optogenetically evoked, and the efficacy of dmM CRH activation in eliciting P-waves was dependent on the brain state of the mouse (Fig. 4g; Supplementary Fig. 5h, i). The probability that a laser pulse was successfully followed by a P-wave within 100 ms was above 90%

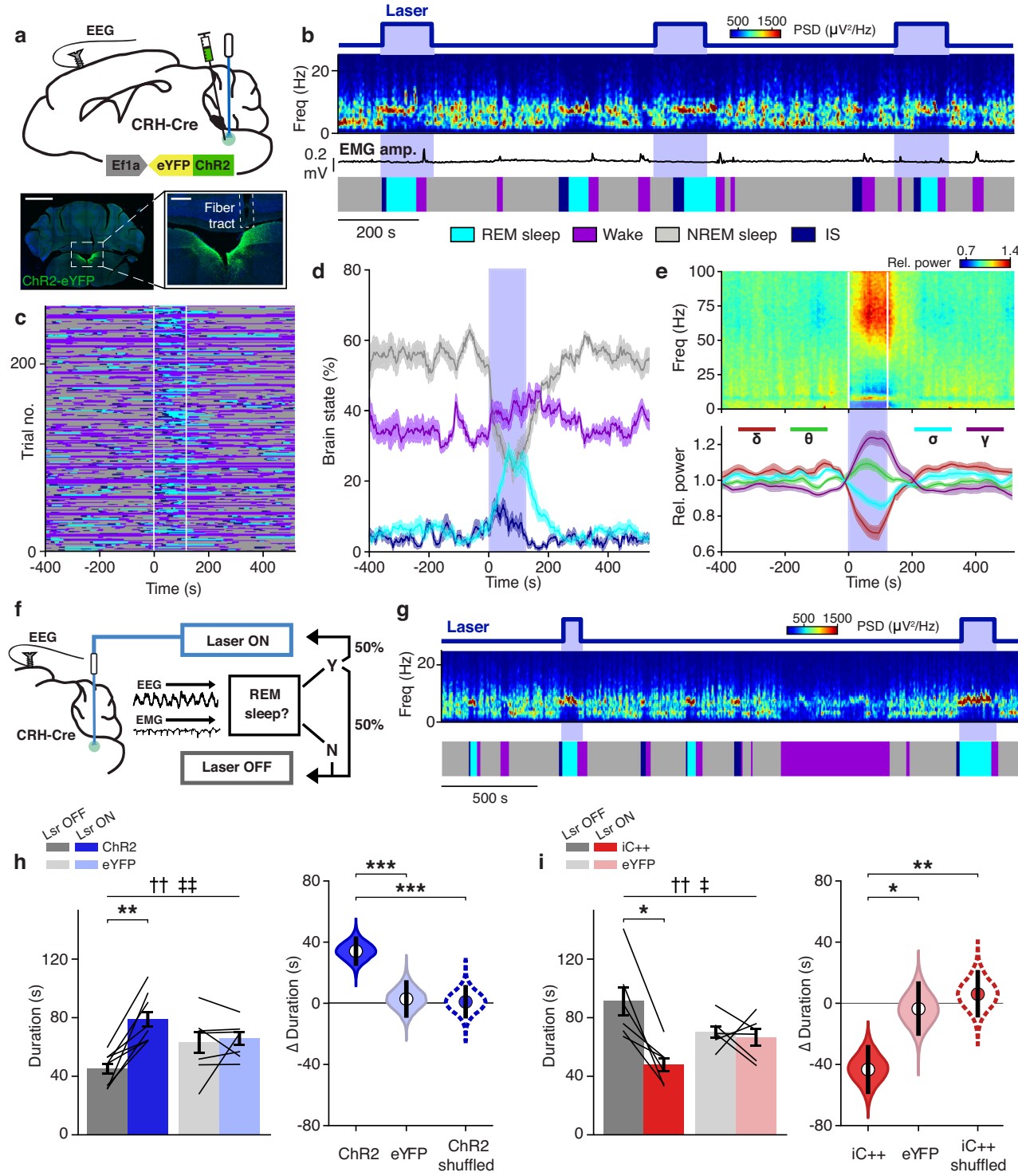

during NREM sleep and IS, 66% during REM sleep, and only 34% during wake. In contrast, fewer than 5% of laser pulses in eYFP controls were succeeded by a P-wave (Supplementary Fig. 5g), and these laser-associated P-waves constituted only 0.4% of the total number of P-waves in eYFP mice (Supplementary Fig. 5i). As an additional control, we jittered the time points of the laser pulses during REM sleep to generate a set of randomly shuffled pulses (Methods). Similar to the laser success rate in eYFP mice, only 4% of time-shuffled pulses were closely followed by a P-wave (Fig. 4g right), corroborating that the optogenetic induction of P-waves in ChR2 mice is a direct result of dmM CRH neuron activation. We also measured the delay time

between the laser onset and the negative peak of the elicited P-wave, and found that the latency for true laser-triggered P-waves forms a bell-shaped distribution with a clear maximum at 30 ms, while the P-waves associated with randomly shuffled pulses occurred with equal prob-ability across the 100 ms detection window (Fig. 4h).

To examine the relationship between P-wave timing and theta activity during REM sleep, we assessed whether these events were modulated by the phase of the theta oscillations. We found that both spontaneous and laser-triggered P-waves exhibited significant phase preferences (Fig. 4i; Methods), with the greatest proportion of P-waves occurring during the rising phase (−π to 0 rad) of the theta oscillation,

**Fig. 3 | Optogenetic manipulation of dmM CRH neurons bidirectionally modulates REM sleep expression. a** Top, experimental approach for optogenetically manipulating dmM CRH neurons. Bottom, coronal fluorescence images of dmM in a CRH-Cre mouse expressing ChR2-eYFP (green). Blue, Hoechst stain. Scale bars, 1.5 mm (left), 300 μm (right). **b** Example optogenetic experiment, including laser stimulation trials, EEG spectrogram, EMG amplitude, and sleep annotation. **c** Behavioral state of mice surrounding each laser stimulation trial (rows; $n = 8$ mice). **d** Percentages of REM sleep, NREM sleep, IS, and wake surrounding the laser stimulation interval (blue shading; $n = 8$ mice). **e** Impact of laser stimulation on the EEG spectrogram and different power bands ($n = 8$ mice). Top, laser-trial-averaged EEG spectrogram. Each frequency was normalized by its mean power across the recording. Bottom, time course of δ (0.5–4 Hz), θ (6–10 Hz), σ (11–15 Hz), and γ (55–99 Hz) power. **f** Schematic of closed-loop laser stimulation protocol (Methods). **g** Example recording with closed-loop excitation in a ChR2 mouse. **h** Left, mean duration of REM sleep episodes with and without laser in ChR2 ($n = 8$) and eYFP

mice ($n = 7$) during closed-loop activation. Mixed ANOVA ($P_{laser} = 0.001$; $P_{laserXvirus} = 0.006$) with Holm post-hoc ($P = 0.002$). Right, bootstrapped sampling distributions of the mean duration difference between laser-on and laser-off REM sleep episodes in ChR2 and eYFP mice, as well as in ChR2 mice after randomly shuffling REM sleep episodes (Methods). Equal-tail bootstrap ($P_{ChR2vsEYFP} = 9.00e-4$; $P_{ChR2vsShuffledChR2} = 2.00e-4$, uncorrected). **i** Left, mean duration of REM sleep episodes with and without laser in iC++ ($n = 6$) and eYFP mice ($n = 6$) during closed-loop inhibition. Mixed ANOVA ($P_{laser} = 0.005$; $P_{laserXvirus} = 0.012$) with Holm post-hoc ($P = 0.013$). Right, bootstrapped sampling distributions (equal-tail bootstrap, $P_{iC++vsEYFP} = 0.015$; $P_{iC++vsShufflediC++} = 0.002$, uncorrected). For violin plots, center dots represent the distribution mean, and black lines represent the 95% confidence interval (CI). Error bars and shadings indicate ±SEM. Lines, individual mice. † denotes main effect of laser; ‡ denotes interaction between laser and virus. *$P < 0.05$; **$P < 0.01$; ***$P < 0.001$. Statistical details (**h**, **i**) shown in Supplementary Table 1. Source data are provided as a Source Data file.

as previously reported in mice[25]. Additionally, the success rate of laser pulses during REM sleep was negatively correlated with the mean power and frequency of theta oscillations immediately preceding laser stimulation (Supplementary Fig. 5d–f), suggesting that the timing of P-waves is gated by the theta activity. The dependence of P-wave induction on the phase, power, and frequency of the theta oscillations may explain the lower efficacy of 66% for laser stimulation during REM sleep, compared with the 90% success rate during NREM sleep (Fig. 4g).

Phasic REM sleep is characterized by a variety of transient events, including rapid eye movements and skeletal muscle twitches in addition to P-waves[2,31]. To determine whether the dmM CRH neurons may play a role in regulating phasic muscle activity, we analyzed the nuchal EMG surrounding laser-induced and spontaneous P-waves. We found that both types of P-waves coincided with large, localized deflections in the raw EMG signal (Supplementary Fig. 5j, k top), and analysis of the mean EMG amplitude revealed sharp increases in muscle activity during the 100 ms intervals surrounding both events, compared with the preceding baseline intervals (Supplementary Fig. 5k bottom). Interestingly, laser pulses that failed to elicit P-waves were also not associated with significant changes in the EMG signal. These findings suggest that P-waves may be a gateway to both theta bursts and muscle twitches, since neither of these events can be triggered by dmM CRH neuron stimulation in the absence of a P-wave.

**Chemogenetic manipulation of dmM CRH neurons bidirectionally modulates REM sleep amount and P-wave frequency**
Our optogenetic experiments showed that minutes-long excitation or inhibition of dmM CRH neurons is sufficient to promote or reduce REM sleep, respectively, while brief stimulation (10–20 ms) reliably triggered P-waves. To examine the effects of sustained changes in dmM CRH neuron activity on REM sleep and P-waves, we used designer receptors exclusively activated by designer drugs (DREADDs) to chemogenetically activate or suppress the activity of this population. We injected Cre-dependent AAVs expressing either excitatory (hM3D(Gq)-mCherry) or inhibitory (hM4D(Gi)-mCherry) DREADDs into the dmM of CRH-Cre mice (Fig. 5a), and recorded EEG, EMG, and LFP signals for 5 h following intraperitoneal injection of the DREADD agonist clozapine-N-oxide (CNO). To control for potential off-target effects of CNO[32,33], we also performed these experiments in mice expressing only mCherry.

Performing double FISH for *cFos* and *Crh* after CNO injection in hM3D(Gq) mice revealed extensive *cFos* expression in dmM CRH neurons, confirming that chemogenetic excitation effectively activates these neurons in vivo (Supplementary Fig. 6a, b). Administration of CNO to hM3D(Gq) mice significantly increased the percent time spent in REM sleep and the frequency of REM episodes compared with saline injections in the same animals and with CNO-treated mCherry controls (Fig. 5b–e). In contrast, CNO injection in

hM4D(Gi) animals significantly decreased both the amount and frequency of REM sleep (Fig. 5h, i). Chemogenetic inhibition of dmM CRH neurons also reduced the mean duration of REM sleep bouts (Fig. 5j), while there was no effect on the duration in the excitatory DREADD experiment (Fig. 5f). Examining the impact of chemogenetic manipulations on other brain states, we found that CNO treatment in hM3D(Gq) mice decreased the amount of wake (Supplementary Fig. 6c) and increased the amount of IS (Supplementary Fig. 6e), while hM4D(Gi) animals in the CNO condition spent less time in IS and more time in NREM sleep (Supplementary Fig. 7a–c). Notably, DREADD activation of dmM CRH neurons significantly increased the likelihood of transitions from IS to REM sleep (Fig. 5g), while inhibition reduced transitions to IS (Fig. 5k). This result is consistent with the observed increase in the IS → REM transition probability observed for optogenetic excitation of dmM CRH neurons (Supplementary Fig. 4f), providing further evidence that their activity is important for facilitating transitions to REM sleep. None of the above metrics were impacted by CNO administration in mCherry mice (Fig. 5d–k; Supplementary Fig. 6c–e; Supplementary Fig. 7a–c), and CNO did not significantly change the delta, theta, sigma, or beta power in the EEG of the hM3D(Gq), hM4D(Gi), or mCherry groups (Supplementary Fig. 6h; Supplementary Fig. 7f). Combined with our optogenetic findings, these experiments highlight the important role of dmM CRH neuron activity in REM sleep regulation.

To determine whether chemogenetic manipulation of dmM CRH neuron activity also modulates P-waves in addition to REM sleep, we compared the P-wave frequency between saline and CNO conditions in hM3D(Gq), hM4D(Gi), and mCherry groups. In mice expressing the excitatory DREADD, treatment with CNO significantly increased the P-wave frequency during REM sleep (Fig. 6a–c) and IS (Supplementary Fig. 6f, g) compared with saline injection, while no difference was found for mice expressing only mCherry. In contrast, the P-wave frequency during REM sleep and IS was reduced in the CNO condition for hM4D(Gi) mice, but not for mCherry controls (Fig. 6d, e; Supplementary Fig. 7d, e). To determine the likelihood of observing the same significant effects in experiments with identical sample size and inter-mouse variability in the P-wave density, we estimated for each experimental group the sampling distribution of the mean difference in P-wave frequency between saline and CNO trials (Methods). Comparing the CNO-induced change in P-wave frequency between DREADD-expressing and control animals, we found no overlap between the 95% CIs calculated for hM3D(Gq) and mCherry mice (Fig. 6c), nor between the CIs obtained for hM4D(Gi) mice and their mCherry counterparts (Fig. 6e). These findings demonstrate that sustained activation or inactivation of dmM CRH neurons results in the facilitation or reduction, respectively, of both REM sleep and P-waves.

In contrast to our optogenetic experiments, DREADD activation did not impact the P-wave frequency during NREM sleep, a finding

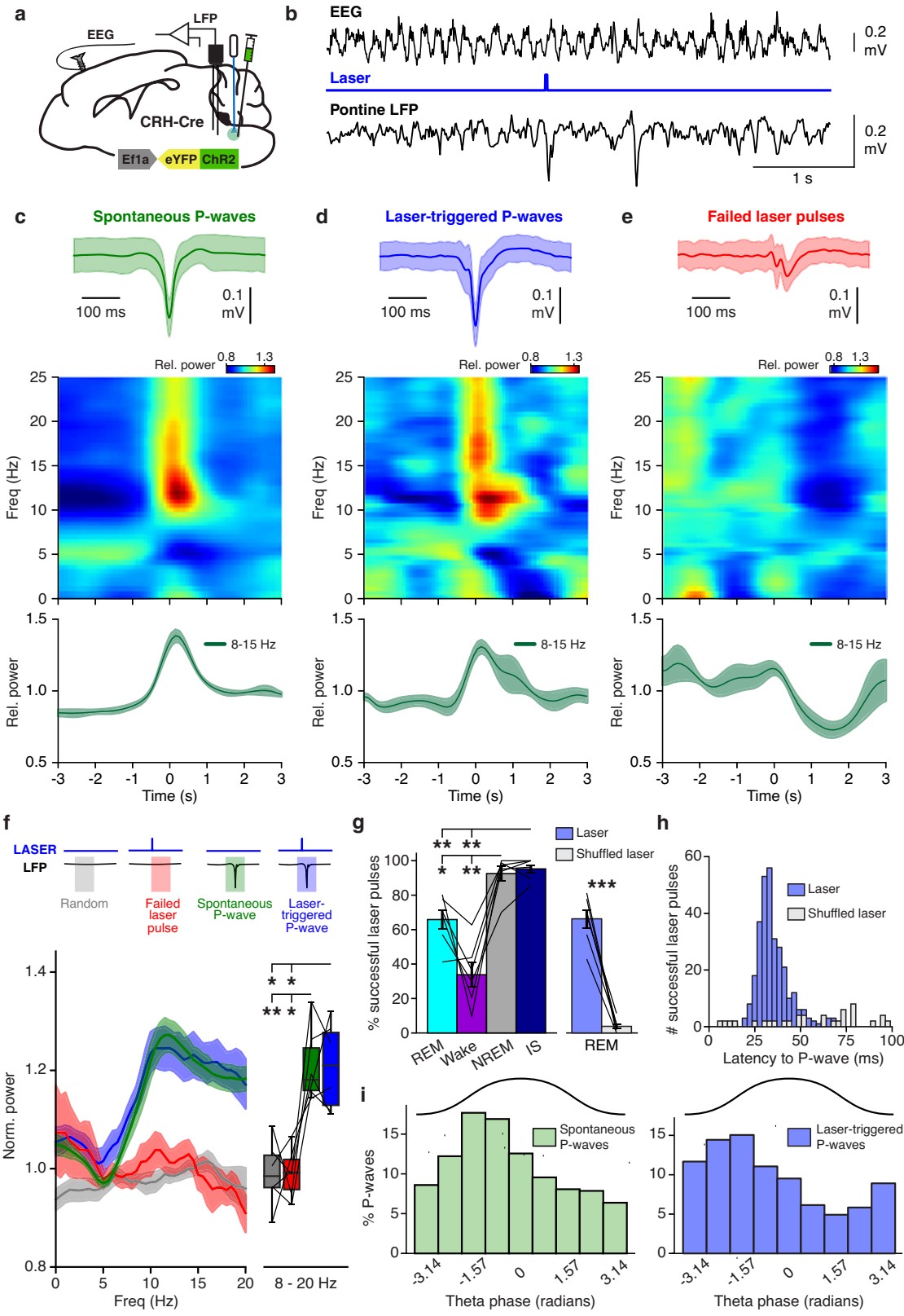

which may be explained by the state-dependent activity of dmM CRH neurons. As endogenous dmM CRH activity is low and stable during NREM sleep (Fig. 1e, f), tonic activation through hM3D(Gq) may not produce sufficiently large activity peaks required to elicit a P-wave. In contrast, since dmM CRH neuron activity strongly fluctuates during REM sleep, sustained excitation likely increases the chance that spikes in the calcium activity surpass the threshold for P-wave induction.

## Discussion

Our experiments demonstrate that CRH-expressing medullary neurons regulate both REM sleep and P-waves in mice, thus forming a hub promoting REM sleep with increased P-wave density. Using fiber photometry, we showed that dmM CRH neurons are selectively active during REM sleep (Fig. 1), and that the activity of this population is temporally correlated with P-waves and theta bursts (Fig. 2).

**Fig. 4 | Brief optogenetic activation of dmM CRH neurons elicits P-waves.**
**a** Experimental approach for optogenetically triggering and recording P-waves.
**b** Example recording with spontaneous and a laser-triggered P-wave during REM
sleep, including EEG trace, laser stimulation trial, and LFP trace. **c–e** Average
waveform and EEG correlates of spontaneous P-waves (**c**), laser-triggered P-waves
(**d**), and failed laser pulses (**e**) during REM sleep ($n = 6$ mice). Top, averaged LFP
signal (shading, ±SD); middle, averaged EEG spectrogram; bottom, averaged power
in the high theta frequency band (8–15 Hz). Time point 0 s corresponds to the
negative peak of P-waves (**c**, **d**) or onset of failed laser pulses (**e**). **f** Left, normalized
power spectral density during the 1 s interval surrounding spontaneous P-waves,
laser-triggered P-waves, failed laser pulses, and randomly selected control
time points during REM sleep ($n = 6$ mice). Right, mean normalized power within
the 8–20 Hz range. One-way repeated measures ANOVA ($P = 1.06\text{e}{-5}$) with Bon-
ferroni post-hoc (control vs spontaneous P-waves, $P = 0.002$; control vs laser-
triggered P-waves, $P = 0.025$; failed laser vs spontaneous P-waves, $P = 0.021$; failed
laser vs laser-triggered P-waves, $P = 0.017$). For boxplots, center lines represent the

median, box limits represent the interquartile range, and whiskers represent the
remaining distribution. **g** Left, percentage of laser pulses successfully triggering a
P-wave for each brain state ($n = 6$ mice). One-way repeated measures ANOVA
($P = 8.95\text{e}{-7}$) with Bonferroni post-hoc (REM vs NREM, $P = 0.036$; REM vs IS,
$P = 0.005$; Wake vs NREM, $P = 0.005$; Wake vs IS, $P = 0.004$). Right, success rate of
true vs time-shuffled laser pulses during REM sleep. Paired t-test ($P = 1.36\text{e}{-4}$).
**h** Latencies of laser-triggered P-waves for true and time-shuffled laser pulses ($n = 6$
mice). **i** Phase preferences of spontaneous and laser-triggered P-waves relative to
EEG theta oscillations ($n = 6$ mice). Both phase distributions were non-uniform
(Rayleigh test, spontaneous P-waves, $P = 1.83\text{e}{-13}$; laser-triggered P-waves,
$P = 9.06\text{e}{-9}$). For analysis of spectrograms (**c–e**) and PSDs (**f**), each frequency
component was normalized by its mean power across the 6 s interval surrounding
the P-wave or failed laser onset. Error bars and shadings indicate ±SEM unless
otherwise stated. Lines, individual mice. *$P < 0.05$; **$P < 0.01$; ***$P < 0.001$. Statistical
details (**f**, **g**) shown in Supplementary Table 1. Source data are provided as a Source
Data file.

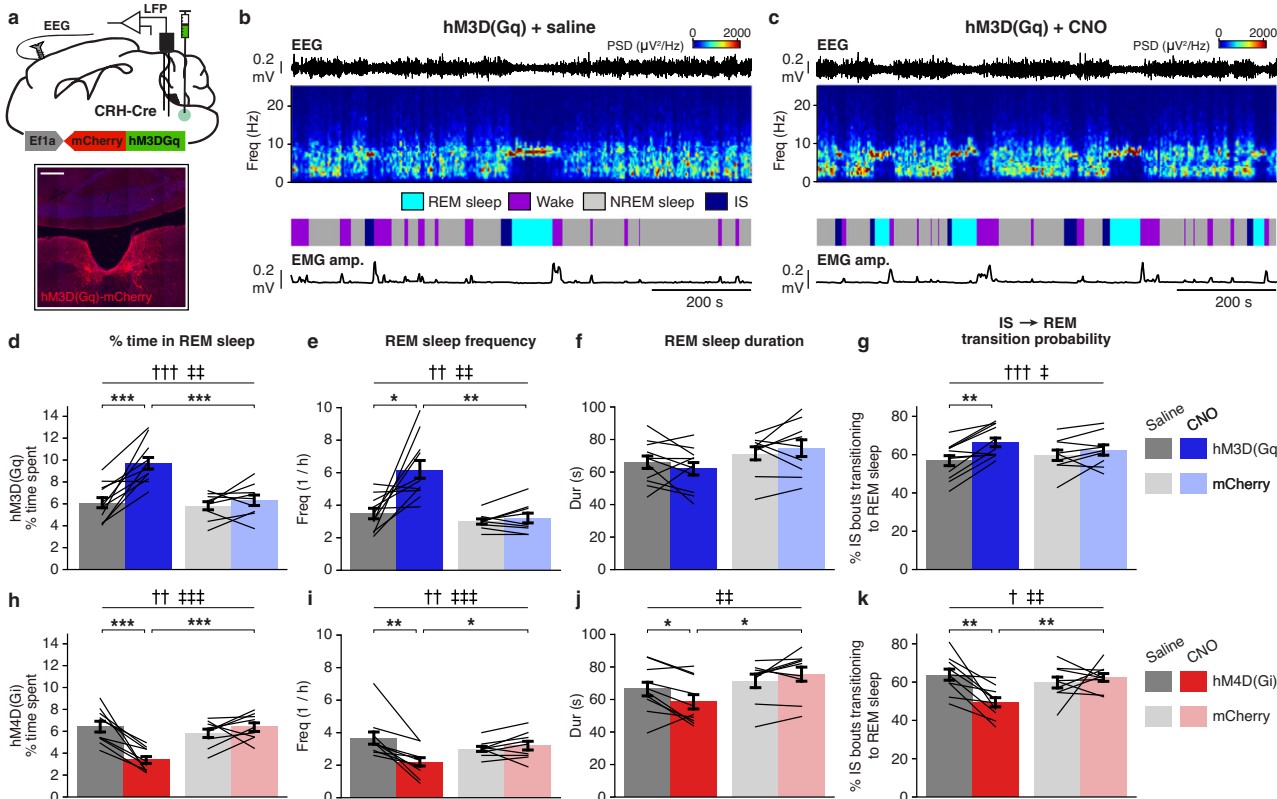

**Fig. 5 | Chemogenetic manipulation of dmM CRH neurons bidirectionally
modulates REM sleep amount. a** Top, experimental approach for chemogeneti-
cally manipulating dmM CRH neurons and recording P-waves. Bottom, coronal
fluorescence image of dmM in a CRH-Cre mouse expressing hM3D(Gq)-mCherry
(red). Blue, Hoechst stain. Scale bar, 300 μm. **b**, **c** Example saline (**b**) and CNO (**c**)
trials in an hM3D(Gq) mouse, including EEG trace, EEG spectrogram, sleep annota-
tion, and EMG amplitude. **d** Percent time spent in REM sleep during saline vs CNO
trials in hM3D(Gq) ($n = 11$) and mCherry mice ($n = 9$). Mixed ANOVA ($P_{\text{drug}} = 3.55\text{e}{-5}$;
$P_{\text{drugXvirus}} = 0.001$) with Holm post-hoc (hM3D(Gq), saline vs CNO, $P = 3.65\text{e}{-4}$;
CNO, hM3D(Gq) vs mCherry, $P = 6.38\text{e}{-4}$). **e** Frequency of REM sleep episodes in
hM3D(Gq) ($n = 11$) and mCherry mice ($n = 9$). Mixed ANOVA ($P_{\text{drug}} = 0.002$;
$P_{\text{drugXvirus}} = 0.008$) with Holm post-hoc (hM3D(Gq), saline vs CNO, $P = 0.010$; CNO,
hM3D(Gq) vs mCherry, $P = 0.001$). **f** REM sleep duration in hM3D(Gq) ($n = 11$) and
mCherry mice ($n = 9$). **g** Percentage of IS bouts resulting in a transition to REM sleep
in hM3D(Gq) ($n = 11$) and mCherry mice ($n = 9$). Mixed ANOVA ($P_{\text{drug}} = 4.10\text{e}{-4}$;
$P_{\text{drugXvirus}} = 0.026$) with Holm post-hoc (hM3D(Gq), saline vs CNO, $P = 0.002$).

**h** Percent time spent in REM sleep during saline vs CNO trials in hM4D(Gi) ($n = 10$)
and mCherry mice ($n = 9$). Mixed ANOVA ($P_{\text{drug}} = 0.003$; $P_{\text{drugXvirus}} = 1.84\text{e}{-4}$) with
Holm post-hoc (hM4D(Gi), saline vs CNO, $P = 8.19\text{e}{-4}$; CNO, hM4D(Gi) vs mCherry,
$P = 3.73\text{e}{-5}$). **i** Frequency of REM sleep episodes in hM4D(Gi) ($n = 10$) and mCherry
mice ($n = 9$). Mixed ANOVA ($P_{\text{drug}} = 0.003$; $P_{\text{drugXvirus}} = 6.79\text{e}{-4}$) with Holm post-hoc
(hM4D(Gi), saline vs CNO, $P = 0.003$; CNO, hM4D(Gi) vs mCherry, $P = 0.041$). **j** REM
sleep duration in hM4D(Gi) ($n = 10$) and mCherry mice ($n = 9$). Mixed ANOVA
($P_{\text{drug}} = 0.378$; $P_{\text{drugXvirus}} = 0.005$) with Holm post-hoc (hM4D(Gi), saline vs CNO,
$P = 0.046$; CNO, hM4D(Gi) vs mCherry, $P = 0.028$). **k** Percentage of IS bouts
resulting in a transition to REM sleep in hM4D(Gi) ($n = 10$) and mCherry mice ($n = 9$).
Mixed ANOVA ($P_{\text{drug}} = 0.022$; $P_{\text{drugXvirus}} = 0.005$) with Holm post-hoc (hM4D(Gi),
saline vs CNO, $P = 0.004$; CNO, hM4D(Gi) vs mCherry, $P = 0.003$). Error bars indi-
cate ±SEM. Lines, individual mice. † denotes main effect of drug (saline vs CNO); ‡
denotes interaction between drug and virus. *$P < 0.05$; **$P < 0.01$; ***$P < 0.001$. Sta-
tistical details (**d–k**) shown in Supplementary Table 1. Source data are provided as a
Source Data file.

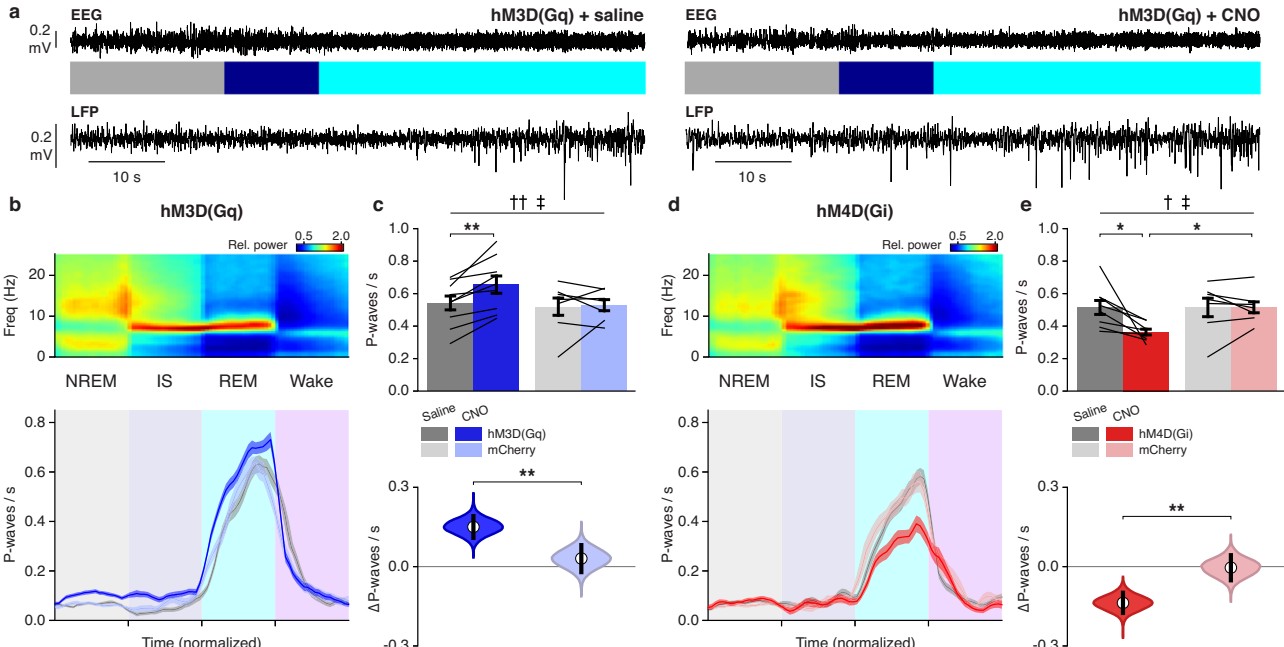

**Fig. 6 | Chemogenetic manipulation of dmM CRH neurons bidirectionally modulates P-wave frequency. a** Example recordings from saline (left) and CNO (right) trials in an hM3D(Gq) mouse, including EEG trace, sleep annotation, and pontine LFP. **b** Top, EEG spectrogram during NREM → IS → REM→wake transitions, averaged across mice. Bottom, mean P-wave frequency for saline and CNO trials in hM3D(Gq) mice ($n = 9$) and CNO trials in mCherry mice ($n = 7$). The duration of each state episode was normalized in time. **c** Top, mean P-wave frequency in REM sleep during saline vs CNO trials in hM3D(Gq) ($n = 9$) and mCherry mice ($n = 7$). Mixed ANOVA ($P_{drug} = 0.004$; $P_{drugXvirus} = 0.014$) with Holm post-hoc (hM3D(Gq), saline vs CNO, $P = 0.001$). Bottom, bootstrapped sampling distributions of the mean difference in P-wave frequency between CNO and saline trials in hM3D(Gq) and mCherry mice (Methods). Equal-tail bootstrap ($P = 0.009$). **d** Mean EEG spectrogram (top) and P-wave frequency (bottom) during NREM → IS → REM→wake

transitions for saline and CNO trials in hM4D(Gi) mice ($n = 8$) and CNO trials in mCherry mice ($n = 7$). **e** Top, mean P-wave frequency in REM sleep during saline vs CNO trials in hM4D(Gi) ($n = 8$) and mCherry mice ($n = 7$). Mixed ANOVA ($P_{drug} = 0.020$; $P_{drugXvirus} = 0.044$) with Holm post-hoc (hM4D(Gi), saline vs CNO, $P = 0.046$; CNO, hM4D(Gi) vs mCherry, $P = 0.010$). Bottom, bootstrapped sampling distributions of the mean difference in P-wave frequency between CNO and saline trials in hM4D(Gi) and mCherry mice. Equal-tail bootstrap ($P = 0.002$). For violin plots, center dots represent the distribution mean, and black lines represent the 95% CI. EEG spectrograms (**b, d**) were normalized at each frequency by its mean power across the recording. Error bars and shadings indicate ±SEM. Lines, individual mice. † denotes main effect of drug (saline vs CNO); ‡ denotes interaction between drug and virus. *$P < 0.05$; **$P < 0.01$. Statistical details (**c, e**) shown in Supplementary Table 1. Source data are provided as a Source Data file.

Optogenetic stimulation of dmM CRH neurons initiated and maintained REM sleep, while inhibition reduced the REM bout duration (Fig. 3). Brief activation of this population reliably triggered P-waves as well as theta bursts (Fig. 4). Finally, chemogenetic activation of dmM CRH neurons promoted REM sleep and increased the frequency of P-waves during REM sleep and IS, while DREADD inhibition reduced both REM sleep (Fig. 5) and the P-wave frequency (Fig. 6).

Previous work has shown that inhibitory neurons in the dmM promote REM sleep through suppression of REM-off populations in the locus coeruleus (LC), the ventrolateral periaqueductal gray (vlPAG), and the dorsal and median raphe nuclei[19,20,34,35]. However, given the glutamatergic nature of the CRH neurons (Supplementary Fig. 1d), it is likely that this population controls REM sleep through a different mechanism. Anatomical studies have revealed dense projections from the nucleus prepositus to the dorsal pons[23,36–38], which is critical for generating many of the distinctive neurophysiological features of REM sleep[1,2,31,39,40]. We found axonal projections of the dmM CRH population in the subcoeruleus nucleus (Supplementary Fig. 1f, i), where we recorded P-waves and which contributes to the expression of REM sleep as well as muscle atonia during this state[24,41–43]. Lesioning of the subcoeruleus nucleus in rodents, and the analogous peribrachial nucleus in cats, can completely abolish P-waves during REM sleep[44–46], raising the possibility that the dmM CRH neurons promote P-waves by directly exciting the P-wave generator. We also found dmM CRH projections to the dorsal tegmental nucleus (DTg) and the NI (Supplementary Fig. 1g–i), which are thought to be important relays in the septohippocampal pathway that controls the theta rhythm during

REM sleep[47–49]. Interestingly, the NI contains a large number of neurons expressing the CRH receptor-1[50], whose activity is phase-locked to hippocampal theta waves and strongly increased by CRH[51,52]. Given previous research establishing the NI as a key node in the brainstem networks generating and modulating theta oscillations[26,27,53–55], we speculate that CRH-dependent interactions with the NI may underlie the induction of theta bursts by activating dmM CRH neurons, and may also be involved in the initiation of REM sleep itself. In addition, local excitation of REM sleep-promoting GABAergic neurons in the dmM may also induce REM sleep[20]. Neuronal activity in the dmM may be suppressed by projections from areas with REM-off neurons such as the dorsal raphe[37], while inputs from REM sleep-active neurons in the pons such as the pontine reticular formation[36,37] or the PPT[38] may activate the dmM CRH neurons. We propose that as the animal transitions to REM sleep, GABAergic neurons in the medulla proceed to inhibit REM-off populations in the LC, vlPAG, and raphe nuclei, while the CRH neurons provide excitatory drive to the pontine generators of P-waves and theta oscillations, thereby promoting a form of REM sleep characterized by an enhanced P-wave density. Feedback projections from the PPT, which contains P/PGO wave-activated neurons[56,57], may further enhance the density of P-waves during REM sleep. It will be important for future studies to disentangle the precise functional roles of the different postsynaptic targets excited by the dmM CRH neurons, and to determine whether signaling at each site is mediated by glutamate or by the CRH peptide.

Tonic REM sleep is interleaved with phasic bursts of brain activity that manifest in a variety of forms, including P-waves and EEG theta

bursts. In our optogenetic experiments, laser-triggered P-waves were associated with theta bursts (Fig. 4d; Supplementary Fig. 5c), consistent with previous findings that P-waves and transient theta accelerations coincide with each other[7–9,25]. Projections of the dmM CRH neurons to areas that regulate both P-waves (subcoeruleus nucleus) and theta rhythms (NI and DTg) may mechanistically underlie the temporal association between these two phasic events. Previous tracing studies have revealed that the subcoeruleus nucleus is reciprocally connected with the pontine reticular formation, which in turn, along with the NI, sends direct efferent projections to theta generators in the septohippocampal system[24,49,58–60]. These anatomical interconnections may further synchronize the timing of P-waves and theta bursts. We also found that the success rate of optogenetically induced P-waves during REM sleep was greater during epochs with lower theta power and frequency (Supplementary Fig. 5d–f) and during the rising phase of the theta oscillation (Fig. 4i), the latter being consistent with the previously described phase preferences of spontaneous P-waves[25]. Our finding that laser pulses failing to trigger a P-wave were also unable to induce theta bursts (Fig. 4e) suggests that dmM CRH neurons modulate theta oscillations through P-waves. Thus, while activation of dmM CRH neurons can strongly modulate the ongoing frequency and amplitude of the theta oscillations, this effect is influenced by the ongoing phase of the theta rhythm, indicating the presence of a control mechanism coordinating P-wave induction with the theta phase. Given that cells in the NI and DTg have been shown to fire in synchrony with theta oscillations[47,51], we speculate that dmM CRH inputs to these areas are most effective when they coincide with the phase of the theta rhythm where the excitability of NI/DTg neurons is highest. Alternatively, circuits controlling theta rhythms may send feedback projections to the dmM, thereby directly influencing the excitability of CRH neurons and consequently the success rate of optogenetic stimulation.

Both P-waves and theta bursts have been found to play important roles in memory consolidation during REM sleep[11–14,61]. Theta bursts coincide with brief epochs of enhanced synchrony within the hippocampus[10] and between the hippocampus and other areas implicated in memory processing, such as the amygdala and the retrosplenial cortex[11,15,62,63]. Our finding that dmM CRH neurons promote both REM sleep and P-waves, which in turn are coupled with theta bursts, thus indicates a potential role for this medullary population in REM sleep-dependent cognitive functions; factors affecting the excitability of these medulla neurons may consequently influence the coordination of neural activity in the forebrain during REM sleep. The capability to manipulate the frequency of P-waves provides a powerful tool to study their causal role in memory processing during REM sleep.

## Methods

### Animals
All experiments were approved by the Institutional Animal Care and Use Committee (IACUC) at the University of Pennsylvania and conducted in accordance with the guidelines set by the National Institutes of Health Office of Laboratory Animal Welfare Policy. Experiments were performed in male or female CRH-IRES-Cre (Jackson Laboratory stock no. 012704) and male C57BL/6 mice (Jackson Laboratory stock no. 000664). Animals were housed on a 12-h light-dark cycle (lights on at 7 am and off at 7 pm) and were aged 6–12 weeks at the time of surgery. The colony room was maintained at an ambient temperature of 20–23 °C and humidity of 40–60%. Male and female CRH-IRES-Cre mice were distributed equally across groups in each experiment. All mice were group-housed with free access to food and water.

### Surgical procedures
All surgeries were performed following the IACUC guidelines for rodent survival surgery. Prior to surgery, mice were given meloxicam

subcutaneously (5 mg/kg)[20]. Mice were anesthetized using isoflurane (1–4%) and placed in a stereotaxic frame with a heating pad to maintain the body temperature during the procedure. After asepsis, the skin was incised to expose the skull. For unilateral virus injection or implantation, a small burr hole was drilled above the dmM (anteroposterior (AP) −6.6 mm, mediolateral (ML) −0.2 mm), while two holes were drilled (ML ± 0.2 mm) for bilateral injections. After surgery, bupivacaine (2 mg/kg) was administered at the incision site, and we monitored any signs of pain or distress every 10 min until animals were fully recovered from anesthesia and ambulatory. Afterwards, mice were further monitored at least once daily for 3 days and then at least twice weekly for any signs of pain or distress and changes in body weight.

For fiber photometry recordings of dmM CRH neurons, 0.5–1.0 µl of AAV1-Syn-Flex-GCaMP6s-WPRE-SV40 (University of Pennsylvania vector core) was unilaterally injected into the target area of CRH-Cre mice using a Nanoject II (Drummond Scientific) via a glass micropipette (dorsoventral (DV) −3.7 mm). After virus injection, an optic fiber (0.4 mm diameter) was placed on top of the injection site (DV −3.4 mm). For optogenetic activation experiments, 0.5–1.0 µl of AAV1-EF1a-DIO-hChR2-eYFP-WPRE-hGH (University of Pennsylvania vector core) was unilaterally injected into the dmM (DV −3.7 mm). For optogenetic inhibition, we injected 1.0 µl of AAV-Ef1a-DIO-iC++-eYFP (University of North Carolina vector core), and for controls, we injected 0.5–1.0 µl of AAV2-Ef1a-DIO-eYFP (University of Pennsylvania vector core). For all optogenetic experiments, an optic fiber (0.2 mm diameter) was implanted above the dmM (DV −3.4 mm) in a separate surgery 2–3 weeks later. For chemogenetic activation experiments, 250 nl of AAV8-hSyn-DIO-hM3D(Gq)-mCherry (Addgene) was bilaterally injected into the dmM (DV −3.7 mm). For chemogenetic inhibition, we injected 250 nl of AAV8-hSyn-DIO-hM4D(Gi)-mCherry (Addgene), and control mice were injected with AAV8-hSyn-DIO-mCherry (Addgene).

EEG signals were recorded using stainless steel wires attached to two screws on top of the right parietal and left frontal cortex[20]. The reference screw was placed on top of the left cerebellum. For EMG recordings, one or two stainless steel wires were inserted into the neck muscles. To record P-waves, we used bipolar electrodes consisting of two PFA-coated stainless steel wires (127 µm diameter, A-M Systems), glued together with their tips vertically separated by 1.0 mm. An additional burr hole was drilled (AP −5.4 mm, ML 0.9 mm), and the electrodes were implanted with the dorsal tip in the subcoeruleus nucleus (DV −3.5 mm) recording the primary LFP signal, and the ventral tip in the caudal pontine reticular nucleus (DV −4.5 mm) recording the reference signal. All electrodes, screws, connectors, and optic fibers were secured to the skull using dental cement. For fiber photometry and chemogenetic experiments, EEG/EMG/LFP recording electrodes were implanted directly following virus injection, and for optogenetic experiments, they were implanted alongside the optic fiber in a subsequent surgery.

For fiber photometry, opto-, and chemogenetic experiments, we excluded animals where no virus expression could be detected or where expression was also found in the ventral medulla, as well as mice in which the optic fiber tip was located below the virus expression. For P-wave recordings, we excluded mice in which neither electrode tip was located in the subcoeruleus nucleus, as no P-waves could be detected in these animals.

### Histology
Mice were deeply anesthetized and transcardially perfused with 0.1 M phosphate-buffered saline (PBS) followed by 4% paraformaldehyde (PFA) in PBS[20]. After removal, brains were fixed overnight in 4% PFA and then cryoprotected by storing in 30% sucrose by volume in PBS solution for at least one night. After embedding and freezing, brains were sliced into 60 µm sections using a cryostat (Thermo Scientific HM525 NX) and mounted directly onto glass slides.

For immunohistochemistry, brain sections were washed in PBS, permeabilized using PBST (0.3% Triton X-100 in PBS) for 30 min and then incubated in blocking solution (5% normal donkey serum in PBST) for 1 h at room temperature (RT). To stain eYFP-expressing neurons, brain sections were subsequently incubated with a chicken anti-GFP primary antibody (Aves Lab, GFP-1020, 1:1,000) diluted in PBS overnight at 4 °C. The next day, brain sections were incubated for 2 h at RT with a species-specific secondary antibody conjugated with green Alexa fluorophore (Jackson ImmunoResearch Laboratories, Inc., 703-545-155, 1:500; donkey anti-chicken) diluted in PBS. The slides were washed with PBS, counterstained with Hoechst solution (33342, Thermo Scientific), and coverslipped with Fluoromount-G (Southern Biotechnic). Fluorescence images were taken using a fluorescence microscope (Microscope, Leica DM6B; Camera, Leica DFC7000GT; LED, Leica CTR6 LED).

### Fluorescence in situ hybridization (FISH)

For in situ hybridization, the mouse brain tissue was removed and fixed as described above. 48 h after removal, brains were cryosectioned at 30 μm and slides were baked for 1 h at 60 °C, followed by overnight storage at −80 °C. Fluorescence in situ hybridization was performed using the RNAscope® Fluorescent Multiplex Kit (Advanced Cell Diagnosis), and all solutions and probes were acquired and prepared according to the instructions of the manufacturer. Prior to use, probes were warmed to 40 °C and then cooled to RT, while Protease III solution and Amplifiers 1–4 were equilibrated to RT and mixed by inverting the tube. We used target probes for *crh* (316091; detection channel-1), *vglut1* (416631; detection channel-3), *vglut2* (319171; detection channel-3), and *gad2* (439378; detection channel-3). Slides were washed in PBS and dehydrated at RT in a graded ethanol (EtOH) series (50, 75, and 100% in PBS, 5 min for each grade). Slides were then boiled (99–100 °C) for 10 s in distilled water (DW), followed by 5 min in 1X target retrieval solution (ACD cat. no. 322001). Tissue sections were briefly washed twice in PBS and once in 100% EtOH, air dried, and outlined on the slide using the ImmedgeTM hydrophobic barrier pen (ACD). Slides were incubated at 40 °C in Protease III solution for 30 min, then briefly washed three times in fresh DW at RT. Brain sections were then hybridized with the probe mixture for 4 h at 40 °C, and subsequently washed at RT for 2 × 2 min with fresh 1X RNAscope® washing buffer. Probe signals were amplified by incubation at 40 °C in Amp1 (60 min), Amp2 (30 min), Amp3 (60 min), and Amp4B (30 min). Between each amplification step, slides were washed for 2 × 2 min in the washing buffer at RT. Following treatment with Amp4B, tissue sections were washed with PBS and counterstained with Hoechst solution, followed by coverslipping and imaging as described above.

### Virus and mRNA expression analysis

To generate heatmaps of virus expression[20] (Fig. 1d) or *Crh* mRNA distribution (Supplementary Fig. 1a), coronal reference images for the appropriate AP coordinates were adapted from the Allen Reference Atlas – Mouse Brain (atlas.brain-map.org). For each reference image, the corresponding histology section was overlaid and fitted to the reference. Regions containing fluorescence-labeled cell bodies were manually outlined, and the complete set of outlines was analyzed by custom Python programs. For each location on the reference image, the number of brain sections or mice expressing the fluorescence marker was represented by the intensity of the color at that site.

### Polysomnographic recordings

Sleep recordings were performed in the animal's home cage or in a cage to which the mouse had been habituated for at least 3 days, placed within a sound-attenuating box[20]. All recordings were performed during the light phase between 9 am and 6 pm. For opto- and chemogenetic studies, EEG, EMG and LFP signals were recorded using an RHD2132 amplifier (Intan Technologies, sampling rate 1 kHz)

connected to an RHD USB Interface Board (Intan Technologies) with RHD2000 interface software (Intan Technologies, version 1.5.2). For fiber photometry recordings, a TDT RZ5P amplifier (sampling rate 1.5 kHz) was used. Signals were referenced to a common ground screw, inserted on top of the cerebellum. Videos were recorded using a camera (FLIR, Chameleon3 or ELB, Mini USB camera) placed above the mouse cage. During the recordings, EEG, EMG, and LFP electrodes were connected to flexible recording cables using a small connector. Brain states were scored manually by visual inspection of the EEG and EMG signals, EEG spectrograms, and EMG power using a graphical user interface programmed in Python (https://github.com/tortugar/Lab/tree/master/PySleep). EEG and EMG spectrograms were computed with consecutive Fast Fourier Transforms (FFTs) with Hanning window, calculated for sliding, half-overlapping 5 s windows resulting in 2.5 s time resolution of the hypnogram. The EMG power was computed by integrating the EMG spectrogram in the range 10–500 Hz. States with low amplitude, fast EEG activity and increased EMG tone were scored as wake. REM sleep was characterized by dominant theta (θ) oscillations, low delta (δ) power, and minimal EMG tone. States with high-amplitude δ activity and low EMG tone were classified as NREM sleep. IS was identified by a gradual increase in θ power in the parietal EEG channel and a decrease in δ power, coinciding with the presence of sleep spindles in the frontal EEG channel. In the case that IS was followed by REM sleep, the offset of IS was marked by the disappearance of δ activity and the presence of dominant θ power in both EEG channels, together with the absence of sleep spindles. In the case of a wake transition, the offset of IS coincided with a reduction in the EEG amplitude and an increase in EMG tone. For closed-loop optogenetic experiments, REM sleep was automatically classified in real time (see Optogenetic manipulation), and manually verified offline using the graphical user interface. Annotators were blinded to the timing of the laser.

### Fiber photometry

We performed calcium imaging experiments 3 to 6 weeks after surgery. Animals were habituated for at least two days to the recording setup. After habituation, sleep recordings were performed during the light cycle (between 9 am and 6 pm) and lasted for 2 h. For fiber photometry experiments, the implanted optic fiber was connected to a flexible patch cable. In addition, a flexible cable was connected to the EEG/EMG/LFP electrodes via a mini-connector. For calcium imaging, a first LED (Doric Lenses) generated the excitation wavelength of 465 nm and a second LED emitted 405 nm light, which served as a control for photobleaching and motion artifacts, as the emission signal from the 405 nm illumination is independent of the intracellular calcium concentration. The 465 and 405 nm signals were modulated at two different frequencies (210 and 330 Hz). Both lights were passed through dichroic mirrors before entering a patch cable attached to the optic fiber. Fluorescence signals emitted by GCaMP6s were collected by the optic fiber and passed via the patch cable through a dichroic mirror and GFP emission filter (Doric Lenses) before entering a photoreceiver (Newport Co.). Photoreceiver signals were relayed to an TDT RZ5P amplifier and demodulated into two signals using TDT's Synapse software, corresponding to the 465 and 405 nm wavelengths. For further analysis, we used custom-written Python scripts. First, both signals were low-pass filtered at 2 Hz using a 4th order digital Butterworth filter. Next, using linear regression, we fitted the 405 nm to the 465 nm signal. Finally, the linear fit was subtracted from the 465 nm signal (to correct for photo-bleaching or motion artifacts) and the difference was divided by the linear fit yielding the ΔF/F signal. To determine the brain state, EEG and EMG signals were recorded together with LFP and fluorescence signals using the RZP amplifier. We excluded fiber photometry recordings which contained sudden shifts in the baseline (likely due to a loose connection between optic fiber and patch cord), or with no REM sleep episodes.

## Optogenetic manipulation

For optogenetic experiments, mice were tethered to an optic fiber patch cable in addition to the cable used for EEG/EMG/LFP signals, and recorded for 5 h. For open-loop stimulation, we repeatedly delivered 10 Hz pulse trains (10 ms up, 90 ms down) lasting for 120 s generated by a blue 473 nm laser (2 mW, Laserglow). Inter-stimulation intervals were randomly chosen from a uniform distribution ranging from 15 to 25 min. TTL pulses to trigger the laser were generated by a Raspberry Pi, which in turn was controlled by a graphical user interface custom-built using Python[20]. For optogenetic closed-loop stimulation, the program identified whether the animal was in REM sleep or not based on real-time spectral analysis of the EEG and EMG signals. The onset of REM sleep was defined as the point at which the EEG θ/δ ratio exceeded a hard threshold (mean + one std of θ/δ), which was calculated using previous recordings from the same animal. REM sleep lasted until the θ/δ ratio dropped below a soft threshold (mean of θ/δ) or if the EMG amplitude passed an offline calculated threshold (mean + 0.5 std of amplitude). To compare REM sleep episodes with and without laser stimulation within the same recording session, the laser was turned on only for a randomly selected subset of 50% of REM sleep episodes. For ChR2-mediated closed-loop activation, 10 Hz pulses (473 nm, 2 mW) were continuously delivered throughout each REM sleep episode that was selected for closed-loop stimulation. For iC++-mediated closed-loop inhibition, illumination was continuously presented (473 nm, 4 mW). Control eYFP mice for ChR2 and iC++ mice underwent the same open-loop and closed-loop protocols. For experiments analyzing optogenetically induced P-waves, ChR2 mice were repeatedly presented with brief step pulses (10–20 ms, 2 mW) with inter-stimulation intervals ranging from 10 to 60 s. We excluded optogenetic sleep recordings with strong EEG artifacts, or with no REM sleep episodes.

## Chemogenetic manipulation

For chemogenetic experiments, animals were tethered to a flexible cable for EEG/EMG/LFP recordings[20]. On recording days, mice received intraperitoneal (i.p.) injections of saline or CNO (Clozapine N-oxide dihydrochloride, Tocris, Cat. No. 6329). Mice expressing hM3D(Gq) in the dmM received 0.25 mg/kg CNO, while mice expressing hM4D(Gi) received a dose of 5 mg/kg. Recording sessions began immediately after injection, and each animal contributed 2 saline and 2 CNO recordings to the dataset. mCherry control mice received both CNO doses, and underwent 4 saline and 4 CNO recording sessions in total. The order of injection days (saline vs CNO) was randomly determined.

## cFos expression analysis

To quantify the impact of optogenetic excitation on the activity of dmM CRH neurons (Supplementary Fig. 4a–c), we delivered repeated 10 Hz laser stimulations (2 mW, 473 nm) for 10 s each minute in CRH-Cre mice expressing ChR2-eYFP in the dmM. After 2 h, mice were removed from the recording cage and immediately perfused. To validate the chemogenetic excitation method (Supplementary Fig. 6a, b), we administered i.p. injections of 0.25 mg/kg CNO to CRH-Cre mice expressing hM3D(Gq) in the dmM, immediately placed them in their recording cage, and sacrificed them after 2 h. Following the protocol described above, we performed double FISH using probes for *crh* (316091; detection channel-1) and *fos* (316921, detection channel-3) and captured fluorescence images of the dmM. Cell counting was performed in four brain sections from each animal, with two images showing the left hemisphere and two images showing the right hemisphere.

## LFP analysis

To detect P-waves, the two pontine LFPs were band-pass filtered (1–50 Hz), and the reference LFP signal was subtracted from the primary signal. For animals with P-waves or artifacts in the reference channel, the primary LFP was filtered and used for analysis. To automatically detect P-waves, we applied a threshold (mean − 4.5 * std of the signal), and labeled all waveforms crossing it as P-waves. We eliminated outlier waveforms with amplitudes larger than three standard deviations from the mean (>0.5 mV) or half-widths greater than two standard deviations from the mean (>80 ms), as these properties were characteristic of artifacts in the LFP produced by motor noise. For each recording, the LFP was visually inspected to confirm the presence of well-separated P-waves, and to verify the accuracy of the automatic detection in identifying real P-waves. Animals without clear P-waves or with significant artifacts in the primary LFP were excluded.

The timing of P-waves was defined as the timing of the negative peak. A clustered P-wave was defined as a P-wave separated from its nearest neighbor by an inter-P-wave interval of less than 520 ms, and a single P-wave was defined by an inter-P-wave interval of above 780 ms. These thresholds represent the 10th and 15th percentile values, respectively, of all inter-P-wave intervals calculated from a set of 18 recordings (*n* = 9 mice) collected in the absence of experimental manipulations.

In optogenetic experiments, laser-triggered P-waves were defined as P-waves occurring within 100 ms of the onset of laser stimulation, and spontaneous P-waves were defined as P-waves occurring outside this 100 ms interval. Laser pulses associated with at least one laser-triggered P-wave were defined as "successful", and laser pulses with no P-waves within 100 ms of onset were defined as "failed". The latency of an optogenetically induced P-wave was defined as the time elapsed between the onset of the laser pulse and the negative peak of the P-wave. "Shuffled" laser pulses (Fig. 4g, h) were generated by randomly jittering (up to ±5 s) the time point of each laser pulse during REM sleep, with "successful" shuffled pulses defined by the presence of one or more "laser-triggered" P-waves within the subsequent 100 ms window.

## EMG analysis

To examine the skeletal muscle activity surrounding P-waves and laser pulses, we first collected the raw EMG signal over a 6 s time window (−3 to +3 s) centered around each spontaneous P-wave, laser-triggered P-wave, and failed laser pulse during REM sleep. All events of a given type (e.g. spontaneous P-waves) were pooled across mice, and the mean surrounding EMG activity was calculated by averaging the local signals for all events in the dataset (Supplementary Fig. 5k top). Next, we computed the EMG spectrogram with sliding, half-overlapping 100 ms windows for a temporal resolution of 50 ms, and we integrated the frequencies from 10–100 Hz to determine the EMG amplitude within each time bin. For each P-wave and laser pulse during REM sleep, we isolated the EMG amplitude over the surrounding time window (−1 to +1 s), and z-scored within the collected 2 s interval to identify local changes in the muscle activity relative to the event (Supplementary Fig. 5k bottom). To test whether P-waves and laser pulses were temporally associated with significant EMG changes, we calculated for each mouse the mean normalized EMG amplitude during the 100 ms intervals surrounding spontaneous P-waves, laser-triggered P-waves, and failed laser pulses (−50 to +50 ms), and we compared these values with the mean normalized amplitude during the preceding 100 ms baseline intervals (−150 to −50 ms) using paired t-tests.

## Spectrotemporal correlation analysis

To identify features of the EEG spectrogram associated with dmM CRH calcium activity, we adapted a receptive field model[64] to predict CRH neural activity from the spectrogram during REM sleep. Intuitively, we estimated a spectrotemporal filter using linear regression that predicts for each time point the CRH calcium response. In more detail, we first computed the EEG spectrogram $E(f_i, t_j)$ using 2 s windows with 80% overlap, resulting in a time resolution $dt$ of 400 ms. Each spectrogram frequency was normalized by its mean power across the recording, and

the parameter $E(f_i, t_j)$ specifies the relative amplitude of frequency $f_i$ for time point $t_j$. We then downsampled the ΔF/F response using the same time resolution as for the spectrogram and extracted all time bins with REM sleep for analysis.

To predict the calcium response, the EEG spectrogram was linearly filtered with the kernel $H(f_i, t_l)$. In analogy to receptive fields estimated using similar approaches for sensory neurons, we used the term "spectral field" for $H(f_i, t_l)$. The spectral field can be described as the set of coefficients that optimally relate the neural activity at time $t_j$ to the EEG spectrogram at time $t_{j+l}$, where $l$ represents the lag between the time points of the spectrogram and the neural response. The estimated frequency components of $H(f_i, t_l)$ ranged from $f_1 = 0.5$ Hz to $f_{nf} = 25$ Hz, while the time axis ranged from $-n_T * dt = -4$ s to $n_T * dt = 4$ s. Thus, $H(f_i, t_l)$ comprises $n_f$ frequencies and $2 * n_T + 1$ time bins over a window from −4 to +4 s relative to the neural response. The convolution of the EEG spectrogram with the spectral field can be expressed as

$$\hat{r}(t_j) = r_0 + \sum_{l=-n_T}^{n_T} \sum_{i=1}^{n_f} E(f_i, t_{j+l}) H(f_i, t_l) \quad (1)$$

The scalar parameter $r_O$ denotes a constant offset, and $\hat{r}(t_j)$ denotes the predicted neural response. The optimal spectral field $H(f_i, t_l)$ minimizes the mean-squared error between the predicted and measured neural response at time $t_j$. To account for the large number of estimated parameters, we included a regularization term in the error function[65], which penalizes large kernel components and therefore guards against overfitting of the model. The spectral field for each recording was estimated using 5-fold cross-validation to determine the regularization parameter (λ) that optimized the average model performance on the test sets. Kernels were averaged across all recordings for individual animals; the spectrotemporal correlation in Fig. 1h represents the mean spectral field across all mice.

### EEG spectrogram calculation and power estimation

The power spectral density (PSD) of the parietal EEG was computed using Welch's method with Hanning window for sliding, half-overlapping 2 s intervals. To analyze the impact of optogenetic laser stimulation on the EEG of experimental and control animals (Supplementary Fig. 4m), we computed the PSD with and without the laser in mice expressing ChR2 and eYFP. To test the effect of the laser within each mouse group, we calculated for a given brain state the mean power of the delta (δ, 0.5–4 Hz), theta (θ, 6–10 Hz), sigma (σ, 11–15 Hz), and beta (β, 15.5–20 Hz) frequency bands during laser-on and laser-off epochs. For statistical comparison, we used two-way repeated measures ANOVA with laser and frequency band as within factors, followed by pairwise t-tests with Holm correction. To compare the effects of laser stimulation between ChR2 and EYFP mice, we computed for each frequency band the difference in the EEG power between laser-on and laser-off brain states, and tested whether these values were significantly different across mouse groups using mixed ANOVA with virus and frequency band as between and within factors. We performed the same statistical analysis to quantify the EEG PSDs for chemogenetic excitation (Supplementary Fig. 6h) and inhibition experiments (Supplementary Fig. 7f), with frequency band and drug (saline vs CNO) as the within factors and virus (hM3D(Gq) or hM4D(Gi) vs mCherry) as the between factor.

To analyze time-dependent changes in the EEG, we used two different methods to compute and normalize the EEG spectrogram. Plots showing the averaged EEG spectrogram over several minutes of time, such as during brain state transitions (e.g. Fig. 2d) or optogenetic stimulation intervals (e.g. Fig. 3e), were computed using sliding 5 s windows with 50% overlap, resulting in a temporal resolution of 2.5 s. In order to examine relative EEG changes across minutes-long time intervals including multiple brain states, each frequency component in

the spectrogram was normalized by the mean power of that component over the entire recording. Conversely, plots showing the averaged EEG spectrogram surrounding millisecond-timescale events such as P-waves (e.g. Fig. 2f) or brief laser pulses (e.g. Fig. 4e) were computed using sliding 2 s windows with 95% overlap, resulting in a finer time resolution of 100 ms. For these analyses, the spectrogram was first collected over a window from −3 to +3 s surrounding the time point ($t = 0$ s) of each event. To identify local changes in the EEG relative to the event, each frequency component in the surrounding spectrogram was normalized by its mean power over the collected 6 s interval.

Following normalization, spectrograms were averaged within each mouse, and time dependent power bands (e.g. δ, θ, σ, and γ in Fig. 3e; 8–15 Hz in Fig. 2f) were calculated by computing for each time bin the mean power across the frequencies within the given band. In Fig. 4f, we show the normalized power spectrum of the EEG during spontaneous P-waves, laser-triggered P-waves, failed laser pulses, and randomly selected control time points during REM sleep. After collecting and locally normalizing the spectrograms for each individual event as described above, we isolated the 1 s intervals surrounding spontaneous P-waves, laser-triggered P-waves, and control points (−0.5 s to +0.5 s) and following the onset of failed laser pulses (0 to +1 s), and we plotted the averaged power for each frequency up to 20 Hz. To quantify the difference in spectral profiles, we calculated for each mouse the mean normalized power of frequencies from 8 to 20 Hz, and used one-way repeated measures ANOVA to test whether these values significantly differed among the four plotted events. We applied the same normalization procedure to compare the mean high theta power (8–15 Hz) during 1 s intervals surrounding single vs clustered spontaneous (Supplementary Fig. 3g right) and laser-triggered P-waves (Supplementary Fig. 5c right) using paired t-tests to determine whether the mean normalized power was significantly different between P-wave types.

### Analysis of ΔF/F activity and P-wave frequency during brain state transitions

To analyze the time course of dmM CRH neuron calcium activity during brain state transitions from X to Y (Supplementary Fig. 2c–e), we binned the ΔF/F activity in 2.5 s intervals, to align it with the time resolution of the EEG spectrogram and the hypnogram. We aligned the ΔF/F signals for all X → Y transitions from all mice relative to the time point of the transition ($t = 0$ s), ensuring that the minimum duration of brain state episodes was 40 s for NREM and REM sleep, and 15 s for IS and wake. To determine the time point at which the ΔF/F activity started significantly increasing or decreasing, we used the first 5 s of state X as baseline. For each of the $n$ X → Y transitions recorded from all mice in the experiment, we averaged the ΔF/F amplitude during the baseline bin, resulting in $n$ baseline values. We then subsequently compared this vector with the $n$ time-averaged ΔF/F values computed for consecutive, non-overlapping 5 s bins using consecutive paired t-tests. To account for multiple comparisons, we divided the significance level (α = 0.05) by the number of comparisons (Bonferroni correction). The same statistical procedure was used to quantify changes in the ΔF/F activity during the 20 s interval (Fig. 2i, k) and 2 s interval (Fig. 2j) surrounding P-waves.

To analyze the time course of the P-wave frequency during state transitions (Supplementary Fig. 3b–d), we calculated the rate of P-waves per second in each 2.5 s time bin, and performed statistical analysis across all X → Y transitions as described above. To plot the averaged P-wave frequency during NREM → IS → REM→wake transitions (Fig. 2d; Fig. 5m, o; Supplementary Fig. 3e), we temporally normalized the duration of each brain state episode. The time-normalized P-wave frequency was averaged for all transitions from each mouse and plotted as the mean frequency across mice. This method was also used to plot the ΔF/F activity during NREM → IS → REM→wake transitions (Supplementary Fig. 2b).

## Cross-correlation between ΔF/F and P-wave activity

To quantify the relative timing relationship between dmM CRH neuron calcium activity and P-waves (Supplementary Fig. 2h), we first translated the time points of detected P-waves in each recording into a binary "P-wave train", discretized in 5 ms bins with 1's representing P-waves and 0's representing no P-waves. For each REM sleep period lasting at least 30 s, we collected the corresponding ΔF/F signal and P-wave train and computed the full linear cross-correlation of the two signals. The resulting vector, representing the time shift of ΔF/F activity with respect to the P-wave train, was divided by the number of P-waves in the sequence. We next isolated the correlation values corresponding to relative time delays ranging from −1 to +1 s, with the ΔF/F signal preceding the P-wave signal at negative time points and following the P-wave signal at positive time points. Finally, we averaged the correlation windows across all REM sleep episodes within each mouse, and plotted the mean time lag of the dmM CRH calcium activity relative to P-waves across mice. To quantify during which time points the cross-correlation significantly differed from baseline, we compared the correlation values during a baseline interval (−1 to −0.8 s) to the values during each subsequent 0.2 s interval using consecutive paired t-tests with Bonferroni correction for multiple comparisons, as described in the previous section. As a control, we randomly jittered (up to ±10 s) the time points of true P-waves, and repeated the same analysis for the time-shuffled P-wave trains.

## Analysis of P-waves and laser success rate relative to theta oscillations

To analyze the proportion of spontaneous and laser-triggered P-waves occurring at each phase of the theta oscillation (Fig. 4i), we applied a band-pass filter (6–12 Hz) to the parietal EEG during each REM sleep episode lasting at least 30 s. For each P-wave during REM sleep, the instantaneous phase (−π to π radians) was determined by applying the Hilbert transform to the filtered EEG signal and computing the phase angle. We calculated the percentage of spontaneous and laser-triggered P-waves in each phase bin, and used the Rayleigh test for non-uniformity to determine whether these distributions exhibited significant phase preferences.

To determine whether there were significant differences in the EEG spectrogram preceding laser pulses that succeeded vs failed in eliciting a P-wave during REM sleep (Supplementary Fig. 5d), we computed the spectrogram using consecutive 2 s windows with 80% overlap, resulting in a time resolution of 100 ms. We normalized each spectrogram frequency by its mean power across each recording and collected the normalized spectrogram over the 3 s interval immediately prior to each laser pulse. Supplementary Fig. 5e shows the mean normalized power in the theta frequency band (6–12 Hz) preceding each successful and failed laser pulse, and Supplementary Fig. 5f shows the mean theta frequency. We determined the mean theta frequency for each laser trial by calculating the center of mass (spectral centroid) for the θ range using $\theta_{center} = \sum_{i=6}^{12} f_i \cdot p_i / \sum_{i=6}^{12} p_i$, where $p_i$ denotes the value of the PSD for frequency $f_i$. We used a point-biserial correlation to determine whether the success of laser pulses was significantly impacted by the mean theta power or the frequency preceding the laser pulse.

## Bootstrapping analyses

In Fig. 3h, i, we tested using bootstrapping whether optogenetic activation or inhibition of dmM CRH neurons significantly alters the duration of laser-on REM sleep periods, compared with laser-off periods without stimulation. We assume that the original dataset contains $n$ mice, and that each mouse $i$ has a total of $m_i$ REM sleep episodes, with $m_{i,on}$ and $m_{i,off}$ denoting the number of laser-on episodes and laser-off episodes, respectively. First, we randomly chose $n$ mice with replacement. For each of these mice, we randomly selected (with replacement) $m_{i,on}$ laser-on episodes and $m_{i,off}$ laser-off episodes, and calculated the difference between the mean durations of the resampled laser-on and laser-off episodes. Finally, we computed the mean difference in REM sleep duration in the selected sample by averaging across all $n$ mice. Repeating this procedure 10,000 times for experimental and control mice, we generated for each group a sampling distribution of the mean difference in duration between laser-on and laser-off episodes, which was used to determine the 95% confidence interval (CI). To estimate the difference that would be expected if the laser had no effect, we randomized the designations of REM sleep episodes before bootstrapping; for each mouse, we randomly selected $m_{i,on}$ REM episodes to label as laser-on, and labeled the remaining $m_{i,off}$ REM sleep episodes as laser-off. Bootstrapping this shuffled dataset yielded a sampling distribution of the mean difference in duration between the randomly assigned laser-on and laser-off REM sleep episodes. For statistical comparison of two distributions, we calculated for each bootstrap iteration the difference in the laser-induced duration change between groups, and used the resulting distribution of differences to compute the equal-tail bootstrap P-value. We used the same bootstrapping procedure to estimate the mean difference in the P-wave frequency for REM periods during saline and CNO trials in the chemogenetic excitation (Fig. 6c bottom) and inhibition (Fig. 6e bottom) experiments; for each of the 10,000 iterations, we randomly selected $n$ mice and calculated the mean difference in the P-wave frequency between saline and CNO trials across the $n$ mice. To test whether the resulting sampling distributions (shown as violin plots with 95% CIs) were significantly different, we again computed the equal-tail P-value.

## Statistical tests

Statistical analyses were performed using the Python modules scipy.stats (https://scipy.org/), pingouin (https://pingouin-stats.org/), and statsmodels.stats (https://statsmodels.org/). We did not predetermine sample sizes, but cohorts were similarly sized as in other relevant sleep studies[21,66]. All statistical tests were two-sided. In Fig. 4i, the Rayleigh test for non-uniformity was performed. In Supplementary Fig. 5e, f, point-biserial correlation was performed. All other data were compared using t-tests or using repeated measures or mixed ANOVA followed by multiple comparisons tests (pairwise t-tests with Bonferroni or Holm correction). Box plots and violin plots were used to illustrate the distribution of data points. For box plots, the upper and lower edges of the box correspond to the interquartile range (25th to 75th percentile) of the dataset and the horizontal line in the box depicts the median, while the whiskers indicate the remaining distribution. For violin plots, the center dot represents the distribution mean and the black line shows the 95% CI. Results are presented as mean ± SEM unless otherwise stated. For all tests, a (corrected) P-value < 0.05 was considered significant. The statistical results for each figure are provided in the figure legends and Supplementary Table 1.

## Reporting summary

Further information on research design is available in the Nature Portfolio Reporting Summary linked to this article.

# Data availability

All processed data necessary to interpret, verify, and extend the results of this study are available in the Source Data file. Raw data files are available from the corresponding author within 2-4 weeks of request. Initial examination of gene expression patterns was performed using the Allen Mouse Brain Atlas (available from mouse.brain-map.org). Coronal brain schemes were adapted from the Allen Reference Atlas - Mouse Brain (available from atlas.brain-map.org). Source data are provided with this paper.

## Code availability

All original code used for data analysis is available in the GitHub repository[67] at https://doi.org/10.5281/zenodo.7921731.

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

## Acknowledgements

This work was supported by the National Institutes of Health (NIH), National Heart, Lung, and Blood Institute (NHLBI) (R01HL149133 to F.W. and F31HL154855 to A.L.S.) and National Institute of Neurological Disorders and Stroke (NINDS) (R01NS1100865 to S.J.), by a NARSAD Young Investigator grant by the Brain & Behavior Research Foundation (#27799 to F.W.), and by a grant from the Margaret Q. Landenberger Foundation (to F.W.). We thank S. Forastieri and J. Stucynski for help with histology and sleep annotations, and members of the Weber and Chung labs for helpful feedback and comments on the manuscript.

## Author contributions

A.L.S., S.C., and F.W. conceived and designed the study. A.L.S. performed all optogenetic, chemogenetic, and fiber photometry experiments and analyzed all data. J.B. built the setup for optogenetic sleep recordings including the software to run experiments and the setup for fiber photometry experiments. A.L.S. and F.W. wrote the manuscript.

## Competing interests

The authors declare no competing interests.
