## [Peer Review File · Nature Communications]

REVIEWER COMMENTS

Reviewer #1 (Remarks to the Author):

The manuscript by Schott and colleagues reports on the identification of a group of excitatory CRH-expressing neurons in the dorsomedial medulla (dmM-CRH neurons) that when excited favors the occurrence of P-waves and REM sleep in the mouse. In a set of convincing and straightforward experiments using state of the art methods, the authors demonstrate the relationship between activity of this population of neurons and the frequency of P-waves and REM sleep episodes by recording their spontaneous activity and by opto- and chemo-genetically activating and inhibiting these neurons acutely for sustained periods of time. Moreover, they show that dmM CRH neurons project to other brainstem nuclei known to be involved in P-wave and theta production and in REM sleep regulation.

I have no major concerns with the observations per se but have some problems with putting the findings in context to appreciate their importance. First, what exactly is meant by a 'hub' and 'controlling' in the title with respect to REM sleep and P-waves? dmM-CRH neurons seem one of several neuronal groups targeting midbrain P-wave/REM sleep generators that when activated alter REM sleep and P-waves. Even within the dmM the authors mention several other populations similarly affecting these REM sleep features. The discussion acknowledges that the circuitry underlying P-wave/theta/REM sleep regulation is complex and it is unclear from the data what specific integrative role dmM-CRH neurons play to qualify as a hub (my interpretation of a 'controlling hub'). Would deleting these neurons eliminate P-waves and REM sleep? Moreover, the rationale to focus on this population is not mentioned.

Apart from this overall comment putting the importance of dmM-CRH neurons into context, I have a few specific comments:

1- The authors conclude that their findings "demonstrate a dual role of CRH-expressing medullary neurons in regulating REM sleep and P-waves" but the two are tightly coupled. Which of the current results show this dual role? Could activity of dmM-CRH neurons trigger REM sleep and increased P-waves is simply a consequence of the state change? Or the other way around; even the brief laser activations triggered both P-waves and theta bursts; i.e., the hallmark of REM sleep.

2- The chemogenetically induced sustained activation of the dmM-CRH neurons seemed to result in a more phasic type of REM sleep as P-waves per unit of time in REM sleep increased. Could this interpretation be strengthened by e.g. quantifying the number of EMG twitches?

3- The laser-induced changes in REM sleep appear to trigger a homeostatic process such that the overall amount of REM sleep during the recording did not change. This observation was mentioned to support the claim that the brain state promoted by dmM CRH neurons is genuine REM sleep. By that same logic, is chemogenetically promoted REM sleep not real REM sleep (as no signs of such homeostasis occurred)?

4- Calcium activity profiles during REM sleep seem to indicate that dmM-CRH neurons not only contribute to REM sleep initiation but also to its maintenance. The chemogenetic experiment does not confirm this as REM sleep bout duration was not affected.

5- EEG theta frequencies above 10Hz were observed which is really much faster than those normally observed during REM sleep (the 8 - 15 Hz range was mentioned). I could not find

enough detail on the analyses of the EEG spectra but it was mentioned that "each frequency was normalized by its mean power across the recording". If this means that the power in each frequency bin was normalized to the mean power for that specific frequency bin over the entire recording, then the faster frequencies might be an artefact of this normalization. Do you see the same high theta frequencies also when looking at absolute EEG power (or when using a single reference value to normalize all frequency bins)?

6- Just out of curiosity: do p-waves occur during theta activity in wakefulness or is this a REM sleep specific association?

Reviewer #2 (Remarks to the Author):

A lot of experimental data already support a role of the medio-dorsal medulla (dmM) in the expression of REM sleep. Their precise analysis demonstrates that in fact this functional role is essentially due to a contingent of dmM GABA/glycine inhibitory neurons within the dorsal paragigantocellular nucleus (dPGi), via their sustained overall inhibition of waking-active monoaminergic neurons (including Locus Coeruleus where dPGi neurons send direct projections). The present experimental work by Schott and collaborators consists not in producing an additional work on dPGi neurons but in studying a different contingent of dmM neurons (without efferent inputs to LC as shown in supplementary Fig.1f), excitatory in nature as expressing glutamate/CRH markers and located in praepositus hypoglossi (PH) and medial vestibular nuclei (mVN), both structures lying dorsally to dPGi and forming the floor of the 4th ventricle. The great originality of this work is that such medullary nuclei have never been considered, in my knowledge, as crucial for REM sleep expression but basically for the control of eye movements and visuo-motor coordination in connection with inferior olivary complex and cerebellum, two functions a-priori independent of REM sleep, even if this sleep state is by definition characterized by the presence of rapid eye movements ! At this level, one could ask what was the rationale (not explained in Introduction) to develop such an interest for this particular and unexpected population of CRH neurons within PH / mVN and whether there was experimental data before starting experiments supporting their potential and significant role in REM sleep regulation or generation such as anatomical or functional features, in-vivo unit firing or intrinsic electrophysiological properties ? Besides, it remains that a beautiful aspect of this study is the use of complementary cutting-edge fiber photometry (calcium imaging), bidirectional optogenetic or chemogenetic technologies in CRH-cre mice to provide original and converging evidence for the critical importance of the dmM CRH neurons for dually regulating REM sleep and one of its phasic hallmarks, pontine-P waves. The experimental work in principle appears to be strong and convincing, albeit with some minor technical concerns (see below). This is a nicely written manuscript, which has a very clear logical inductive progression from the introduction into the results, and for much of the discussion, which clearly misses, which is quite frustrating, a serious tentative to integrate these newly discovered REM-promoting neurons to the complex neuronal networks decrypted over the last decades and subserving this physiological function. In this context, it would be of particular interest to determine which of their interconnections (inputs and outputs) with other REM-promoting cell groups could precisely promote the increase of their activity at the onset of REM sleep and in close relationship with P waves. Finally, it is worth mentioning that iconography and Figures are in general of superior quality, nicely constructed, pertinent, complete, easy to read and illustrative, including supplementary ones. To my point of view, there is no major concerns regarding this work and manuscript but a pretty long list of minor comments and technical points that should be addressed:

1)Methods

- The number of transgenic CRH-cre mice used for each set of experiments must more clearly specify in Results section and Figures.
- AAVs injection volumes used for fiber photometry and optogenetic (1ul) appear very large when compared to those used in similar studies in the field, including the present one with only 250 nl for chemogenetics. What is the rationale for such differential volumes for targeting the same brainstem nuclei? Is there a risk of AAVs leakage in LCR when infusing such large volumes in areas so close to the 4th ventricle, with a potential diffusion to other CRH cell groups ?
- For optogenetic experiments, AAVs injection and optic fiber implantation were done unilaterally in dmM. Does it mean that data reported result from the light-induced stimulation or inhibition of CRH neurons in only one hemisphere? If yes, why not trying bilateral stimulations to obtain more drastic effects? Did you eventually notice some physiological or behavioral side-effects due to the placement of the optic fiber tip (or miniscope) into the 4th ventricle (as shown in Fig. 3a), likely perturbing LCR flows ?
- What is the extent of CRH neurons recruited during either optogenetic and chemogenetic activation? Did you check the efficiency of these two kinds of stimulation on the activation of CRH neurons, for example using cFos? We finally do not have any notion of the proportion of dmM CRH neurons targeted in each experiment, which is an important parameter that should be reported in the manuscript for interpreting physiological data achieved.
- A very low dose (0.25 mg/kg) and a very high one (5mg/kg) of CNO were used for the chemogenetic excitation and inhibition, respectively. Why using two different doses, whereas 1mg/kg is usually used in similar studies? On the same topic, no effects on both sleep-waking cycle and power spectral densities across vigilance states were found after CNO administration in control mice. This is quite surprising since recent publications have shown that CNO is not an inert compound and may have by itself non-specific dose-dependent pharmacological effects on vigilance states or EEG rhythmic activities. When analyzing in detail data illustrated in supplementary Fig. 6d-f after CNO treatments in control mice, a tendency (although not significant) is yet present for a dose-dependent increase of REM sleep percentage, duration and bout frequency. Moreover, the baseline values of the same parameters look like somewhat different after saline treatment in control vs. experimental mice (supplementary Fig. 6d-f vs Fig. 5d-g vs Fig. 5 l-n). For example, the mean duration of REM sleep episodes is around 60 sec in experimental mice while they lasted more than 80 seconds in control mice, a difference that could have finally some importance at statistic levels and regarding data significance. How can these differences between samples of mice be explained? Thus two obvious questions came in my mind: again, how many mice were included in the different samples? Did you take into consideration data obtained in control mice for your statistical comparison of saline vs. CNO effects in experimental mice or did you consider that CNO has no potential side-effect? In other words, what were your control conditions for your analyses? This crucial aspect must be more clearly explained in the manuscript.

2)Results

- In my mind and in line with anatomical data from Allen mouse brain atlas, vglut1 is not that much expressed within brainstem nuclei compared to forebrain and cortex mantles, in contrast to vglut2 which is a specific marker of brainstem glutamate neurons. Maybe it could be more relevant to illustrate vglut2 ISH within dmM in the Fig. 1a than in supplementary Fig.1a.
- Page 6 (end of the first paragraph: "Our fiber photometry experiments in both REM sleep and regulation its phasic components". What are included exactly in these "phasic components", just P-waves (correlated to phasic theta activation) as nicely demonstrated in the present study or all other phasic events that characterized REM sleep, as distal

muscle twitches, vibrissa movements, penile erection, and of course REMs themselves (as PH / mVN are involved in the control of horizontal eye movements)? In that context, did you observe an increased frequency of REMs (or other phasic events) during REM sleep in response to both optogenetic (short or long) and chemogenetic activation of CRH-neurons? If yes, did they occur in a synchronized way? In all cases, it would be of particular interest to be more precise on that aspect in Results and Discussion sections.

- Pages 7-8: "Thus, laser-induced changes in REM sleep appear to trigger endogenous homeostatic processes regulating the overall amount of REM sleep". That is really an interesting hypothesis regarding the potential role of these CRH neurons. However, it seems too early to conclude in that way since homeostatic mechanisms of REM sleep were not at all experimentally challenged in this study. Based on short lasting optogenetic stimulations, the present data only show elegantly that CRH neurons seem to play a role firstly in the ultradian rhythm of REM sleep. It is generally believed that processes of REM sleep homeostasis take place within the hypothalamus, although a definitive demonstration is still lacking. In that context, did you observe direct efferent projections emanating specifically from CRH neurons and reaching the lateral hypothalamic area where numerous neurons are activated during REM sleep rebound in response to its specific deprivation? Is there any possibility that CRH themselves may be a part of these homeostatic processes and if yes, how to demonstrate that using your experimental paradigms?

- Page 9: "The efficacy of dmM CRH neurons being above 90% during NREM sleep and IS, 66% during REM sleep....". I'm not sure to understand what exactly mean this sentence and percentages, confusing and somewhat contradictory to the general message. Does it mean it is easier to generate P-waves during natural SWS (during which only occasional spontaneous P-waves occur) than during REM sleep (during which numerous single and clusters of P-waves occur) by short optogenetic excitation of CRH neurons? How can we explain that seemingly unexpected results?

3) Figures

- Figure 1f: What mean negative activity values of calcium imaging (below zero levels) during Waking and NREM sleep?

- Figure 3h-k: As previously underlined for the chemogenetic data (see above and supplementary Fig.6d-f), how to explain such important differences in REM sleep duration in control conditions between Chr2 vs. iC++ experiments (with a 2x difference)?

Reviewer #3 (Remarks to the Author):

The authors investigated the brainstem cellular circuits that contribute to the generation of PGO waves. PGO or P-waves in mice are thought to coordinate brain activity and regulate functional sleep processes including memory consolidation. The author's investigation aims at providing additional evidence on the nature of cells in the dorsomedial medulla (dmM) contributing to P-waves in mice. The authors identified a sub-population of excitatory cells – i.e., corticotropin-releasing-hormone, (CRH) - neurons in the dmM that using calcium imaging (fibre photometry) show that these cells increase their activity during REM in freely-moving mice. Opto-/pharmaco-genetics approaches aimed to show the CRH causal modulation of expression of REM sleep and of P-waves.

This study provides some evidence of a novel cell population modulating REM sleep and important features such as the P-waves. This is an important work for the field of (experimental) sleep and the general understanding of the modulation of vigilance states. However, there are major concerns that need to be addressed before publication. Although it is clear that there is an increase in the activity of CRH cells during REM sleep, the evidence of the modulation of P-waves is weak and the causal evidence to determine

whether or not these cells are indeed causal to the generation of P-waves is inconclusive. In addition, the clarity of the statements and interpretation of the results should be improved, in particular regarding the REM modulation and P-wave generation.

Major changes:

1- Authors stated that the activity of CRH neurons is modulated by REM sleep and P-waves during REM sleep without supporting experimental evidence. The increase of activity of these cells occurs after the peak of the P-wave and therefore the causal link made by the authors is not supported by the data.

Figure 1 reports a very long distance between the optical fibre placed within the ventricle (used for photometry) and the targeted cells. How can the authors be sure that they actually do record the calcium transients of those cells? For instance, transients could result from movement artefacts or changes in CSF volumes across states. This should be clarified. In addition, Figure 1. Please include a higher magnification of the cells expressing GCaMP (d, middle panel). Current images do not really show a healthy cell.

Figure 2, what is the rationale for recording P-Waves in the LC ? – this should be clarified. Moreover, Fig 2i suggests that activity lags from the peak of the P-wave in single events and it is more correlated with the second, smaller peak in the cluster events than the first one. This may be due to the experimental design (electrode and imaging in different locations) or the alignment of the electrophysiological signals with calcium signals. A cross-correlation analysis will strengthen should be provided to support the author's claim. Figure 4h shows that the latency to the induced P-waves happened a few milliseconds. This does not support the author's claims that the activity of CRH controls P-waves at all.

2-Please clarify 1) the changes in the temporal resolution of this analysis and 2) how many of the total amount of laser pulses induced clear P-waves as well as the proportion of detected PW/triggered PW. It is not clear if this is the case in 4h (sham laser?). Authors need to provide a clear quantification of the total PW trigger in YFP and Ch2 animals as well as YFP- IC++.

3- Validation of technique is missing. The authors must provide in vivo validation of Optogenetic and DREADDS effect on the activity of CRH cells.

4-Author statement: "dmM CRH activity in both inducing REM sleep and regulating its phasic component" in page 6. Evidence supporting this claim is weak:

The optogenetic manipulations and the DREADDS experiments. The values of light off (control; fig. 3 h) between for control conditions of ChR2, YFP, IC++ experiments are quite different. This high variability negatively impacts the significance of the results.

Furthermore, the statistical analysis is very misleading. Statistical analysis should test for the difference between control (YFP) and experimental conditions (ChR and IC++) rather than the OFF-laser conditions. This is a major flaw and a lack of scientific rigour.

The power spectrum across different vigilance states (Suppl. Figure 4 and Suppl. Figure 6a-b between Gq and Gi) suggests that the random stimulation may be affecting not only REM but also other vigilant states. Quantification per frequency band must be provided with proper statistical analysis (only qualitative analysis is shown in Suppl. Fig. 4d and e). In addition, homogenise y-axis to the same scale. Lastly, Suppl. Figure 4 should reflect the same quantification as Suppl. Fig. 6 c-g.

Importantly, the lack of modulation of P-waves further calls for the need for state-specific optogenetic manipulation of the CRH activity specifically during REM sleep.

5-Recent work investigated the effect of CNO on sleep and demonstrated modulation of sleep in animals with control vectors (Traut et al. 2022). Although the authors claimed to show the mCherry + CNO, Suppl Figure 6d-h refers to mCherry + NaCl. Thus, authors must provide the appropriate control (e.g. AVV-mCherry + CNO; both at concentrations tested) and use proper statistical analysis, i.e, between experimental groups: NaCl, mCherry+ CNO and mCherry- h3MDGs.

6- P-waves have been classically associated with eye movement, as mentioned by the authors. Did they record the EOG (electrooculogram) by any chance? this would definitely strengthen the findings and the conclusions of the study.

7-The discussion further raised additional side questions. The authors discussed that activation of CRH terminals in the NI could provide evidence to support a role in theta modulation. This is an interesting claim; however, it is not supported by any experimental data. If this is the author's hypothesis, they should provide supporting data. Furthermore, an open question remains as to whether CRH terminals modulate Theta and P-wave through distinct or the same circuit. Similarly, manipulating the terminals in the SubC may help elucidate the role of CRH neurons in -P-wave versus REM sleep. Such a dataset will further strengthen the study.

Minor changes:

In suppl. Figure 1. CRH projections are difficult to appreciate in panels E and G. Please provide a sagittal or coronal section of the axonal distributions across the brain.

Suppl. Figure 2 legend clearly states which neurons are being targeted. In panels b, c and d it is not clear if the average values represent all the transitions in one session, per animal of different recording sessions or between animals. Please specify.

Suppl. Figure 4 g: the colour scale is missing.

Suppl. Figures 6 a and b: y-axis should be homogenized.

Reviewer #1 (reviewer's comments in italics)

The manuscript by Schott and colleagues reports on the identification of a group of excitatory CRH- expressing neurons in the dorsomedial medulla (dmM-CRH neurons) that when excited favors the occurrence of P-waves and REM sleep in the mouse. In a set of convincing and straightforward experiments using state of the art methods, the authors demonstrate the relationship between activity of this population of neurons and the frequency of P-waves and REM sleep episodes by recording their spontaneous activity and by opto- and chemo-genetically activating and inhibiting these neurons acutely for sustained periods of time. Moreover, they show that dmM CRH neurons project to other brainstem nuclei known to be involved in P-wave and theta production and in REM sleep regulation.

Response Text

We thank the reviewer for the feedback on our manuscript and for encouraging us to better conceptualize our findings.

Q1 (Reviewer 1)

I have no major concerns with the observations per se but have some problems with putting the findings in context to appreciate their importance. First, what exactly is meant by a 'hub' and 'controlling' in the title with respect to REM sleep and P-waves? dmM-CRH neurons seem one of several neuronal groups targeting midbrain P-wave/REM sleep generators that when activated alter REM sleep and P-waves. Even within the dmM the authors mention several other populations similarly affecting these REM sleep features. The discussion acknowledges that the circuitry underlying P-wave/theta/REM sleep regulation is complex and it is unclear from the data what specific integrative role dmM-CRH neurons play to qualify as a hub (my interpretation of a 'controlling hub'). Would deleting these neurons eliminate P-waves and REM sleep?

Response Text

We appreciate the reviewer's criticism regarding the conceptualization of our key findings. We have realized that we have not provided sufficient explanation for our interpretation of why the dmM CRH neurons act as a hub for REM sleep and P-wave regulation.

P-waves are a characteristic of REM sleep, as they primarily occur during REM sleep. Therefore, it is likely that brain areas capable of triggering REM sleep itself can also (at least indirectly) trigger P-waves, simply because P-waves occur during REM sleep. However, evidence also suggests that P-waves and REM sleep are regulated by anatomically and functionally distinct mechanisms; indeed, P-waves can occur outside of REM sleep (Datta & Hobson, 1994; Farber et al., 1980), and lesioning the P-wave generator has been shown to cause REM sleep without P-waves (Datta & Hobson, 1995; Mavanji et al., 2004). Moreover, the frequency of P-waves during REM sleep strongly depends on prior wake experience, and the P-wave density during post-learning REM sleep is highly correlated with successful overnight memory consolidation (Datta, 2000; Datta & O'Malley, 2013). Given this context, two key questions arise: first, with important implications for memory processing, which brain areas facilitate REM sleep with an *increased* P-wave frequency? And second, which brain areas can directly trigger P-waves, and not just indirectly via promoting REM sleep?

We found that dmM CRH neurons promote two functional aspects of REM sleep: brief (10–20 ms) optogenetic activation can directly trigger P-waves (**Fig. 4**), and minutes-long activation can induce and maintain REM sleep (**Fig. 3**). Accordingly, sustained chemogenetic activation of dmM CRH neurons leads to more REM sleep with more P-waves, consequently increasing the number of P-waves *per minute* of REM sleep (**Fig. 5**). While we did not delete these neurons,

chemogenetic inactivation led to less REM sleep with a reduced P-wave frequency. In other words, these findings suggest that the activity of dmM CRH neurons functionally contributes to a more phasic form of REM sleep, reflected in an increased P-wave density. Therefore, we believe that the term ‘hub’ is justified to describe the role of these neurons in REM sleep control. To emphasize the specific integrative capabilities of the dmM CRH neurons, we now mention in the Introduction that P-waves and REM sleep expression appear to be governed by largely independent mechanisms (page 2, lines 30-33), and we clarify the interpretation of our results as evidence for a dual role of this population in regulating both of these processes (page 3, lines 57-58).

Q2 (Reviewer 1)

Moreover, the rationale to focus on this population is not mentioned.

Response Text

Most previous work on the regulation of REM sleep by the dmM was primarily focused on GABAergic neurons in the region (Goutnagy et al., 2008; Kaur et al., 2001; Clement et al., 2014; Stucynski et al., 2022). However, the dmM is a heterogeneous area containing diverse cell types (McCrea & Horn, 2006), whose roles in brain state control are largely unexplored. We therefore searched through the *in situ* hybridization data at the Allen Brain Atlas (© 2015 Allen Institute for Brain Science) for candidate genes expressed within the dmM. The *CRH* gene was an ideal marker because its expression pattern is spatially restricted to the dmM, making it technically easier to target this population through virus injections without concerns of co-infecting neurons in neighboring areas. We obtained the CRH-Cre mouse line and tested the effects of dmM CRH neuron stimulation on REM sleep. In parallel, we performed *in situ* hybridization to determine the molecular identity of the CRH population. To our surprise, the CRH population was excitatory, thus constituting a functionally unexplored population of REM sleep-promoting dmM neurons, not overlapping with the previously studied GABAergic populations.

To better explain why we chose to investigate the CRH neurons, we now state in the revised Introduction that most previous studies focused on GABAergic neurons, while other dmM subpopulations are largely uncharacterized (page 3, lines 46-49). In addition, we now mention at the beginning of the Results that we started this project by searching for genes specifically expressed in the dmM (page 3, lines 64-67).

Q3 (Reviewer 1)

Apart from this overall comment putting the importance of dmM-CRH neurons into context, I have a few specific comments:

1 - The authors conclude that their findings “demonstrate a dual role of CRH-expressing medullary neurons in regulating REM sleep and P-waves” but the two are tightly coupled. Which of the current results show this dual role? Could activity of dmM-CRH neurons trigger REM sleep and increased P-waves is simply a consequence of the state change? Or the other way around; even the brief laser activations triggered both P-waves and theta bursts; i.e., the hallmark of REM sleep.

Response Text

This question is closely related to the reviewer’s concern about the term ‘hub’ for describing the role of dmM CRH neurons in REM sleep control, and we again agree that our conclusions must be better put into context. Our assertion that dmM CRH neurons support a dual role in REM

sleep regulation is based on the findings that their activation directly triggers P-waves and promotes REM sleep. If dmM CRH neurons only triggered P-waves (with concurrent EEG theta bursts), then we would expect chemogenetic activation to increase the frequency of P-waves but not the amount of REM sleep itself. Conversely, if dmM CRH neurons only promoted REM sleep, we would expect their activation to result in more REM sleep and an unchanged P-wave frequency. Instead, as a consequence of their dual role in promoting REM sleep and P-waves, we found that activating the dmM CRH neurons caused more REM sleep with a higher rate of P-waves, and inactivating this population resulted in less REM sleep with fewer P-waves.

Q4 (Reviewer 1)

2 - The chemogenetically induced sustained activation of the dmM-CRH neurons seemed to result in a more phasic type of REM sleep as P-waves per unit of time in REM sleep increased. Could this interpretation be strengthened by e.g. quantifying the number of EMG twitches?

Response Text

We appreciate the suggestion to examine the relationship between dmM CRH neurons and other indicators of phasic REM sleep, and we agree that such an analysis would strengthen our results. As recommended by the reviewer, we aimed to quantify phasic muscle twitches in the nuchal EMG during REM sleep.

Although clear twitches can be occasionally seen as increases in the amplitude of the EMG signal, we found that in our recording system, in many cases these peaks were not sufficiently distinguishable from background noise to allow for accurate detection and quantification by an automated algorithm. These technical challenges, due in part to the presence of large heartbeat ticks (R-waves) in the EMG signal during REM sleep, unfortunately prevented a thorough analysis of muscle twitches for our chemogenetic activation studies. Therefore, instead of directly quantifying the number or frequency of muscle twitches, we instead used P-waves and short optogenetic laser pulses as triggers to quantify changes in the surrounding EMG amplitude, allowing us to determine whether brief activation of dmM CRH neurons is sufficient to induce phasic muscle twitches (revised **Supplementary Fig. 5j,k**). We found that during REM sleep, both spontaneous and laser-triggered P-waves coincided with sharp increases in the EMG amplitude, while laser pulses that failed to elicit P-waves were not associated with any significant changes in EMG activity. Altogether, the results from our optogenetic experiments indicate that dmM CRH neurons promote P-waves concomitant with theta bursts and muscle twitches. Furthermore, P-waves appear to be a gateway to the latter two events, as a given laser stimulation can only trigger a theta burst and/or EMG twitch if it also successfully triggers a P-wave. We report these findings in our revised Results (page 11, lines 262-272).

Q5 (Reviewer 1)

3 - The laser-induced changes in REM sleep appear to trigger a homeostatic process such that the overall amount of REM sleep during the recording did not change. This observation was mentioned to support the claim that the brain state promoted by dmM CRH neurons is genuine REM sleep. By that same logic, is chemogenetically promoted REM sleep not real REM sleep (as no signs of such homeostasis occurred)?

Response Text

In our optogenetic closed-loop experiments, manipulation of dmM CRH neuron activity is restricted to a subset (~50%) of REM sleep episodes. Consequently, this protocol leaves long intervals without modulation of neural activity, allowing for quantification of the homeostatic

responses that may result from laser-induced changes in the duration of REM sleep episodes. In contrast, chemogenetic manipulation results in hours-long activation or inhibition of dmM CRH neurons throughout the entire recording, which may compete with simultaneous homeostatic processes. Assuming that the REM sleep homeostat is still intact, the overall sleep pattern following CNO injection is likely influenced by two factors: (1) direct effects induced by changes in the activity of dmM CRH neurons, and (2) potential homeostatic responses resulting from changes in the amount of REM sleep. To disentangle the influence of these two factors would require further experiments that directly challenge the REM homeostat (e.g. REM sleep deprivation) in combination with manipulation of the dmM neurons throughout the deprivation or recovery period. We believe that such experiments, although very interesting, would be beyond the scope of the present revision. However, based on visual inspection of the raw EEG and EMG signals and additional statistical comparison of the EEG power spectra in the revised manuscript (**Supplementary Fig. 6f,i**), we found that REM sleep episodes in CNO recordings are indistinguishable from REM sleep episodes in control recordings, and we can therefore conclude that these states represent genuine REM sleep.

We now understand that the mentioned sentence, without further explanation, may cause misunderstandings. In the revised Results we have therefore removed the sentence, and instead explain in more detail why our findings from closed-loop stimulation of dmM CRH neurons indicate the presence of a homeostatic process (page 9, lines 203-211).

Q6 (Reviewer 1)

4 - Calcium activity profiles during REM sleep seem to indicate that dmM-CRH neurons not only contribute to REM sleep initiation but also to its maintenance. The chemogenetic experiment does not confirm this as REM sleep bout duration was not affected.

Response Text

The REM sleep-maintaining properties of dmM CRH neurons are supported by the closed-loop stimulation experiments (**Fig. 3h,i** in the revised manuscript). Increasing the number of animals for the chemogenetic experiments, we found that chemogenetic inhibition also shortens the duration of REM sleep episodes, consistent with a role of these neurons in maintaining REM sleep (revised **Fig. 5j**). For chemogenetic activation, however, the duration of REM sleep episodes is not changed (revised **Fig. 5f**). A possible reason why hM3D(Gq) activation did not extend the duration of REM sleep bouts could be potential state-dependent effects of chemogenetic activation. While the increase in the CRH neuron activity afforded by hM3D(Gq) during NREM sleep was effective in increasing the number of NREM-to-REM sleep transitions (revised **Fig. 5g**), the activity increase during REM sleep may not be enough (in contrast to optogenetic activation) to increase the duration of REM sleep episodes. Alternatively, as sustained chemogenetic activation of dmM CRH neurons facilitates transitions from NREM to REM sleep, the resulting increase in the REM sleep amount may lower the homeostatic need for REM sleep, thereby counteracting extensions of REM sleep episodes. A previous study showed that entering REM sleep with low REM sleep propensity leads to shorter REM sleep episodes (Park et al., 2021), an effect that may mask the REM sleep-maintaining effect of chemogenetic activation.

Q7 (Reviewer 1)

5 - EEG theta frequencies above 10Hz were observed which is really much faster than those normally observed during REM sleep (the 8 - 15 Hz range was mentioned). I could not find enough detail on the analyses of the EEG spectra but it was mentioned that "each frequency

was normalized by its mean power across the recording". If this means that the power in each frequency bin was normalized to the mean power for that specific frequency bin over the entire recording, then the faster frequencies might be an artefact of this normalization. Do you see the same high theta frequencies also when looking at absolute EEG power (or when using a single reference value to normalize all frequency bins)?

Response Text

The reviewer is correct in their interpretation of the phrase "each frequency was normalized by its mean power across the recording", and we thank them for pointing out this potential source of confusion. For a given EEG frequency, we computed the power in each 2.5 s time bin and divided by the mean power (for that frequency) across the entire recording, in order to show the *relative* changes in frequency power over time. Some type of normalization is necessary because higher frequencies generally have diminished power, and would not be visible in the EEG spectrogram without scaling. We used this approach to show all frequencies of interest when plotting the mouse-averaged EEG spectrogram over a minutes-long time interval, typically for brain state transitions (e.g. **Fig. 2d**) or optogenetic stimulation intervals (e.g. **Fig. 3e**). Importantly, this normalization procedure does *not* result in particularly fast theta frequencies; for example, the above referenced figures show the theta band during REM sleep in the typical 6–9 Hz range. This type of normalization is appropriate for analyzing sleep patterns across time periods of several minutes or more, which include multiple brain states.

In contrast, when analyzing changes in the REM sleep EEG occurring relative to P-waves, we would argue that the normalization should only include the EEG during REM sleep. Further, to identify local changes in the EEG that temporally correlate with P-waves (e.g. **Fig. 2f**; **Fig. 4c-e**), we used only a 6 s window surrounding each P-wave for normalization. More precisely, for each P-wave, we collected the raw EEG spectrogram of the 6 s time window surrounding the trough of the waveform, and calculated the mean power of each frequency over the collected window. We used these values to normalize the power in each frequency bin, so the resulting spectrogram showed the relative changes in frequency power from 3 seconds before the P-wave to 3 seconds after. Thus, the colors of the heatmaps do not show the absolute values, but rather the relative increase or decrease of the EEG power for the corresponding frequencies.

As suggested by the reviewer, we tested whether P-waves are associated with high theta frequencies in the non-normalized EEG. Examining the raw EEG spectrogram in the 6 s window surrounding P-waves during REM sleep, we observed that P-waves (0 s) were correlated with a transient acceleration of the theta band (**Reviewer Figure 1.1a**). To quantify this association, we compared the power spectral densities (PSDs) for the raw EEG during REM sleep with single P-waves, clustered P-waves, and no P-waves (**Reviewer Figure 1.1b**). This analysis confirmed that the absolute EEG power in the high theta range (8–15 Hz) was significantly higher for single P-waves than for REM sleep without P-waves, and increased further for clustered P-waves ($P < 0.001$, one-way ANOVA followed by t-tests with Holm correction). To clarify the different normalization procedures used in our analyses, we have provided additional details in the revised Methods section (pages 30-31, lines 737-750).

REVIEWER FIGURE 1.1. P-waves are correlated with increases in the absolute power of high theta frequencies in the EEG.

a Averaged raw EEG spectrogram surrounding P-waves during REM sleep ($n = 9$ mice). The negative peak of the P-waves is aligned with time point 0 s.

b Left, power spectral density (PSD) of the EEG surrounding single P-waves, clustered P-waves, and no P-waves during REM sleep ($n = 9$ mice). For both single and clustered P-waves, the 2 s time windows surrounding the trough of each waveform were excised from the spectrogram; any remaining time points were analyzed as “tonic REM”, or REM sleep without P-waves. For all three groups, the mean absolute power of each frequency was first computed for each mouse and then plotted as the average across mice. Right, total EEG power in the high theta (8-15 Hz) frequency range. One-way repeated measures ANOVA ($F(2, 16) = 107.91$, $P = 1.76e-6$), followed by pairwise t-tests with Holm correction (no P-waves vs single P-waves, $T(8) = 11.91$, $P = 6.79e-6$; no P-waves vs clustered P-waves, $T(8) = 10.85$, $P = 9.21e-6$; single P-waves vs clustered P-waves, $T(8) = 8.40$, $P = 3.08e-5$). Error bars, \pm SEM; lines, individual mice. *** $P < 0.001$.

Q8 (Reviewer 1)

6 - Just out of curiosity: do p-waves occur during theta activity in wakefulness or is this a REM sleep specific association?

Response Text

Interestingly, we did not find an association between theta power and P-waves during wake. As shown in **Reviewer Figure 1.2**, comparing the absolute power spectral densities of the EEG between wakefulness with P-waves and without P-waves, we found no significant difference in the power of high theta frequencies, suggesting that the association between P-waves and transient increases in theta oscillations is specific to REM sleep.

REVIEWER FIGURE 1.2. P-waves during wakefulness are not associated with EEG theta oscillations.

a Left, power spectral density (PSD) of the EEG during wakefulness, with and without P-waves ($n = 9$ mice). Right, total EEG power in the high theta (8-15 Hz) frequency range (paired t-test, $T(9) = 1.796$, $P = 0.110$). Error bars, \pm SEM; lines, individual mice.

Clément, O. *et al.* The Inhibition of the Dorsal Paragigantocellular Reticular Nucleus Induces Waking and the Activation of All Adrenergic and Noradrenergic Neurons: A Combined Pharmacological and Functional Neuroanatomical Study. *PLoS One* **9**, e96851 (2014).

Datta, S. Avoidance task training potentiates phasic pontine-wave density in the rat: A mechanism for sleep-dependent plasticity. *J. Neurosci.* **20**, 8607–8613 (2000).

Datta, S. & Hobson, J. A. Neuronal activity in the caudolateral peribrachial pons: relationship to PGO waves and rapid eye movements. *J. Neurophysiol.* **71**, 95–109 (1994).

Datta, S. & Hobson, J. A. Suppression of ponto-geniculo-occipital waves by neurotoxic lesions of pontine caudo-lateral peribrachial cells. *Neuroscience* **67**, 703–712 (1995).

- Datta, S. & O'Malley, M. W. Fear extinction memory consolidation requires potentiation of pontine-wave activity during REM sleep. *J. Neurosci.* **33**, 4561–4569 (2013).
- Farber, J., Marks, G. A. & Roffwarg, H. P. Rapid eye movement sleep PGO-type waves are present in the dorsal pons of the albino rat. *Science* **209**, 615–617 (1980).
- Goutagny, R. *et al.* Role of the dorsal paragigantocellular reticular nucleus in paradoxical (rapid eye movement) sleep generation: a combined electrophysiological and anatomical study in the rat. *Neuroscience* **152**, 849–857 (2008).
- Kaur, S., Saxena, R. & Mallick, B. N. GABAergic neurons in prepositus hypoglossi regulate REM sleep by its action on locus coeruleus in freely moving rats. *Synapse* **42**, 141–150 (2001).
- Mavanji, V., Ulloor, J., Saha, S. & Datta, S. Neurotoxic lesions of phasic pontine-wave generator cells impair retention of 2-way active avoidance memory. *Sleep* **27**, 1282–1292 (2004).
- McCrea, R. A. & Horn, A. K. E. Nucleus prepositus. in *Progress in Brain Research* (ed. Büttner-Ennever, J. A.) vol. 151 205–230 (Elsevier, 2006).
- Park, S.-H. *et al.* A probabilistic model for the ultradian timing of REM sleep in mice. *PLoS Comput. Biol.* **17**, e1009316 (2021).
- Sano, K., Iwahara, S., Senba, K., Sano, A. & Yamazaki, S. Eye movements and hippocampal theta activity in rats. *Electroencephalography and Clinical Neurophysiology* **35**, 621–625 (1973).
- Stucynski, J. A., Schott, A. L., Baik, J., Chung, S. & Weber, F. Regulation of REM sleep by inhibitory neurons in the dorsomedial medulla. *Curr. Biol.* **32**, 37–50 (2022).

Reviewer #2 (reviewer's comments in italics)

A lot of experimental data already support a role of the medio-dorsal medulla (dmM) in the expression of REM sleep. Their precise analysis demonstrates that in fact this functional role is essentially due to a contingent of dmM GABA/glycine inhibitory neurons within the dorsal paragigantocellular nucleus (dPGi), via their sustained overall inhibition of waking-active monoaminergic neurons (including Locus Coeruleus where dPGi neurons send direct projections). The present experimental work by Schott and collaborators consists not in producing an additional work on dPGi neurons but in studying a different contingent of dmM neurons (without efferent inputs to LC as shown in supplementary Fig. 1f), excitatory in nature as expressing glutamate/CRH markers and located in praepositus hypoglossi (PH) and medial vestibular nuclei (mVN), both structures lying dorsally to dPGi and forming the floor of the 4th ventricle. The great originality of this work is that such medullary nuclei have never been considered, in my knowledge, as crucial for REM sleep expression but basically for the control of eye movements and visuo-motor coordination in connection with inferior olivary complex and cerebellum, two functions a-priori independent of REM sleep, even if this sleep state is by definition characterized by the presence of rapid eye movements !

Response Text

We thank the reviewer for these supportive comments!

Q9 (Reviewer 2)

At this level, one could ask what was the rational (not explained in Introduction) to develop such an interest for this particular and unexpected population of CRH neurons within PH / mVN and whether there was experimental data before starting experiments supporting their potential and significant role in REM sleep regulation or generation such as anatomical or functional features, in-vivo unit firing or intrinsic electrophysiological properties?

Response Text

Previous studies have implicated a role for the dmM in promoting REM sleep, chiefly through GABAergic projections to REM-off neurons in the pons and midbrain (Goutnagay et al., 2008; Kaur et al., 2001; Verret et al., 2006; Stucynski et al., 2022). In addition to these inhibitory neurons, the dmM contains multiple subgroups of glutamatergic and peptidergic neurons (McCrea & Horn, 2006), most of which have not been studied in the context of sleep-wake regulation. We therefore decided to systematically search through the Allen brain atlas (© 2015 Allen Institute for Brain Science) to identify genes specifically expressed in the dmM. In particular, we noticed that neurons expressing the *CRH* gene formed a small, anatomically isolated population within the borders of the dmM, providing a precise molecular handle for cell-type specific recording and manipulation techniques. Using fluorescence *in situ* hybridization (FISH), we identified the dmM CRH neurons as glutamatergic and therefore functionally distinct from the previously studied GABAergic population in this area, and our preliminary optogenetic experiments showed that stimulating the CRH neurons promoted REM sleep.

We fully agree that our initial motivation for this study is not clearly explained. In our revised Introduction, we provide additional context about the unexplored nature of non-GABAergic dmM neurons (page 3, lines 46-49), and in the Results, we highlight the spatially restricted expression pattern of the *CRH* gene within the dmM (page 3, lines 64-67) as part of our rationale for investigating this population.

Q10 (Reviewer 2)

Besides, it remains that a beautiful aspect of this study is the use of complementary cutting-edge fiber photometry (calcium imaging), bidirectional optogenetic or chemogenetic technologies in CRH-cre mice to provide original and converging evidence for the critical importance of the dmM CRH neurons for dually regulating REM sleep and one of its phasic hallmarks, pontine-P waves. The experimental work in principle appears to be strong and convincing, albeit with some minor technical concerns (see below).

This is a nicely written manuscript, which has a very clear logical inductive progression from the introduction into the results, and for much of the discussion, which clearly misses, which is quite frustrating, a serious tentative to integrate these newly discovered REM-promoting neurons to the complex neuronal networks decrypted over the last decades and subserving this physiological function. In this context, it would be of particular interest to determine which of their interconnections (inputs and outputs) with other REM-promoting cell groups could precisely promote the increase of their activity at the onset of REM sleep and in close relationship with P waves.

Response Text

Following the reviewer's recommendation, we have included in the revised Discussion section more detail about how the dmM CRH may be inter-connected with other REM sleep-promoting cell groups, and we suggest possible inputs shaping the activity of these neurons (page 15, lines 369-372).

Q11 (Reviewer 2)

Finally, it is worth mentioning that iconography and Figures are in general of superior quality, nicely constructed, pertinent, complete, easy to read and illustrative, including supplementary ones. To my point of view, there is no major concerns regarding this work and manuscript but a pretty long list of minor comments and technical points that should be addressed:

1)Methods

- The number of transgenic CRH-cre mice used for each set of experiments must more clearly specify in Results section and Figures.

Response Text

We thank the reviewer for their considerate feedback, and we apologize for the incomplete sample size information. We now state in the legend of each subfigure the number of mice used for the experiment(s) shown.

Q12 (Reviewer 2)

- AAVs injection volumes used for fiber photometry and optogenetic (1ul) appear very large when compared to those used in similar studies in the field, including the present one with only 250 nl for chemogenetics. What is the rationale for such differential volumes for targeting the same brainstem nuclei? Is there a risk of AAVs leakage in LCR when infusing such large volumes in areas so close to the 4th ventricle, with a potential diffusion to other CRH cell groups ?

Response Text

Because the dmM CRH population is spatially very restricted and therefore difficult to target, we initially used high injection volumes in order to bilaterally transfect as many neurons as possible, followed by histology to confirm that there was no off-target expression. Later, we observed that smaller injection volumes worked well for chemogenetic experiments, but often resulted in poor virus expression for optogenetics and fiber photometry, a difference which may be explained by a higher efficiency of the AAV8 serotyped vector in comparison with the AAV1/2 serotypes (Broekman et al., 2006). Since our target population is not directly neighbored by other CRH neurons, we were less concerned about the possibility of co-infection. To determine whether viral expression was present in areas outside the dmM, we closely examined the histology of several mice injected with 1 μ L of ChR2-eYFP. As shown in **Reviewer Fig. 2.1**, we did not find labeled cell bodies in other CRH neuron groups, including Barrington's nucleus, the inferior olivary complex, or the paraventricular hypothalamus.

REVIEWER FIGURE 2.1. Coronal brain sections with brain areas containing CRH

neurons. A CRH-Cre mouse was injected with AAV-DIO-ChR2-eYFP into the dmM. We found no ChR2-eYFP expression in areas expressing CRH neurons outside the dmM, such as the paraventricular nucleus (PVN), Barrington's nucleus (B) and inferior olive (IO).

a Coronal fluorescence image of the paraventricular nucleus of the hypothalamus in a CRH-Cre mouse injected with 1 μ L ChR2-eYFP in the dmM. Scale bar, 200 μ m. PVN, paraventricular nucleus; f, fornix; 3V, third ventricle.

b Coronal fluorescence image of the rostral pons. Scale bar, 200 μ m. LC, locus coeruleus; B, Barrington's nucleus; scp, superior cerebellar peduncle; 4V, fourth ventricle.

c Coronal fluorescence image of the ventral medulla. Scale bar, 200 μ m. IO, inferior olivary complex; py, pyramidal tract.

Q13 (Reviewer 2)

For optogenetic experiments, AAVs injection and optic fiber implantation were done unilaterally in dmM. Does it mean that data reported result from the light-induced stimulation or inhibition of CRH neurons in only one hemisphere? If yes, why not trying bilateral stimulations to obtain more drastic effects? Did you eventually notice some physiological or behavioral side-effects due to the placement of the optic fiber tip (or miniscope) into the 4th ventricle (as shown in Fig. 3a), likely perturbing LCR flows ?

Response Text

Although we performed unilateral virus injections and fiber implantations for our optogenetic experiments, the bulk of the dmM CRH neuron population is located near the midline, and our target location was consequently only 0.2 mm mediolateral. Histological analysis revealed substantial virus expression on the contralateral side of the dmM (**Fig. 3a**), and fluorescence *in*

situ hybridization (FISH) for *cFos* and *Crh* showed that optogenetic stimulation activates CRH neurons on both sides (revised **Supplementary Fig. 4c**, and further detailed in the next comment). Thus, this finding suggests that our protocol was sufficient to manipulate the activity of dmM CRH population on both sides.

We did not observe abnormal behavior in animals with optic fibers implanted in the dmM (e.g. ChR2 mice) compared to their counterparts without fibers (e.g. hM3D(Gq) mice). Both groups appeared similar in measures of surgery recovery time, baseline sleep behavior, and general health indicators such as body weight, grooming, or motor activity. Other experimenters in our lab using similar surgical techniques in similar brain areas also did not report noticeable side effects (Stucynski et al., 2022).

Q14 (Reviewer 2)

- What is the extend of CRH neurons recruited during either optogenetic and chemogenetic activation? Did you check the efficiency of these two kinds of stimulation on the activation of CRH neurons, for example using cFos? We finally do not have any notion of the proportion of dmM CRH neurons targeted in each experiment, which is an important parameter that should be reported in the manuscript for interpreting physiological data achieved.

Response Text

We fully agree with the reviewer about the importance of analyzing the efficacy of optogenetic and chemogenetic activation in recruiting dmM CRH neurons. To address this concern, we quantified the proportion of dmM CRH neurons that were activated by our experimental stimulation protocols, using the expression of *cFos* as a marker of neural activity. We used double FISH to identify dmM neurons expressing both *Crh* and *cFos* mRNA.

We found that both optogenetic (**Supplementary Fig. 4a-c**) and chemogenetic (**Supplementary Fig. 6a,b**) activation techniques were very successful in recruiting dmM CRH neurons. In ChR2 mice, 89.7% of the counted CRH neurons co-expressed *cFos* (538/600 cells, $n = 3$ mice), and for hM3D(Gq) mice, 90.4% of the CRH neurons were activated (461/510 cells, $n = 3$ mice). Notably, laser stimulation delivered through a unilateral optic fiber was effective in bilaterally activating CRH neurons (**Supplementary Fig. 4c**), likely due to the close proximity of this population to the midline. We also detected *cFos* expression in some non-CRH dmM neurons, whose activation may be due to local excitation by the dmM CRH neurons. We have included these data in **Supplementary Figs. 4 and 6**, and summarized the findings in the revised Results section (page 7, lines 150-154; page 12, lines 285-287), and we describe the experimental protocol for the *cFos* studies in the revised Methods (page 26, lines 629-640).

Q15 (Reviewer 2)

- A very low dose (0.25 mg/kg) and a very high one (5mg/kg) of CNO were used for the chemogenetic excitation and inhibition, respectively. Why using two different doses, whereas 1mg/kg is usually used in similar studies?

Response Text

In our lab's experience with multiple mouse lines and brain areas, we have noticed that for mice expressing the excitatory DREADD hM3D(Gq) very low doses of CNO often have strong effects. In fact, for chemogenetic activation we avoid doses over 1 mg/kg, since we have seen that such doses can lead to seizure-like states, particularly when activating excitatory neurons. Conversely, chemogenetic inhibition using hM4D(Gi) typically requires larger doses to produce a

sizable effect. Consistent with our experience, several previous studies employing both chemogenetic activation and inhibition have used different CNO doses for hM3D(Gq) and hM4D(Gi). For instance, Naganuma et al. (2019) used 0.3 mg/kg and 1.5 mg/kg for chemogenetic activation and inhibition, respectively, in the lateral hypothalamus. Zhang et al. (2019) injected 1 mg/kg to excite and 5 mg/kg to inhibit neurons in the midbrain, while Funk et al. (2017) and Torontali et al. (2019) applied 5 mg/kg and 10 mg/kg for excitation/inhibition of neural populations in the cortex and pons, respectively.

Q16 (Reviewer 2)

On the same topic, no effects on both sleep-waking cycle and power spectral densities across vigilance states were found after CNO administration in control mice. This is quite surprising since recent publications have shown that CNO is not an inert compound and may have by itself non-specific dose-dependent pharmacological effects on vigilance states or EEG rhythmic activities. When analyzing in detail data illustrated in supplementary Fig. 6d-f after CNO treatments in control mice, a tendency (although not significant) is yet present for a dose-dependent increase of REM sleep percentage, duration and bout frequency.

Response Text

We agree with the reviewer that given recent findings of CNO causing non-specific changes in sleep, we must rigorously validate that the results of our chemogenetic experiments are truly due to manipulation of dmM CRH neuron activity. To account for off-target effects of CNO injection on the sleep-wake cycle, including potential dose-dependent increases in REM sleep, we have expanded the statistical analyses in **Fig. 5** and **Supplementary Figs. 6** and **7** by directly comparing hM3D(Gq) and hM4D(Gi) animals with their appropriate mCherry controls (see below). We have also provided quantification for the impact of CNO on different power bands of the EEG (**Supplementary Fig. 6h**; **Supplementary Fig. 7f**). Within a given group of mice expressing hM3D(Gq), hM4D(Gi), or mCherry, we compared the EEG of each brain state between saline and CNO trials, using two-way repeated measures ANOVA with drug and frequency band as within factors. We found that the 0.25 mg/kg dose of CNO used for chemogenetic excitation did not significantly alter the EEG in hM3D(Gq) or mCherry mice, compared with saline. However, for the 5 mg/kg dose used in chemogenetic inhibition experiments, we found significant interactions between drug (saline vs CNO) and frequency band power during IS in hM4D(Gi) mice and during NREM sleep in mCherry mice. Post-hoc t-tests with Holm correction did not reveal a significant difference in the power of any frequency band between saline and CNO trials for either group, though in mCherry mice there was a trend ($P = 0.10$) toward a CNO-induced increase in delta power during NREM sleep. This tendency is consistent with a recent study reporting an enhanced low frequency power during NREM sleep following treatment with higher doses of CNO (Traut et al., 2022). The statistical methods used for these EEG comparisons are described in the revised Methods section (page 30, lines 721-736).

Q17 (Reviewer 2)

Moreover, the baseline values of the same parameters look like somewhat different after saline treatment in control vs. experimental mice (supplementary Fig. 6d-f vs Fig. 5d-g vs Fig. 5 l-n). For example, the mean duration of REM sleep episodes is around 60 sec in experimental mice while they lasted more than 80 seconds in control mice, a difference that could have finally some importance at statistic levels and regarding data significance. How can these differences between samples of mice be explained? Thus two obvious questions came in my mind: again, how many mice were included in the different samples? Did you take into consideration data

obtained in control mice for your statistical comparison of saline vs. CNO effects in experimental mice or did you consider that CNO has no potential side-effect? In other words, what were your control conditions for your analyses? This crucial aspect must be more clearly explained in the manuscript.

Response Text

We appreciate the reviewer's criticism regarding the control conditions for our chemogenetic experiments, and we agree that our statistical analysis must be more clearly explained. Initially, we evaluated the effects of the drug dose separately for the hM3D(G)q, hM4D(G)i, and mCherry control groups. In our revised **Fig. 5** and **Supplementary Figs. 6 and 7**, we now also directly compare each experimental group with its mCherry control, using mixed ANOVA with virus and drug as between and within factors, followed by pairwise t-tests with Holm correction. Thus, our analysis takes into account any baseline differences between mouse groups, as well as non-specific effects of CNO. To exclude the possibility that the mentioned differences in REM sleep (e.g. mean REM duration) between experimental and control mice were the result of a small sample size, we performed the same experiments in additional cohorts. After updating our dataset, we indeed found no significant differences in sleep behavior between hM3D(G)q and mCherry mice during saline trials, nor between hM4D(G)i mice and their mCherry controls (**Fig. 5d-k; Supplementary Fig. 6c-e; Supplementary Fig. 7a-c**), suggesting that the mentioned differences were within the range of variability across animals.

We observed that the frequency of P-waves during REM sleep strongly differed across animals. Given this large variability, we performed additional estimation statistics (Calin-Jageman & Cumming, 2019) to confirm the robustness of our results. More precisely, we used bootstrapping to estimate the sampling distribution of the mean difference in P-wave frequency (**Methods**) between saline and CNO recordings for experimental and control mice. We used these distributions to estimate for each mouse group the 95% confidence intervals (CIs) for CNO-induced changes in P-wave frequency. For both chemogenetic activation and inhibition, the 95% CIs for the experimental and control groups did not overlap (**Fig. 5n,p** bottom), demonstrating that exciting or inhibiting dmM CRH neurons significantly increased or decreased, respectively, the P-wave frequency during REM sleep.

In the revised manuscript, we now report in the Figure Legends for each analysis in **Fig. 5**, **Supplementary Fig. 6**, and **Supplementary Fig. 7** the main effect of the drug (saline vs CNO), the interaction between drug and virus (hM3D(G)q or hM4D(G)i vs mCherry), and any significant post-hoc comparisons.

Q18 (Reviewer 2)

2)Results

- In my mind and in line with anatomical data from Allen mouse brain atlas, vglut1 is not that much expressed within brainstem nuclei compared to forebrain and cortex mantles, in contrast to vglut2 which is a specific marker of brainstem glutamate neurons. Maybe it could be more relevant to illustrate vglut2 ISH within dmM in the Fig. 1a than in supplementary Fig. 1a.

Response Text

In fact, while writing the manuscript, we were debating about the same question. Vglut2 is widely expressed throughout brainstem and subcortical areas and therefore commonly used as a marker for excitatory neurons in those regions. In contrast, Vglut1, while being widely expressed throughout the cortex, is only present in very specific brainstem areas. In the end,

because Vglut1 is sparsely expressed in the brainstem but nevertheless closely overlaps with the CRH population, we felt that their relationship provided a more informative description of the CRH neurons. But to make clear to the general readership that Vglut1 is a more unconventional and therefore surprising marker for excitatory brainstem neurons, we now specifically mention in the revised Results that Vglut1 is sparsely expressed in the medulla (page 4, lines 71-73).

Q19 (Reviewer 2)

- Page 6 (end of the first paragraph: "Our fiber photometry experiments in both REM sleep and regulation its phasic components". What are included exactly in these "phasic components", just P- waves (correlated to phasic theta activation) as nicely demonstrated in the present study or all other phasic events that characterized REM sleep, as distal muscle twitches, vibrissa movements, penile erection, and of course REMs themselves (as PH / mVN are involved in the control of horizontal eye movements)? In that context, did you observe an increased frequency of REMs (or other phasic events) during REM sleep in response to both optogenetic (short or long) and chemogenetic activation of CRH- neurons ? If yes, did they occur in a synchronized way? In all cases, it would be of particular interest to be more precise on that aspect in Results and Discussion sections.

Response Text

We now recognize that our use of the phrase "phasic components" is ambiguous, as we intended only to refer to the P-waves and theta bursts recorded in our experiments. We also agree that our study would benefit from an analysis of other types of phasic events mentioned by the reviewer. As our lab is not currently set up to record REMs or vibrissa movements, we tried to automatically detect muscle twitches in the nuchal EMG during REM sleep. Unfortunately, we found that at least in our recording system, the peaks in EMG amplitude corresponding to phasic twitches were not consistently well-separated from background fluctuations during REM sleep, with a particular confounder being the prominent ticks caused by heartbeats (R-waves). Though the lack of a direct twitch detection method prevented further analyses with our chemogenetic tools, we found that we could use other events of interest, such as detected P-waves and brief optogenetic laser pulses, as triggers to examine the EMG amplitude in the surrounding time window. We found that both spontaneous and laser-triggered P-waves during REM sleep were temporally associated with large transient increases in the EMG amplitude, while there were no significant changes in the EMG activity surrounding laser pulses that failed to trigger a P-wave (**Supplementary Fig. 5j,k**). These findings indicate that dmM CRH neurons promote P-waves along with phasic changes in muscle activity and theta oscillations, and further suggest that P-waves are a necessary prerequisite for the occurrence of latter two events. We discuss the results of this experiment in the revised Results section (page 11, lines 262-272).

Q20 (Reviewer 2)

- Pages 7-8: "Thus, laser-induced changes in REM sleep appear to trigger endogenous homeostatic processes regulating the overall amount of REM sleep". That is really an interesting hypothesis regarding the potential role of these CRH neurons. However, it seems too early to conclude in that way since homeostatic mechanisms of REM sleep were not at all experimentally challenged in this study. Based on short lasting optogenetic stimulations, the present data only show elegantly that CRH neurons seem to play a role firstly in the ultradian rhythm of REM sleep. It is generally believed that processes of REM sleep homeostasis take place within the hypothalamus, although a definitive demonstration is still lacking. In that context, did you observe direct efferent projections emanating specifically from CRH neurons

and reaching the lateral hypothalamic area where numerous neurons are activated during REM sleep rebound in response to its specific deprivation? Is there any possibility that CRH themselves may be a part of these homeostatic processes and if yes, how to demonstrate that using your experimental paradigms?

Response Text

In our optogenetic closed-loop experiments, the laser was only turned on during a randomly selected subset (~50%) of the detected REM sleep episodes (**Fig. 3f**). Closed-loop activation of dmM CRH neurons increased the duration of REM sleep episodes compared with laser-off episodes, while closed-loop inhibition had the opposite effect. Interestingly, we found that the average duration of REM sleep, calculated across both laser-on and laser-off episodes, was unchanged in the experimental groups (Chr2 and iC++) compared to the control groups, and that the total amount of REM sleep was approximately the same as well (**Supplementary Fig. 4I** in the revised manuscript). This finding suggests the existence of a homeostat that keeps track of the average duration and amount of REM sleep; i.e., when a REM sleep episode is extended by Chr2-mediated closed-loop activation, the animal's homeostatic need for REM sleep may be momentarily fulfilled or even exceeded, and as a consequence, ensuing laser-off episodes may be shortened to keep the mean amount of REM sleep within the right range. Similarly, shortening of REM sleep episodes by iC++ stimulation may be followed by longer laser-off episodes to compensate for lower levels of REM sleep. Consistent with this notion, a recent study in mice showed that the duration of two successive REM sleep periods is anti-correlated (Park et al., 2021).

While our closed-loop data speaks to the existence of a homeostat, several key questions remain unresolved. First, it is unclear from the present experiments on which timescale the homeostatic process operates. A prominent view in the REM sleep literature is that the overall amount and timing of REM sleep are regulated by two separate systems; a short-term process (acting on the ultradian time scale) influences when REM sleep is initiated and how long it lasts, while a long-term process regulates the daily quota of REM sleep (Franken, 2002). Second, it is unknown whether the dmM CRH neurons are part of the REM sleep homeostat or, as pointed out by the reviewer, to what extent they interact with areas involved in REM sleep homeostasis. Anterograde viral tracing did not reveal direct axonal projections to the lateral hypothalamus (**Reviewer Fig. 2.2**), though this does not preclude the possibility that dmM CRH neurons influence REM sleep homeostasis through some indirect mechanism. We fully agree with the reviewer that to address these questions would require challenging the REM homeostat using REM sleep deprivation, while manipulating the activity of dmM CRH neurons during the deprivation and recovery period. While such experiments would be of great interest, we believe that they are beyond the scope of the present revision. However, we have realized that the sentence in question without further explanation may cause misunderstandings. In the revised Results section, we have dropped this statement and instead explain in more detail why our findings from closed-loop manipulation of CRH neuron activity suggests the presence of a homeostatic process preserving the average duration and amount of REM sleep (page 9, lines 203-211).

REVIEWER FIGURE 2.2. dmM CRH neurons do not send direct projections to the lateral hypothalamus.

a Coronal fluorescence image of the hypothalamus in a CRH-Cre mouse injected with 1 μ L ChR2-eYFP in the dmM. Scale bar, 0.5 mm. LH, lateral hypothalamus; MEA, medial amygdalar nucleus; mtt, mammillothalamic tract; f, fornix; int, internal capsule.

Q21 (Reviewer 2)

- Page 9: "The efficacy of dmM CRH neurons being above 90% during NREM sleep and IS, 66% during REM sleep....". I'm not sure to understand what exactly mean this sentence and percentages, confusing and somewhat contradictory to the general message. Does it mean it is easier to generate P- waves during natural SWS (during which only occasional spontaneous P-waves occur) than during REM sleep (during which numerous single and clusters of P-waves occur) by short optogenetic excitation of CRH neurons? How can we explain that seemingly unexpected results?

Response Text

The observed discrepancy in laser success may be explained to a large degree by the relationship between P-waves and theta oscillations during REM sleep. We show in **Fig. 4i** that both spontaneous and laser-triggered P-waves exhibit significant EEG phase preferences, with the greatest proportion of P-waves occurring during the rising phase ($-\pi$ to 0 rad) of the theta oscillation, as previously reported in mice (Tsunematsu et al., 2020). The success rate of laser pulses in triggering P-waves was also influenced by the power and frequency of the ongoing theta oscillations during REM sleep (**Supplementary Fig. 5d-f**). Under optimal conditions (low theta power and frequency, rising phase), the success rate was close to 100%. As mentioned in the Discussion, the modulation of the success rate by the theta oscillations suggests the presence of a postsynaptic mechanism influencing the impact of CRH neuron laser stimulation on the downstream neurons. This explanation is in line with electrophysiological recordings in the DTg and NI demonstrating that neurons in these downstream areas exhibit a strong modulation of their spiking activity by theta oscillations. Alternatively, feedback projections from theta-generating circuits may modulate the excitability of dmM neurons, thereby influencing the impact of laser stimulation on dmM CRH activity.

For clarification, we now explicitly state in the revised Results section that the dependence of P-wave induction on theta oscillations may explain the lower efficacy of laser pulses during REM sleep compared with NREM sleep (page 11, lines 258-261).

Q22 (Reviewer 2)

3) Figures

- *Figure 1f: What mean negative activity values of calcium imaging (below zero levels) during Waking and NREM sleep ?*

Response Text

We quantified calcium activity as the change in the GCaMP6s-dependent fluorescence signal $F(t)$ relative to the baseline signal $F_0(t)$ (see **Methods**).

$$\frac{\Delta F}{F}(t) = \frac{F(t) - F_0(t)}{F_0(t)}$$

As a brief summary, in our calcium imaging experiments, we shine light with different wavelengths through the optic fiber into the brain: the excitation wavelength (460 nm) is used to collect the calcium-dependent signal from GCaMP6s, while the isosbestic wavelength (405 nm) is a calcium-independent control for photobleaching and motion artifacts. As in previous studies (Lerner et al., 2015; Cho et al., 2017), we use linear regression to fit the 405 nm to the 465 nm signal; the resulting linear fit serves as the reference signal $F_0(t)$, while the 465 nm signal itself serves as the raw fluorescence signal $F(t)$. While the calcium-dependent fluorescence $F(t)$ cannot be negative, the relative calcium signal $\Delta F/F$ may become negative, depending on the exact value of $F_0(t)$. For example, as the dmM CRH neurons become strongly activated during REM sleep (compared to NREM sleep), the linear fit tries to account for the REM epochs with heightened calcium activity. Consequently, during NREM sleep, the baseline signal $F_0(t)$ is slightly higher than the calcium signal $F(t)$, resulting in negative values of $\Delta F/F$.

Q23 (Reviewer 2)

- *Figure 3h-k: As previously underlined for the chemogenetic data (see above and supplementary Fig.6d-f), how to explain such important differences in REM sleep duration in control conditions between ChR2 vs. iC++ experiments (with a 2x difference)?*

Response Text

This concern is related to the previous comment (Q20) on the sentence “Thus, laser-induced changes in REM sleep appear to trigger endogenous homeostatic processes regulating the overall amount of REM sleep”, and we will therefore repeat some of the arguments, although in greater detail.

When considering the difference in the average duration of laser-off REM sleep episodes between the ChR2 and iC++ groups, we would like to clarify that laser-off REM sleep is not in fact a ‘control condition’ free from experimental influence. Instead, as a laser-on REM episode is extended or shortened by closed-loop stimulation, the resulting change in REM sleep amount may lower or raise the animal’s homeostatic pressure for REM sleep, which in turn directly impacts the duration of a subsequent laser-off REM episode. Importantly, we found that the average REM sleep duration computed across both laser-on and laser-off episodes, as well as the overall percentage of REM sleep, was statistically indistinguishable between ChR2 and iC++ groups and their corresponding controls (**Supplementary Fig. 4I** in the revised manuscript). Thus, while the duration of REM sleep episodes with closed-loop activation was extended, the duration of laser-off episodes was shorter, such that the average duration across laser-on and laser-off episodes and the mean percentage of REM sleep was similar for ChR2 and eYFP mice. Conversely, shortening REM sleep episodes by closed-loop inhibition led to shorter

laser-off episodes. This finding suggests the existence of a homeostat that keeps track of the current amount of REM sleep and accordingly adjusts (or biases) the duration of REM episodes to stay near some optimal amount of REM sleep. Consistent with this concept, a recent study in mice showed that the duration of two successive REM periods is anti-correlated (Park et al., 2021). Thus, laser-induced extensions of REM sleep episodes in Chr2 mice are expected to be followed by shorter laser-off episodes, and vice versa for laser-induced shortening of REM sleep episodes in iC++ mice. In the revised Results section, we now clearly state that these homeostatic mechanisms may explain the observed differences in the duration of laser-off REM bouts between Chr2 and iC++ mice (page 9, lines 197-211).

To more clearly illustrate the relationship between laser-off and laser-on REM sleep bouts, and to demonstrate that the difference in REM episode duration between the two is a robust statistical result rather than a coincidental finding, we have included new statistical analyses in the revised **Fig. 3h,i**. Using bootstrapping analysis (Calin-Jageman & Cumming, 2019), we estimated the distribution of the mean difference in duration between laser-on and laser-off episodes (Δ Duration). First, assuming we had n mice in our original dataset, we chose a new sample of n mice by randomly selecting with replacement. Next, for each mouse i having a total of m_i REM episodes, we randomly selected (with replacement) a new set of m_i REM episodes and calculated the mean difference in duration between the resampled laser-on and laser-off episodes. We repeated this procedure 10,000 times, to bootstrap the distribution of the mean difference (Δ Duration) and to determine a confidence interval (CI). We found that the bootstrapped distribution mean was just under 40 s for the Chr2 dataset and approximately -40 s for the iC++ dataset, while the means for both eYFP groups were near zero. Importantly, the CIs for the Chr2 and iC++ groups lay outside those of their eYFP controls, demonstrating that laser stimulation significantly changes REM sleep duration in experimental subjects, compared with control mice.

Following this analysis, we next estimated the likelihood of observing a result similar to the measured difference between laser-on and laser-off REM episodes simply by chance, given the same number of mice and the same number and distribution of REM sleep durations. More precisely, we randomly shuffled for each mouse the labeling of laser-on and laser-off REM sleep episodes (i.e., if a mouse had m_{on} laser-on trials and m_{off} laser-off trials, we randomly selected m_{on} REM episodes and labeled them as laser-on, and labeled the remaining m_{off} episodes as laser-off). Next, we selected m_i REM episodes from each mouse i as described above, and calculated the mean difference between the duration of shuffled laser-on and laser-off REM episodes. For bootstrapping, we repeated this procedure 10,000 times. For both Chr2 and iC++ datasets, we found that the CI for the true mean difference did not overlap with the CI for the shuffled mean difference, showing that the observed laser-induced difference in REM sleep duration is a true effect, mediated by the stimulation and inhibition of dmM CRH neuron activity. We have included these analyses in **Fig. 3h,i**, and we discuss our findings in the revised Results section (page 8, lines 180-190; page 9, lines 195-196).

Broekman, M. L. D., Comer, L. A., Hyman, B. T. & Sena-Esteves, M. Adeno-associated virus vectors serotyped with AAV8 capsid are more efficient than AAV-1 or -2 serotypes for widespread gene delivery to the neonatal mouse brain. *Neuroscience* **138**, 501–510 (2006).

Calin-Jageman, R. J. & Cumming, G. Estimation for Better Inference in Neuroscience. *eNeuro* **6**, ENEURO.0205-19.2019 (2019).

Cho, J. R. *et al.* Dorsal Raphe Dopamine Neurons Modulate Arousal and Promote Wakefulness by Salient Stimuli. *Neuron* **94**, 1205-1219.e8 (2017).

Franken, P. Long-term vs. short-term processes regulating REM sleep. *J. Sleep Res.* **11**, 17–28 (2002).

Funk, C. M. *et al.* Role of Somatostatin-Positive Cortical Interneurons in the Generation of Sleep Slow Waves. *J Neurosci* **37**, 9132–9148 (2017).

Goutagny, R. *et al.* Role of the dorsal paragigantocellular reticular nucleus in paradoxical (rapid eye movement) sleep generation: a combined electrophysiological and anatomical study in the rat. *Neuroscience* **152**, 849–857 (2008).

Kaur, S., Saxena, R. & Mallick, B. N. GABAergic neurons in prepositus hypoglossi regulate REM sleep by its action on locus coeruleus in freely moving rats. *Synapse* **42**, 141–150 (2001).

Lerner, T. N. *et al.* Intact-Brain Analyses Reveal Distinct Information Carried by SNc Dopamine Subcircuits. *Cell* **162**, 635–647 (2015).

Naganuma, F. *et al.* Lateral hypothalamic neurotensin neurons promote arousal and hyperthermia. *PLOS Biology* **17**, e3000172 (2019).

Park, S.-H. *et al.* A probabilistic model for the ultradian timing of REM sleep in mice. *PLoS Comput. Biol.* **17**, e1009316 (2021).

Stucynski, J. A., Schott, A. L., Baik, J., Chung, S. & Weber, F. Regulation of REM sleep by inhibitory neurons in the dorsomedial medulla. *Curr. Biol.* **32**, 37–50 (2022).

Torontali, Z. A., Fraigne, J. J., Sanghera, P., Horner, R. & Peever, J. The Sublaterodorsal Tegmental Nucleus Functions to Couple Brain State and Motor Activity during REM Sleep and Wakefulness. *Curr. Biol.* **29**, 3803-3813.e5 (2019).

Tsunematsu, T., Patel, A. A., Onken, A. & Sakata, S. State-dependent brainstem ensemble dynamics and their interactions with hippocampus across sleep states. *eLife* **9**, e52244 (2020).

Verret, L., Fort, P., Gervasoni, D., Léger, L. & Luppi, P.-H. Localization of the neurons active during paradoxical (REM) sleep and projecting to the locus coeruleus noradrenergic neurons in the rat. *J. Comp. Neurol.* **495**, 573–586 (2006).

Zhang, Z. *et al.* An Excitatory Circuit in the Periocolomotor Midbrain for Non-REM Sleep Control. *Cell* **177**, 1293-1307.e16 (2019).

Reviewer #3 (reviewer's comments in italics)

The authors investigated the brainstem cellular circuits that contribute to the generation of PGO waves. PGO or P-waves in mice are thought to coordinate brain activity and regulate functional sleep processes including memory consolidation. The author's investigation aims at providing additional evidence on the nature of cells in the dorsomedial medulla (dmM) contributing to P-waves in mice. The authors identified a sub-population of excitatory cells – i.e., corticotropin-releasing-hormone, (CRH) - neurons in the dmM that using calcium imaging (fibre photometry) show that these cells increase their activity during REM in freely-moving mice. Opto-/pharmaco-genetics approaches aimed to show the CRH causal modulation of expression of REM sleep and of P-waves.

This study provides some evidence of a novel cell population modulating REM sleep and important features such as the P-waves. This is an important work for the field of (experimental) sleep and the general understanding of the modulation of vigilance states. However, there are major concerns that need to be addressed before publication. Although it is clear that there is an increase in the activity of CRH cells during REM sleep, the evidence of the modulation of P-waves is weak and the causal evidence to determine whether or not these cells are indeed causal to the generation of P-waves is inconclusive. In addition, the clarity of the statements and interpretation of the results should be improved, in particular regarding the REM modulation and P-wave generation.

Response Text

We thank the reviewer for the thorough feedback and suggestions to strengthen the manuscript.

Q24 (Reviewer 3)

Major changes:

1 - Authors stated that the activity of CRH neurons is modulated by REM sleep and P-waves during REM sleep without supporting experimental evidence. The increase of activity of these cells occurs after the peak of the P-wave and therefore the causal link made by the authors is not supported by the data

Response Text

For **Fig. 2i**, the reviewer comments that the increase in dmM CRH neuron calcium activity follows the peak of the P-wave, and therefore appears to contradict a causal role for this population in P-wave regulation. We have realized that the graph's coarse time scale has obscured details in the exact time course, and we have therefore replotted the $\Delta F/F$ activity using a higher temporal resolution.

When interpreting the fiber photometry data, it is important to also take into account the physical properties of the calcium sensor. Previous studies have shown, through simultaneous recordings of the GCaMP6-dependent fluorescence and single-unit spikes, that the time point of an action potential corresponds to the point at which the calcium signal starts rising from baseline, rather than the peak of the signal (Chen et al., 2013; Siegle et al., 2021). Particularly when a neuron fires a sequence of multiple action potentials, the summation of slowly decaying calcium events can produce a fluorescence signal with a considerably delayed peak; this is especially the case for GCaMP6s, due to its slow off-kinetics. Consequently, when determining whether neuronal activity precedes or follows P-waves, the most informative time point is not the maximum of the calcium signal, but the point where the signal starts significantly increasing above baseline. As summarized below, we performed additional analyses to show that dmM CRH calcium activity started rising from baseline before the time points (negative peak) of the P-waves.

For the revised **Fig. 2j**, we examined the relationship between dmM CRH calcium activity and P-waves in finer detail, during the 2 s time window surrounding P-waves. Comparing the mean fluorescence signal during the baseline interval (-1 to -0.8 s) with each consecutive 0.2 s time bin, we found that the neuronal activity began significantly increasing from baseline 0.4 s before the P-wave occurs ($P < 0.05$, t-tests with Bonferroni correction), consistent with a role of dmM CRH neuron activity in promoting P-waves.

Importantly, however, we agree with the reviewer that calcium imaging experiments alone do not establish a causal link between dmM CRH neuron activity and P-wave induction. Rather, these experiments only provided correlative evidence for an involvement of these neurons in P-wave regulation. To investigate the direct functional relationship between dmM CRH activity and P-waves we therefore performed opto- and chemogenetic manipulations (**Fig. 4; Fig. 5**). To dispel the impression that fiber photometry recordings alone establish a causal link, we now clearly state in the revised Results that these experiments provide only correlative evidence for a role of dmM CRH neurons in P-wave regulation (page 7, lines 145-146).

Q25 (Reviewer 3)

Figure 1 reports a very long distance between the optical fibre placed within the ventricle (used for photometry) and the targeted cells. How can the authors be sure that they actually do record the calcium transients of those cells? For instance, transients could result from movement artefacts or changes in CSF volumes across states. This should be clarified.

Response Text

Importantly, based on the histology picture in **Fig. 1d**, we cannot derive the exact position of the optic fiber tip. From the lesion, we can only conclude that the fiber penetrated through the cerebellum and was thus located within the fourth ventricle. We have realized that drawing the rectangular outline may therefore be misleading, and so we have removed the outline and now instead use an arrow to indicate the clearly visible lesion from the optic fiber in the cerebellum.

From the histology image, the maximum distance between the top of the dmM (where the CRH cells are located) and the bottom of the cerebellum (which was penetrated by the optic fiber) is 300 μm . Based on our lab's experience, this distance is within a reasonable range for recording a reliable GCaMP6s fluorescence signal. Since the dmM CRH neuron population closely neighbors the fourth ventricle, an ideally located optic fiber will directly contact the dorsal surface of the dmM, without piercing the medulla. Any penetration into the dmM risks leaving the fiber tip below many of the target CRH neurons. However, we fully agree with the need for controls to exclude the possibility that the $\Delta F/F$ signal is the result of various sources of noise, instead of reflecting the true calcium-dependent fluorescence of GCaMP6s. Importantly, we used two different LED wavelengths in our fiber photometry experiments. The 465 nm wavelength activated the calcium-dependent GCaMP6s fluorescence, and the isosbestic (405 nm) wavelength served as a control for non-calcium related changes in fluorescence, for instance caused by motion artifacts, photo-bleaching, and possibly changes in CSF volumes (Lerner et al., 2015). To confirm that the recorded $\Delta F/F$ signal specifically represents the calcium activity of dmM CRH neurons, we have repeated the analyses in **Fig. 2i** using the calcium-independent signal for the 405 nm wavelength. We found that P-waves were not associated with significant changes in the isosbestic signal (**Supplementary Fig. 2j**), demonstrating that the rise in the $\Delta F/F$ signal during P-waves in **Fig. 2i,j** represents the calcium-dependent fluorescence of the GCaMP6s sensor. This control analysis has been added to **Supplementary Fig. 2**, and we mention our findings in the revised Results section (page 6, lines 135-136).

Q26 (Reviewer 3)

In addition, Figure 1. Please include a higher magnification of the cells expressing GCaMP (d, middle panel). Current images do not really show a healthy cell.

Response Text

We have included an inset panel in **Fig. 1d** showing GCaMP6s-expressing neurons at higher magnification.

Q27 (Reviewer 3)

Figure 2, what is the rationale for recording P-Waves in the LC ? – this should be clarified.

Response Text

We believe that there may have been a misunderstanding owing to how we labeled “LC” and “SubC” in **Fig. 2a**. Importantly, the pontine LFPs were recorded at the end of the electrode tract (indicated by the white arrowhead), which is located below the LC in the SubC. **Supplementary Fig. 3a** summarizes the location of all the electrode tips. In the revised **Fig. 2a**, we have removed the outline for the LC and increased the size of the arrowhead, to more saliently identify the lesion from the electrode tip.

Q28 (Reviewer 3)

Moreover, Fig 2i suggests that activity lags from the peak of the P-wave in single events and it is more correlated with the second, smaller peak in the cluster events than the first one. This may be due to the experimental design (electrode and imaging in different locations) or the alignment of the electrophysiological signals with calcium signals. A cross-correlation analysis will strengthen should be provided to support the author’s claim.

Response Text

In the original **Fig. 2i**, the timing of the P-wave in the raw LFP trace (top row) cannot be directly compared to the time course of dmM CRH neuron calcium activity (middle and bottom rows) because these plots span different time intervals. Our rationale for using separate time scales was that the calcium signal is slow compared to electrophysiological signals; as mentioned above, the slow kinetics of the GCaMP6s sensor are reflected in the $\Delta F/F$ signal, which consequently may peak only after a burst of action potentials, and it may take hundreds of milliseconds to return to baseline. Therefore, we plotted the calcium activity in the time range of -10 to 10 s to show the full rise and decay of the fluorescence signal surrounding P-waves (0 s). For P-waves themselves, which are fast electrophysiological events lasting less than 100 ms, we chose an interval from -1 to 1 s to make the fine details of the waveforms better visible. We have realized that this arrangement may be confusing, and we have therefore removed the raw trace from the revised **Fig. 2i**, while our new **Fig. 2j** shows both the P-wave and the calcium activity on the same expanded time scale.

We agree with the reviewer that a cross-correlation analysis is an excellent suggestion, providing further insight into the temporal relationship between the dmM CRH neuron activity and P-waves; we have included the results of this analysis in **Supplementary Fig. 2h**, and we mention our findings in the revised Results section (page 6, lines 136-140). During each REM sleep period, we computed the linear cross-correlation of the $\Delta F/F$ signal with a binary ‘P-wave train’ of 0’s and 1’s, where a 1 indicates a P-wave at the given time point. Both signals were discretized using 5 ms bins. Consistent with our results in **Fig. 2i**, we found that the cross-correlation started rising before 0 s, indicating that the activity of dmM CRH neurons starts

increasing slightly before the P-waves. As a control, we randomly jittered the positions of the 1's in each P-wave train, and found no correlation between dmM CRH calcium activity and these shuffled time points.

Q29 (Reviewer 3)

Figure 4h shows that the latency to the induced P-waves happened a few milliseconds. This does not support the author's claims that the activity of CRH controls P-waves at all.

2-Please clarify 1) the changes in the temporal resolution of this analysis and 2) how many of the total amount of laser pulses induced clear P-waves as well as the proportion of detected PW/triggered PW. It is not clear if this is the case in 4h (sham laser?). Authors need to provide a clear quantification of the total PW trigger in YFP and Ch2 animals as well as YFP- IC++.

Response Text

We believe that the reviewer's concern is largely related to the apparent discrepancy in the time scales used for the analysis of the calcium imaging (**Fig. 2**) and the optogenetic experiments (**Fig 4**). While optogenetic stimulation triggers P-waves within tens of milliseconds, the calcium signals appear to evolve over seconds. This apparent inconsistency is likely the result of the slow dynamics of the calcium indicator; as mentioned above, studies simultaneously recording action potentials and GCaMP signals from the same neurons demonstrate that the time point of a spike does not correspond to the peak in the resulting calcium transient, but the time point where the calcium signal starts rising from baseline (Fig. 3a in Chen et al., 2013). Consequently, the peak of the calcium signal is delayed relative to the spike, and lags even further behind the first spike in a burst of action potentials. This effect is further pronounced for the GCaMP6s indicator, a variant with high calcium sensitivity but slower kinetics. As a result, when relating the fluorescence signal with the true underlying spiking activity, the time point where the $\Delta F/F$ signal starts rising is more informative than the (delayed) peak time. In the revised **Fig. 2i-k**, we now provide a more temporally refined analysis of the calcium signals relative to P-waves.

We acknowledge that because of the low temporal resolution of the calcium signal, the fiber photometry experiment provides only correlational evidence for a role of these neurons in P-wave regulation. To provide causal evidence for dmM CRH activity in promoting P-waves, we therefore performed opto- and chemogenetic experiments. Due its millisecond precision, optogenetics was particularly well suited to study the impact of ChR2-mediated dmM CRH activation within the appropriate time scale, as shown in **Fig. 4h**.

Following the reviewer's advice, we quantified for each brain state the total number of successful laser pulses (i.e., pulses inducing a P-wave) and failed laser pulses across all recordings from ChR2 and eYFP mice (**Supplementary Fig. 5h**). We also report the total numbers of laser-triggered and spontaneous P-waves across brain states, and found that approximately 14% (4,019 / 29,582) of the total number of P-waves in ChR2 mice were optogenetically elicited, while laser-triggered P-waves constitute less than 0.5% (40 / 11,422) of all P-waves recorded in eYFP animals (**Supplementary Fig. 5i**). In addition, while approximately 65% of laser pulses in ChR2 mice are followed by a P-wave, in eYFP controls only less than 5% are followed by a P-wave. These new analyses are included in the revised **Supplementary Fig. 5**, and summarized in the revised Results section (page 10, lines 235-242).

In **Fig. 4**, the "sham laser" group is a control condition, verifying that the observed success rate of P-wave induction (**Fig. 4g**, right) and the distribution of delay times between single laser pulses and P-waves (**Fig. 4h**) in ChR2 mice are truly the result of laser stimulation rather than

coincidental occurrences. To calculate the distribution expected if there was no relationship between the laser and P-waves, we randomly shifted (up to ± 5 s) the time point of each true laser pulse during REM sleep and repeated both analyses, with “successful” shuffled pulses defined as those which happen to closely precede one or more “laser-triggered” P-waves simply by chance. This method removes the association between the timing of laser pulses and P-waves while preserving the total number of each event, as well as the overall statistics of REM sleep, and the results of the analysis confirm that the success rate of P-wave induction during REM sleep and the short delay times between laser onset and P-waves are highly unlikely to be observed by chance. We have realized that the term “sham laser” may be misleading and have therefore replaced it with “shuffled laser” to clarify that this is a statistical rather than a physical control.

Of note, we did not apply the step pulse protocol in mice expressing iC⁺⁺. Given the relative infrequency of P-waves (typically fewer than one per second), it would likely require a very large number of trials and mice to have sufficient statistical power to conclude whether silencing these neurons suppresses the number of P-waves throughout the 10 or 20 ms long stimulation window. Instead, we felt that the sustained inhibition of dmM CRH neurons conferred by chemogenetics (**Fig. 5**) is a more appropriate and effective method for studying the necessity of these neurons in P-wave regulation.

Q30 (Reviewer 3)

3-Validation of technique is missing. The authors must provide in vivo validation of Optogenetic and DREADDS effect on the activity of CRH cells.

Response Text

We agree about the importance of validating the optogenetic and chemogenetic methods used to manipulate the dmM CRH neuron activity. To quantify the number of CRH neurons excited by optogenetic or chemogenetic activation, we sacrificed the animals after repeated optogenetic stimulation or CNO injection and then performed double fluorescence in situ hybridization (FISH) with probes against *Crh* and *cFos* (**Methods**).

We found that dmM CRH neurons were effectively recruited by both optogenetic (revised **Supplementary Fig. 4a-c**) and chemogenetic (revised **Supplementary Fig. 6a,b**) activation. In ChR2 mice, 89.7% of the counted dmM CRH neurons were activated following two hours of repeated stimulation with 10 Hz laser trains (10 s laser train every 60 s; 538/600 cells, n = 3 mice). Interestingly, we observed robust bilateral excitation of CRH neurons despite using a unilateral optic fiber, likely owing to their restricted anatomical location close to the midline. For hM3D(Gq) mice, we found that two hours after injection with CNO, 90.4% of the dmM CRH neurons were positive for *cFos* (461/510 cells, n = 3 mice). In both ChR2 and hM3D(Gq) mice, a comparably small number (<10%) of *cFos*-positive cells did not express CRH. Activation of these cells could be the result of local excitation by the glutamatergic CRH neurons. We have included the results of these experiments in the revised **Supplementary Figs. 4 and 6** and describe them in the revised Results section (page 7, lines 150-154; page 12, lines 285-287), and we detail the experimental protocol for the *cFos* analyses in the revised Methods section (page 26, lines 629-640).

Q31 (Reviewer 3)

4-Author statement: “dmM CRH activity in both inducing REM sleep and regulating its phasic component” in page 6. Evidence supporting this claim is weak:

(1) The optogenetic manipulations and the DREDDs experiments. The values of light off (control; fig. 3 h) between for control conditions of ChR2, YFP, IC++ experiments are quite different. This high variability negatively impacts the significance of the results. Furthermore, the statistical analysis is very misleading. Statistical analysis should test for the difference between control (YFP) and experimental conditions (ChR and IC++) rather than the OFF-laser conditions. This is a major flaw and a lack of scientific rigour.

(2) The power spectrum across different vigilance states (Suppl. Figure 4 and Suppl. Figure 6a-b between Gq and Gi) suggests that the random stimulation may be affecting not only REM but also other vigilant states. Quantification per frequency band must be provided with proper statistical analysis (only qualitative analysis is shown in Suppl. Fig. 4d and e). In addition, homogenise y-axis to the same scale.

(3) Lastly, Suppl. Figure 4 should reflect the same quantification as Suppl. Fig. 6 c-g.

(4) Importantly, the lack of modulation of P-waves further calls for the need for state-specific optogenetic manipulation of the CRH activity specifically during REM sleep.

Response Text

(1) We appreciate the reviewer’s criticism about our statistical approach to test effects of closed-loop manipulation on the duration of REM sleep episodes, and we have therefore largely expanded the analysis of these results. Directly comparing between groups using mixed ANOVA with virus (ChR2 or iC++ vs eYFP) and laser (laser on vs laser off) as between and within factors indicated a significant interaction between virus and laser. Post-hoc t-tests with Holm corrections further showed that laser stimulation resulted in a significant difference between the mean durations of laser-on and laser-off REM bouts in mice expressing an opsin, but not in control animals (revised **Fig. 3h,i**), corroborating our initial findings. Importantly, the laser-off condition in these studies should not be interpreted as an undisturbed baseline value that can be compared across different experimental groups, since the mean duration of laser-off REM episodes may be influenced by the optogenetic manipulation during laser-on episodes. For example, a laser-on REM episode which is prolonged or reduced by closed-loop stimulation may impact the animal’s homeostatic need for REM sleep, causing a subsequent laser-off REM episode to be correspondingly shortened or extended. Consistent with the idea of compensatory changes in the duration of laser-off episodes, we noticed that the mean duration of REM sleep (averaged across all laser-on and laser-off episodes) as well as the mean amount of REM sleep was statistically indistinguishable between ChR2 and iC++ mice and their corresponding eYFP controls (revised **Supplementary Fig. 4I**).

Even when assuming that the duration of laser-off episodes is influenced by laser-on episodes (explaining the differences in laser-off bouts between ChR2 and iC++ groups), the statistical key question remains whether the changes in the duration observed for experimental mice are real effects resulting from the manipulation of the dmM CRH neurons, or whether these differences could be merely coincidental, given the present number of mice in each dataset and the distributions of REM sleep durations. To address this concern, we performed further statistical analyses. We first estimated the likelihood of finding a significant difference between the laser-on and laser-off condition if the laser had no effect on REM sleep, using a bootstrapping analysis. Assuming our dataset comprises n mice with m_i REM sleep episodes each ($m_i = m_{i_on}$ laser-on bouts + m_{i_off} laser-off bouts), we shuffled the designations of laser-on and laser-off episodes by randomly selecting m_{i_on} REM episodes to assign as

“laser-on” and labeling the remaining *mi_off* episodes as “laser-off”. For bootstrapping, we randomly chose *n* mice with replacement, and selected for each mouse a random sample (with replacement) of *mi_on* “laser-on” REM bouts and *mi_off* “laser-off” bouts from the shuffled data. Next, we calculated for each mouse the difference between the mean durations of the resampled laser-on and laser-off bouts, and averaged these differences across all *n* mice to compute the laser-induced change in REM sleep duration for the given sample. We repeated this procedure over 10,000 iterations to generate a sampling distribution of the mean difference between shuffled laser-on and laser-off REM episodes (Δ Duration), which we used to determine the 95% confidence interval (CI) and calculate p-values. We found that the CIs estimated for the shuffled ChR2 and iC++ datasets both included 0, suggesting that if there were no relationship between the laser and the duration of REM sleep, we would not expect to find a significant difference between the laser-on and laser-off conditions. Repeating this analysis for the true (non-shuffled) datasets, we found no overlap between the CIs for the true and shuffled mean difference. In other words, the laser significantly prolonged REM sleep in ChR2 mice and shortened REM sleep in iC++ mice, compared to what would be expected by chance from an experiment with identical sample size and distribution of REM sleep durations. We also compared the ChR2 and iC++ groups with mice expressing eYFP, and found that the CIs for the experimental datasets did not overlap with the CIs estimated for their controls, further demonstrating that the observed differences between laser-on and laser-off REM sleep episodes are truly due to optogenetically induced changes in the activity of dmM CRH neurons. These analyses are summarized in the revised Results section (page 8, lines 180-190; page 9, lines 195-196).

(2) We agree about the importance of quantifying the effect of laser stimulation on the EEG power spectrum, and we have added this analysis to the revised **Supplementary Fig. 4**. We also describe the underlying statistical procedures in the revised Methods section (page 30, lines 721-736). Comparing between spontaneous and laser-triggered REM sleep in ChR2 mice, we found no significant difference in the mean normalized power of the delta (δ , 0.5–4 Hz), theta (θ , 6–10 Hz), sigma (σ , 11–15), or gamma (γ , 55–99) frequency bands (revised **Supplementary Fig. 4d**).

To quantify the impact of laser stimulation on the power spectral density (PSD) of the EEG, we computed the PSD and calculated the mean power in each frequency band with and without laser for ChR2 and eYFP mice (revised **Supplementary Fig. 4m**). For each brain state, we compared between laser-on and laser-off episodes within a given mouse group using two-way repeated measures ANOVA with laser and frequency band as within factors (top), and we compared the laser-induced changes in EEG power between ChR2 and eYFP mice using mixed ANOVA with virus and frequency band as between and within factors (bottom). We found that laser stimulation did not produce significant changes in the tested power bands of any brain state in ChR2 or eYFP mice, nor did the effect of the laser differ between control and experimental groups. Repeating this analysis to test the impact of chemogenetic activation (**Supplementary Fig. 6h**), we observed no significant differences in the power spectrum of either hM3D(Gq) or mCherry mice after injection with 0.25 mg/kg CNO vs saline, and found that CNO does not differently affect the EEG of experimental mice compared with control animals. However, for chemogenetic inhibition using 5 mg/kg CNO, we found significant interactions between drug (saline vs CNO) and frequency band power during IS in hM4D(Gi) mice and during NREM sleep in mCherry mice (**Supplementary Fig. 7f**). Post-hoc t-tests with Holm correction did not reveal a significant difference in the power of any frequency band between saline and CNO trials for either mouse group, though in the CNO condition there was a trending increase ($P = 0.10$) of the delta power during NREM sleep for mCherry mice. This observation is consistent with previous reports of enhanced low frequency power during NREM sleep due to

non-specific effects of CNO (Traut et al., 2022). Finally, as recommended by the reviewer, we now use the same y-range for all graphs representing PSDs.

(3) In the original **Supplementary Fig. 6c-g**, we statistically compared the effects of saline, 0.25 mg/kg CNO, and 5 mg/kg CNO on the sleep architecture in mCherry mice, repeating the analyses performed in experimental mice (original **Fig. 5**) to control for off-target effects of CNO. As suggested by the reviewer in the below comment (Q32), we now directly compare the impact of 0.25 mg/kg CNO between hM3D(Gq) and mCherry mice (revised **Fig. 5d-f**; revised **Supplementary Fig. 6c-e**), and we compare the effects of 5 mg/kg CNO between hM4D(Gi) and mCherry mice (revised **Fig. 5h-j**; revised **Supplementary Fig. 7a-c**). Importantly, since chemogenetic manipulation produces hours-long changes in neuronal activity, in these analyses we quantified the effects of CNO over the entire five hour recording period. In contrast, for the optogenetic protocol in **Fig. 3** and **Supplementary Fig. 4**, laser stimulation was precisely delivered in 120 s intervals separated by at least 15 minute intervals. Therefore, to examine the effect of the laser on the brain state of control eYFP mice, we compared the percentage of each brain state during the stimulation interval (0 – 120 s) with that during the preceding baseline interval (-120 – 0 s) (revised **Supplementary Fig. 4d**).

(4) We believe that the reviewer's comment regarding the lack of the modulation of P-waves is related to their previous concern about the relative timing between P-waves and the activation of dmM CRH neurons: There appears to be a discrepancy between the fast optogenetic induction of P-waves (within tens of milliseconds; **Fig. 4h**) and the slow rise in the calcium signal peaking much later (**Fig. 2i**), thus precluding a potential role of the dmM CRH neurons in regulating P-waves. However, as mentioned above, this discrepancy can be resolved when taking into account the slow kinetics of the GCaMP6s sensor and the fact that the timing of action potentials is best reflected by the point where the calcium signal starts rising, rather than the peak of the signal. Taking the slow GCaMP6s dynamics into account, the calcium imaging experiments thus provide correlational evidence for a role of the dmM CRH calcium activity in P-wave regulation.

To test the causal relationship between dmM CRH neurons and P-waves, we optogenetically and chemogenetically manipulated the activity of these neurons. Using optogenetics, we found that brief laser activation of this population was reliably followed by P-waves within 100 ms (**Fig. 4**). As suggested by the reviewer in a previous comment, we have expanded our control analyses by including experiments in eYFP animals subjected to the same laser protocol (**Supplementary Fig. 5g**) and we have clarified our statistical analyses, where we determined the likelihood to observe a quick succession of laser pulses and P-waves simply by chance through temporally shuffling the laser pulses ("Shuffled laser" in **Fig. 4g,h**). These control experiments and analyses corroborate that short activation of dmM CRH neurons directly elicits P-waves. Finally, a role of the CRH neuron activity in P-wave regulation is further supported by the chemogenetic experiments, showing that sustained excitation of this population leads to REM sleep with an increased frequency of P-waves, while chemogenetic inhibition reduces their frequency (**Fig. 5**).

We agree with the reviewer that the causal link between dmM CRH neuron activity and P-waves should be examined in a state-specific manner. The optogenetic protocol in **Fig. 4** used step pulses at a fixed interval, independent of brain state. To analyze state-dependent effects of brief optogenetic activation, we post-hoc tested the effects of the laser pulses depending on whether they occurred during Wake, NREM sleep, REM sleep, or IS (e.g. **Fig. 4g**) or analyzed the effects specifically for REM sleep (e.g. **Fig. 4c-e**), where P-waves are by far most prominent. Though it is reasonable to consider a closed-loop experiment with laser pulses delivered only during REM sleep (or other states), we believe our protocol is equally suitable for

establishing a direct functional relationship between dmM CRH neuron activation, P-wave induction, and the current brain state.

Q32 (Reviewer 3)

5-Recent work investigated the effect of CNO on sleep and demonstrated modulation of sleep in animals with control vectors (Traut et al. 2022). Although the authors claimed to show the mCherry + CNO, Suppl Figure 6d-h refers to mCherry + NaCl. Thus, authors must provide the appropriate control (e.g. AVV-mCherry + CNO; both at concentrations tested) and use proper statistical analysis, i.e, between experimental groups: NaCl, mCherry+ CNO and mCherry-h3MDGs.

Response Text

There may have been a misunderstanding related to the original **Supplementary Fig. 6d-h**. In addition to the results for mCherry mice treated with saline (NaCl), these figures also show the effects of injection with CNO, including both the 0.25 mg/kg dose used for activation and the 5 mg/kg dose used for inhibition. However, we agree with the reviewer that given recent findings that higher CNO doses can induce changes in sleep (Traut et al., 2022; Varin et al., 2018), direct comparisons of hM3D(Gq) and hM4D(Gi) animals with the appropriate mCherry controls are required to account for non-specific effects of CNO. Consequently, we have expanded the statistical analyses in **Fig. 5** and **Supplementary Figs. 6** and **7**, using mixed ANOVA with virus (hM3D(Gq) or hM4D(Gi) vs mCherry) and drug (saline vs CNO) as between and within factors, followed by pairwise t-tests with Holm correction.

Q33 (Reviewer 3)

6-P-waves have been classically associated with eye movement, as mentioned by the authors. Did they record the EOG (electrooculogram) by any chance? this would definitely strengthen the findings and the conclusions of the study.

Response Text

We agree with the reviewer that recording eye movements to test a potential role of the dmM CRH neurons in rapid eye movement regulation would be highly interesting. Our sleep recordings were all performed without EOG and this information is therefore unfortunately not available for analysis.

Q34 (Reviewer 3)

7-The discussion further raised additional side questions. The authors discussed that activation of CRH terminals in the NI could provide evidence to support a role in theta modulation. This is an interesting claim; however, it is not supported by any experimental data. If this is the author's hypothesis, they should provide supporting data. Furthermore, an open question remains as to whether CRH terminals modulate Theta and P-wave through distinct or the same circuit. Similarly, manipulating the terminals in the SubC may help elucidate the role of CRH neurons in -P-wave versus REM sleep. Such a dataset will further strengthen the study.

Response Text

Remaining from our study is the major open question of whether theta bursts and P-waves are regulated by the same downstream area, and therefore the same set of axon terminals, or whether these events are controlled by multiple projections to distinct postsynaptic areas. Based on previous literature and given the axonal projections of dmM CRH neurons (**Supplementary Fig. 1e-i** in the revised manuscript), we proposed that NI and SubC are likely candidate areas mediating the effect of dmM CRH activation on P-waves and theta bursts. We fully agree with the reviewer that more functional experiments would be required to prove a causal role of these projections in P-wave, theta burst or REM sleep regulation, but we believe that such experiments, although logical next steps, would be beyond the scope of the present revision. To make clear in the revised Discussion that our present data does not prove the role of the NI or SubC projections in theta burst or P-wave regulation and that the related statements are simply interesting directions for future work, we have rephrased the wording of the corresponding sentence to mention that “we speculate” (page 15, line 365).

Minor changes:

(1) *In suppl. Figure 1. CRH projections are difficult to appreciate in panels E and G. Please provide a sagittal or coronal section of the axonal distributions across the brain.*

Response Text

As suggested by the reviewer, we have expanded our viral tracing analysis to better characterize the distribution of dmM CRH axons across the brain. In the revised **Supplementary Fig. 1i**, we have included a series of coronal sections spaced approximately 400 μm apart along the anteroposterior (AP) axis, spanning the distance from the dmM CRH cell bodies (AP -6.5 mm) to their most anterior projection target (AP -4.6 mm). Examining each brain area with significant fluorescence labeling, we note that the densest fibers are observed in structures located in the dorsal pons, as shown with greater detail in **Supplementary Fig. 1g,h**.

(2) *Suppl. Figure 2 legend clearly states which neurons are being targeted. In panels b, c and d it is not clear if the average values represent all the transitions in one session, per animal of different recording sessions or between animals. Please specify.*

Response Text

We have realized that we have not provided sufficient detail for our statistical analyses in **Supplementary Fig. 2c-e**. To minimize the risk of photobleaching in our calcium imaging studies, recording sessions lasted for only 2 hours; this shorter time window consequently reduced the number of brain state transitions in our dataset, particularly given the relative infrequency of REM sleep episodes. Due to the limited number of transitions recorded for each mouse, the $\Delta F/F$ signals for all X \rightarrow Y transitions from all mice were collected and averaged together for statistical analysis. To clarify the analyses underlying the graphs in **Supplementary Fig. 2c-e**, we now clearly state in the legend of each subfigure the number of total transitions included in the plotted $\Delta F/F$ time course. In addition, we have realized that our original explanation of the statistical averaging procedure in the Methods section was ambiguously

worded, and we have updated the sentence to specify that the $\Delta F/F$ signals are calculated across all n transitions recorded over the course of the experiment (page 32, lines 776-777).

(3) *Suppl. Figure 4 g: the colour scale is missing.*

Response Text

We thank the reviewer for pointing out this omission. We have added a color legend to this graph (**Supplementary Fig. 4i** in the revised manuscript).

(4) *Suppl. Figures 6 a and b: y-axis should be homogenized.*

Response Text

As suggested by the reviewer, we now use the same range on the y-axes for these graphs (**Supplementary Fig. 6h** and **Supplementary Fig. 7f** in the revised manuscript).

Chen, T.-W. *et al.* Ultra-sensitive fluorescent proteins for imaging neuronal activity. *Nature* **499**, 295–300 (2013).

Lerner, T. N. *et al.* Intact-Brain Analyses Reveal Distinct Information Carried by SNc Dopamine Subcircuits. *Cell* **162**, 635–647 (2015).

Siegle, J. H. *et al.* Survey of spiking in the mouse visual system reveals functional hierarchy. *Nature* **592**, 86–92 (2021).

Traut, J. *et al.* Effects of clozapine-N-oxide and compound 21 on sleep in laboratory mice. 2022.02.01.478652 Preprint at <https://doi.org/10.1101/2022.02.01.478652> (2022).

Varin, C., Luppi, P.-H. & Fort, P. Melanin-concentrating hormone-expressing neurons adjust slow-wave sleep dynamics to catalyze paradoxical (REM) sleep. *Sleep* **41**, zsy068 (2018).

REVIEWERS' COMMENTS

Reviewer #1 (Remarks to the Author):

I thank the authors for their detailed and convincing answers, the additional instructive analyses, as well as the changes made in the manuscript in response to my comments. I have no further comments.

Reviewer #2 (Remarks to the Author):

From my point of view and after a dutiful reading of the revised manuscript and included rebuttal letter, questions and problems related to experimental methodologies and data analyses I raised during my first expertise (sometimes on same wavelength as the two other reviewers) have been addressed point by point in a systematic, documented, rational and extremely satisfactory manner. This is an excellent complementary work carried out by the authors which significantly improves the overall quality and scientific scope of this experimental study in mice, with an interesting knowledge advance in both basic sleep research and the neural networks responsible for REM sleep expression.

Reviewer #3 (Remarks to the Author):

The responses by the authors adequately address the concerns raised in the previous review. I have no further concerns.